# GENERATIVE MODELLING
# WITH INVERSE HEAT DISSIPATION

**Severi Rissanen, Markus Heinonen & Arno Solin**
Department of Computer Science
Aalto University
severi.rissanen@aalto.fi

## ABSTRACT

While diffusion models have shown great success in image generation, their noise-inverting generative process does not explicitly consider the structure of images, such as their inherent multi-scale nature. Inspired by diffusion models and the empirical success of coarse-to-fine modelling, we propose a new diffusion-like model that generates images through stochastically reversing the heat equation, a PDE that locally erases fine-scale information when run over the 2D plane of the image. We interpret the solution of the forward heat equation with constant additive noise as a variational approximation in the diffusion latent variable model. Our new model shows emergent qualitative properties not seen in standard diffusion models, such as disentanglement of overall colour and shape in images. Spectral analysis on natural images highlights connections to diffusion models and reveals an implicit coarse-to-fine inductive bias in them.

## 1 INTRODUCTION

Diffusion models have recently become highly successful in generative modelling tasks (Ho et al., 2020; Song et al., 2021d; Dhariwal & Nichol, 2021). They are defined by a forward process that erases the original image information content and a reverse process that generates images iteratively. The forward and reverse processes of standard diffusion models do not explicitly consider the inductive biases of natural images, such as their multi-scale nature. In other successful generative modelling settings, such as in GANs (Goodfellow et al., 2014), taking multiple resolutions explicitly into account has resulted in dramatic improvements (Karras et al., 2018; 2021). This paper investigates how to incorporate the inductive biases of natural images, particularly their multi-resolution nature, into the generative sequence of diffusion-like iterative generative models.

The concept of resolution itself in deep learning methods has received less attention, and usually, scaling is based on simple pixel sub-sampling pyramids, halving the resolution per step. In classical computer vision, another approach is the so-called Gaussian scale-space (Iijima, 1962; Witkin, 1987; Babaud et al., 1986; Koenderink, 1984), where lower-resolution versions of an image are obtained by running the heat equation, a partial differential equation (PDE, see Fig. 1) that describes the dissipation of heat, over the image. Similarly to subsampling, the heat equation averages out the images and removes fine detail, but an arbitrary amount of effective resolutions is allowed without explicitly decreasing the number of pixels. The scale-space adheres to a set of scale-space axioms, such as rotational symmetry, invariance to shifts in the input image, and scale invariance (Koenderink, 1984; Babaud et al.,

**Information melting forward process** $\frac{\partial \mathbf{u}}{\partial t} = \Delta \mathbf{u}$

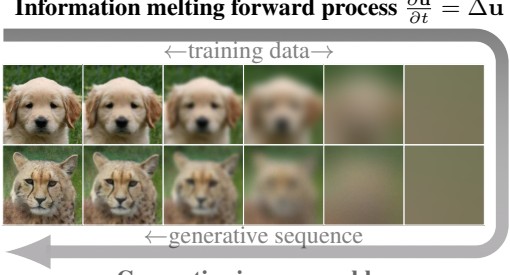

$\leftarrow$training data$\rightarrow$

$\leftarrow$generative sequence$\rightarrow$

**Generative inverse problem**

Figure 1: Example of the forward process (during training) and the generative inverse process (for sample generation).

**Standard diffusion model**

Non-invertible forward process

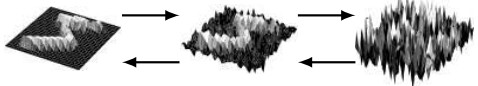

Generative reverse process

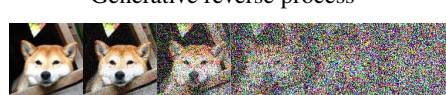

**Inverse heat dissipation model**

Non-invertible forward process

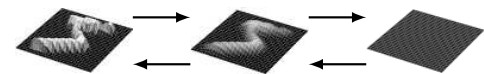

Generative reverse process

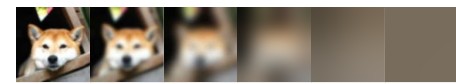

Figure 2: Comparison of generation by generative denoising and inverse heat diffusion, where the focus of the forward process is in the *pixel space* in the left and the *2D image plane* on the right.

1986), also linking to how early biological vision represents signals (Lindeberg, 2013a;b). While scale-space has been utilized in the context of CNN architectures (Worrall & Welling, 2019; Pintea et al., 2021), it has not been considered in generative models.

We investigate inductive biases in diffusion-type generative models by proposing a generative model based on directly reversing the heat equation and thus increasing the effective image resolution, illustrated in Fig. 2. We call it the inverse heat dissipation model (IHDM). The intuition is that as the original image information content is erased in the forward process, a corresponding stochastic reverse process produces multiple plausible reconstructions, defining a generative model. Samples from the prior distribution are easy obtain due to the low dimensionality of averaged images, and we adopt a training data based kernel density estimate.

Our main contributions are: *(i)* We show how to realise the idea of generative modelling with inverse heat dissipation by interpreting a solution of the heat equation with small additive noise as an inference process in a diffusion-like latent variable model. *(ii)* We investigate emergent properties of the heat equation-based model: (a) disentanglement of overall colour and image shape (b) smooth interpolation, (c) the forward process inducing simplicity to the learned neural net function, and (d) potential for data efficiency. *(iii)* By analysing the power spectral density of natural images, we show that standard diffusion models implicitly perform a different type of coarse-to-fine generation, shedding light on their inductive biases, and highlighting connections and differences between our model and standard diffusion models. Code for the methods in this paper is available at: `https://github.com/AaltoML/generative-inverse-heat-dissipation`.

## 2 METHODS

The main characteristic of the forward process is that it averages out the images in the data set, contracting them into a lower-dimensional subspace (see Fig. 2 right). We define it with the heat equation, a linear partial differential equation (PDE) that describes the dissipation of heat:

$$\text{Forward PDE model:} \qquad \frac{\partial}{\partial t}u(x,y,t) = \Delta u(x,y,t), \qquad (1)$$

where $u : \mathbb{R}^2 \times \mathbb{R}_+ \to \mathbb{R}$ is the idealized, continuous 2D plane of one channel of the image, and $\Delta = \nabla^2$ is the Laplace operator. The process is run for each colour channel separately. We use Neumann boundary conditions ($\partial u/\partial x = \partial u/\partial y = 0$) with zero-derivatives at boundaries of the image bounding box. This means that as $t \to \infty$, each colour channel is averaged out to the mean of the original colour intensities in the image. Thus, the image is projected to $\mathbb{R}^3$. In principle, the heat equation could be exactly reversible with infinite numerical precision, but this is not the case in practice with finite numerical accuracy due to the fundamental ill-posed nature of the inverse heat equation. Another way to view it is that with any amount of observation noise added on top of the averaged image, the original image cannot be recovered exactly (see Kaipio & Somersalo, 2006, for discussion).

The PDE model in Eq. (1) can be formally written in evolution equation form as $u(x, y, t) = \mathcal{F}(t) u(x, y, t)|_{t=t_0}$, where $\mathcal{F}(t) = \exp[(t - t_0) \Delta]$ is an evolution operator given in terms of the operator exponential function (see, *e.g.*, Da Prato & Zabczyk, 1992). We can use this general formulation to efficiently solve the equation using the eigenbasis of the Laplace operator. Since we

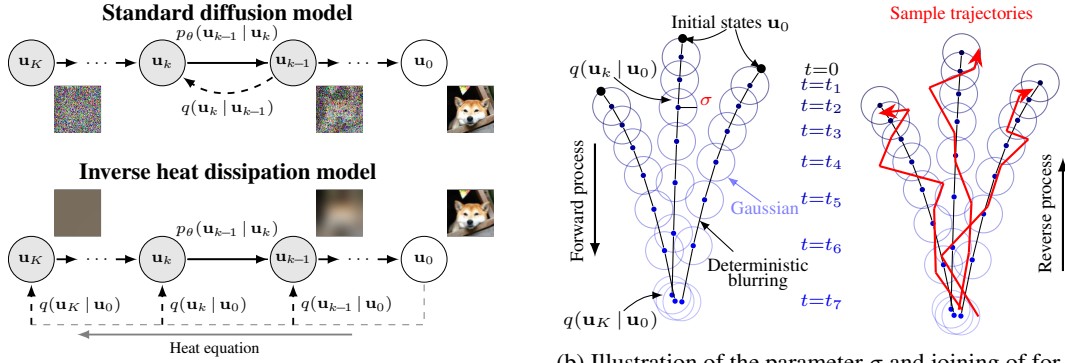

(a) Graphical models

(b) Illustration of the parameter $\sigma$ and joining of forward process paths, enabling branching in the reverse

Figure 3: (a) Graphical model of IHDM vs. a standard diffusion model, highlighting the factorized inference process. (b) A sketch of how the stochasticity of $q(\mathbf{u}_k \mid \mathbf{u}_0)$, controlled by $\sigma$, allows the forward process paths to effectively merge in the probabilistic model. As opposed to trying to invert the fully deterministic heat equation, this makes the reverse conditional distributions well-defined.

use Neumann boundary conditions, the eigenbasis is a cosine basis (see full details in App. A.1). The observed finite-resolution image lies on a grid, meaning that the spectrum has a natural cut-off frequency (Nyquist limit). Thus, we can formally write the operator in terms of a (finite) eigendecomposition $\Delta \triangleq \mathbf{V}\boldsymbol{\Lambda}\mathbf{V}^\top$, where $\mathbf{V}^\top$ is the cosine basis projection matrix, and $\boldsymbol{\Lambda}$ is a diagonal matrix with negative squared frequencies on the diagonal. The initial state is then projected on to the basis with the discrete cosine transform ($\tilde{\mathbf{u}} = \mathbf{V}^\top \mathbf{u} = \mathrm{DCT}(\mathbf{u})$) in $\mathcal{O}(N \log N)$ time. The solution is given by the finite-dimensional evolution model, describing the decay of frequencies

$$\mathbf{u}(t) = \mathbf{F}(t)\,\mathbf{u}(0) = \exp(\mathbf{V}\boldsymbol{\Lambda}\mathbf{V}^\top t)\,\mathbf{u}(0) = \mathbf{V}\exp(\boldsymbol{\Lambda}t)\mathbf{V}^\top \mathbf{u}(0) \;\Leftrightarrow\; \tilde{\mathbf{u}}(t) = \exp(\boldsymbol{\Lambda}t)\tilde{\mathbf{u}}(0), \quad (2)$$

where $\mathbf{F}(t) \in \mathbb{R}^{N \times N}$ (not expanded in practice) is the transition model and $\mathbf{u}(0)$ the initial state. The diagonal terms of $\boldsymbol{\Lambda}$ are the negative squared frequencies $-\lambda_{n,m} = -\pi^2(n^2/W^2 + m^2/H^2)$, where $W$ and $H$ are the width and height of the image in pixels, $n = 0, \ldots, W-1$ and $m = 0, \ldots, H-1$. As $\boldsymbol{\Lambda}$ is diagonal, the solution is fast to evaluate and implementable with a few lines of code, see App. A.1.

The heat equation has a correspondence to the Gaussian blur operator in image processing: In an infinite plane, simulating the heat equation up to time $t$ equivalent to a convolution with a Gaussian kernel with variance $\sigma_B^2 = 2t$ (Bredies & Lorenz, 2018). The heat equation has the advantage that it exposes the theoretical properties of the process, $e.g.$, the frequency behaviour and boundary conditions, and is potentially better generalizable to other forward processes and data domains.

## 2.1 GENERATIVE MODEL FORMULATION

We seek to define a probabilistic model that stochastically reverses the heat equation. Even if one could formally invert Eq. (2), we are not interested in the deterministic inverse problem per se, but in formalizing a generative model with characteristics given by the forward problem. The generative process should also branch into multiple plausible reverse paths. We formally break the reversibility by introducing a small amount of noise with standard deviation $\sigma$ in the forward process and incorporate it into the general mathematical framework for diffusion models (Sohl-Dickstein et al., 2015). Effectively, this sets a lower limit to how low the frequency components can decay before turning into noise. The idea makes the reverse conditional distributions probabilistically well defined, as illustrated in Fig. 3b. We define the time steps $t_1, t_2, \ldots, t_K$ that correspond to latent variables $\mathbf{u}_k$, each of which has the same dimensionality as the data $\mathbf{u}_0$. Our forward process, or formally the variational approximation in the latent variable model, is defined as

Forward process /
Inference distribution
$$q(\mathbf{u}_{1:K} \mid \mathbf{u}_0) = \prod_{k=1}^{K} q(\mathbf{u}_k \mid \mathbf{u}_0) = \prod_{k=1}^{K} \mathcal{N}(\mathbf{u}_k \mid \mathbf{F}(t_k)\,\mathbf{u}_0, \sigma^2 \mathbf{I}), \qquad (3)$$

where $\mathbf{F}(t_k)$ is the linear transformation corresponding to simulating the heat equation until time $t_k$ and the standard deviation $\sigma$ is a small constant (*e.g.*, 0.01 if data is scaled to $[0, 1]$). Note that instead of having the forward be a Markov chain as in regular diffusion models, we factorize the noise in a way that intuitively treats it as observation noise on top of the deterministic heat equation, as also visualized in Fig. 3a. The generative, or reverse process, is a Markov chain that starts with the prior state $\mathbf{u}_K$ and ends at the observed variable $\mathbf{u}_0$. We define it with Gaussian conditional distributions:

$$\text{Reverse process / Generative model} \quad p_\theta(\mathbf{u}_{0:K}) = p(\mathbf{u}_K) \prod_{k=1}^{K} p_\theta(\mathbf{u}_{k-1} \,|\, \mathbf{u}_k) = p(\mathbf{u}_K) \prod_{k=1}^{K} \mathcal{N}(\mathbf{u}_{k-1} \,|\, \boldsymbol{\mu}_\theta(\mathbf{u}_k, k), \delta^2 \mathbf{I}), \quad (4)$$

where $\theta$ are model parameters and $\delta$ is the standard deviation of the noise added during sampling. We show the whole structure in Fig. 3a, where we highlight the structural difference to standard diffusion models. Fig. 3b provides intuition on the noise parameters $\sigma$ and $\delta$; the noise $\sigma$ acts as an error tolerance or relaxation parameter that measures how close two blurred images have to be to become essentially indistinguishable. With a non-zero $\sigma$, an initial state $\mathbf{u}_K$ has a formal probability of going along different paths. The parameter $\delta$ is a free hyperparameter that controls the sampling stochasticity, which in turn defines the trajectory.

Our goal is to maximize marginal likelihood of the data $p(\mathbf{u}_0) = \int p_\theta(\mathbf{u}_0 \,|\, \mathbf{u}_{1:K}) \, p_\theta(\mathbf{u}_{1:K}) \, \mathrm{d}\mathbf{u}_{1:K}$. Taking the VAE-type evidence lower bound for the marginal likelihood with the generative and inference distributions defined, we get

$$-\log p_\theta(\mathbf{u}_0) \leq \mathbb{E}_q\left[ -\log \frac{p_\theta(\mathbf{u}_{0:K})}{q(\mathbf{u}_{1:K} \,|\, \mathbf{u}_0)} \right] \tag{5}$$

$$= \mathbb{E}_q\left[ -\log \frac{p_\theta(\mathbf{u}_K)}{q(\mathbf{u}_K \,|\, \mathbf{u}_0)} - \sum_{k=2}^{K} \log \frac{p_\theta(\mathbf{u}_{k-1} \,|\, \mathbf{u}_k)}{q(\mathbf{u}_{k-1} \,|\, \mathbf{u}_0)} - \log p_\theta(\mathbf{u}_0 \,|\, \mathbf{u}_1) \right] \tag{6}$$

$$= \mathbb{E}_q\bigg[ \underbrace{D_{\mathrm{KL}}[q(\mathbf{u}_K \,|\, \mathbf{u}_0) \,\|\, p(\mathbf{u}_K)]}_{L_K} + \sum_{k=2}^{K} \underbrace{D_{\mathrm{KL}}[q(\mathbf{u}_{k-1} \,|\, \mathbf{u}_0) \,\|\, p_\theta(\mathbf{u}_{k-1} \,|\, \mathbf{u}_k)]}_{L_{k-1}} \underbrace{- \log p_\theta(\mathbf{u}_0 \,|\, \mathbf{u}_1)}_{L_0} \bigg], \tag{7}$$

where the different parts of the process factorize in a similar, although somewhat simpler, way as in diffusion probabilistic models (Sohl-Dickstein et al., 2015; Ho et al., 2020). The terms $L_{k-1}$ are KL divergences between Gaussian distributions

$$\mathbb{E}_q[L_{k-1}] = \mathbb{E}_q\big[ D_{\mathrm{KL}}[q(\mathbf{u}_{k-1} \,|\, \mathbf{u}_0) \,\|\, p_\theta(\mathbf{u}_{k-1} | \mathbf{u}_k)] \big] \tag{8}$$

$$= \frac{1}{2}\left( \frac{\sigma^2}{\delta^2} N - N + \frac{1}{\delta^2} \mathbb{E}_{q(\mathbf{u}_k \,|\, \mathbf{u}_0)}\Big[ \| \underbrace{\boldsymbol{\mu}_\theta(\mathbf{u}_k, k) - \mathbf{F}(t_{k-1}) \, \mathbf{u}_0}_{f_\theta(\mathbf{u}_k, k) - (\mathbf{F}(t_{k-1}) \, \mathbf{u}_0 - \mathbf{u}_k)} \|_2^2 \Big] + 2N \log \frac{\delta}{\sigma} \right), \tag{9}$$

$$\mathbb{E}_q[L_0] = \mathbb{E}_q[-\log p_\theta(\mathbf{u}_0 \,|\, \mathbf{u}_1)] = \frac{1}{2\delta^2} \mathbb{E}_{q(\mathbf{u}_1 \,|\, \mathbf{u}_0)}\Big[ \| \underbrace{\boldsymbol{\mu}_\theta(\mathbf{u}_1, 1) - \mathbf{u}_0}_{f_\theta(\mathbf{u}_1, 1) - (\mathbf{u}_0 - \mathbf{u}_1)} \|_2^2 \Big] + N \log(\delta \sqrt{2\pi}), \tag{10}$$

where $N$ is the number of pixels in the image. We evaluate the loss function with one Monte Carlo sample from the inference distribution $q(\mathbf{u}_{1:K} \,|\, \mathbf{u}_0)$. The losses on all levels are direct MSE losses where we predict a slightly less blurred image from a blurred image that has added noise with variance $\sigma^2$. The sampling proceeds by alternating the mean update steps from the neural network and the addition of Gaussian noise with variance $\delta$. We summarize the training process in Alg. 1, and the sampling process in Alg. 2, both of which are straightforward to implement. In practice, the algorithms mean that we train the neural net to deblur with noise-injection regularization, and sampling consists of alternating deblurring and adding noise. We further parametrize $\boldsymbol{\mu}_\theta(\mathbf{u}_k, k)$ with a skip connection such that $\boldsymbol{\mu}_\theta(\mathbf{u}_k, k) = \mathbf{u}_k + f_\theta(\mathbf{u}_k, k)$, which stabilizes training. The motivation is that we seek to take a small step backwards in a differential equation. With the skip connection, the loss functions in Eq. (9) and Eq. (10) resemble the denoising score matching objective, except that we are not predicting the denoised version of $\mathbf{u}_k$, but a less blurry $\mathbf{u}_{k-1}$.

**Prior.** We can use any standard density estimation technique for the prior distribution $p(\mathbf{u}_K)$ since the blurred images are effectively very low-dimensional. We use a Gaussian kernel density estimate with variance $\delta^2$, which is a reasonable estimate if the blurred images at level $K$ are low-dimensional and close enough to each other. We obtain samples by taking a training example, blurring it with

| **Algorithm 1** Loss function for a single data point | **Algorithm 2** Sampling |
|---|---|

**Algorithm 1** Loss function for a single data point

$\mathbf{u}_0 \sim$ Sample from training data
$k \sim$ Sample from $\{1, \dots, K\}$
$\mathbf{u}_k \leftarrow \mathbf{F}(t_k)\mathbf{u}_0$ ▷ see Eq. (2)
$\mathbf{u}_{k-1} \leftarrow \mathbf{F}(t_{k-1})\mathbf{u}_0$ ▷ see Eq. (2)
$\boldsymbol{\varepsilon} \sim \mathcal{N}(\mathbf{0}, \sigma^2\mathbf{I})$ ▷ Training noise
$\hat{\mathbf{u}}_k \leftarrow \mathbf{u}_k + \boldsymbol{\varepsilon}$ ▷ Perturb image
Loss $\leftarrow \|\boldsymbol{\mu}_\theta(\hat{\mathbf{u}}_k, k) - \mathbf{u}_{k-1}\|_2^2$

**Algorithm 2** Sampling

$k \leftarrow K$ ▷ Start from terminal state
$\mathbf{u} \sim p(\mathbf{u}_K)$ ▷ Sample from the blurry prior
**while** $k > 0$ **do**
  $\boldsymbol{\varepsilon}_k \sim \mathcal{N}(\mathbf{0}, \delta^2\mathbf{I})$ ▷ Sampling noise
  $\mathbf{u} \leftarrow \boldsymbol{\mu}_\theta(\mathbf{u}, k) + \boldsymbol{\varepsilon}_k$ ▷ Reverse step + noise
  $k \leftarrow k - 1$
**end while**

$\mathbf{F}(t_K)$, and adding noise with variance $\delta^2$. Using a kernel density estimate means that the term $L_K$ is constant but also tricky to calculate efficiently for log-likelihood evaluation due to the high-dimensional integral and multiple components in the kernel density estimate. In App. A.3, we provide a further variational upper bound on $L_K$ that can be evaluated without numerical integration.

**Asymptotics.** While the model introduces the desirable explicit multi-scale behaviour, we also have to drop some other established results related to diffusion models, such as theoretical guarantees about Gaussian reverse transitions being optimal in the limit of infinite steps. To provide more intuition, we point out the following connection to the early score-based generative modelling work (Song & Ermon, 2019; 2020): In the limit $K \to \infty$, the loss function $L_k$ becomes equivalent to the denoising score matching loss with noise level $\sigma$, and if $\delta = \sqrt{2}\sigma$, the sampling procedure is equivalent to running Langevin dynamics sampling with a certain step size on a given blur level. Generation then happens by slow annealing of sampling toward a less blurry distribution. In practice, we take directed steps backwards in the heat equation instead and do not limit to $\delta = \sqrt{2}\sigma$, but the result gives intuition for why a Gaussian transition is a good choice and a first guess at the correct ratio of $\delta/\sigma$ (full details in App. A.4).

## 2.2 IMPLICIT COARSE-TO-FINE GENERATION IN DIFFUSION MODELS

The frequency behaviour of natural images clarifies connections and differences between the new model and standard diffusion models. It also explains and characterises the well-known phenomenon that, in practice, diffusion models tend to create informative content in a coarse-to-fine fashion. The power spectral density (PSD) of natural images obeys an approximate power law $1/f^\alpha$, where often $\alpha \approx 2$ (van der Schaaf & van Hateren, 1996; Hyvärinen et al., 2009). When displayed on a log-log scale, the power spectral density is thus approximately a straight line. In App. A.5, we show that if we add isotropic Gaussian noise, the PSD of the noise and the PSD of the original image are additive in expectation. Thus, the highest frequencies get drowned out by the noise while the lower frequencies stay intact. When continuing the process, the noise masks more frequencies until the lowest frequencies have disappeared, as visualised in the red-coloured PSDs in Fig. 4. Thus, in the reverse

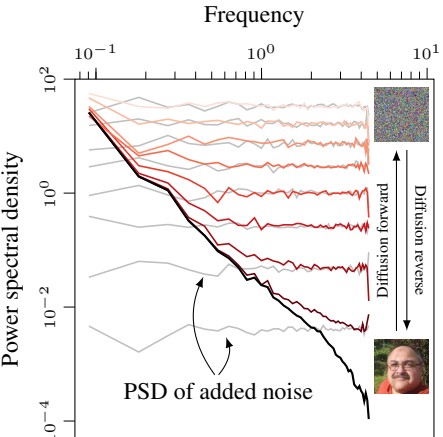

Figure 4: The $1/f^\alpha$ power spectral density in natural images induces an implicit coarse-to-fine inductive bias in diffusion models.

process, the diffusion model generates frequencies starting from the coarse-grained structure and progressing toward fine details. App. B.6 contains more details on the PSD calculations. Concurrently, the implicit spectral inductive bias was also noted in Kreis et al. (2022).

The result also shows differences between our model and standard diffusion models. While the frequency content in standard diffusion models is implicitly removed by drowning it out in noise, we do it explicitly by decaying the highest frequencies faster than the lower ones, as noted in Eq. (2). The noise level $\sigma$ sets a floor for the frequency components. Since the frequencies decay at rates corresponding to the heat equation, our process results in an explicit range of effective resolutions.

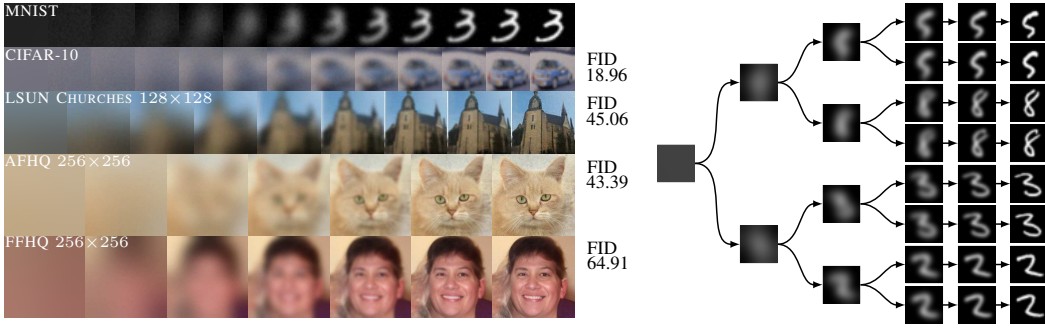

(a) Examples of generative sequences          (b) Hierarchical generation process

Figure 5: (a) Generation sequences for different data sets. Generation starts from a flat image and adds progressively more detail. The FID scores are next to the sequences. (b) The process is stochastic, and any given image in the image can progress towards multiple directions as shown on the right. More uncurated generated samples in App. D.1.

## 3 EXPERIMENTS

We showcase generative sequences, quantitative evaluation, and analyse the noise hyperparameters $\sigma$ and $\delta$. We then study emergent properties, starting with the overall colour and other features becoming disentangled. Next, contrary to standard diffusion models, interpolations in the full latent $\mathbf{u}_{1:K}$ are smooth. We also show that the forward heat process induces structure to the function learned by the neural net. Finally, we show that the model can generalise just from the first 20 MNIST digits.

**Architecture and hyperparameters.** Similarly to many recent works on diffusion models (Ho et al., 2020; Song et al., 2021d; Nichol & Dhariwal, 2021; Dhariwal & Nichol, 2021), we use a U-Net architecture (Ronneberger et al., 2015) with residual blocks and self-attention layers in the low-resolution feature maps. We list the architectural details for different data sets in App. B. We choose $\sigma = 0.01$ (data scaled to [0,1]), although the model is not particularly sensitive to the value, as shown in App. C.1. We use $K = 100$ iteration steps on MNIST (LeCun et al., 1998) 200 steps on CIFAR-10 (Krizhevsky, 2009), AFHQ (Choi et al., 2020), and FFHQ (Karras et al., 2019), and 400 on LSUN-CHURCHES (Yu et al., 2015). The time $t_k$ in in the inference process is spaced logarithmically from near zero to $t_K = \sigma_{B,\max}^2/2$, where $\sigma_{B,\max}$ is the effective length-scale of blurring at the end of the process as described in the beginning of Sec. 2. We set it to half the width of the image in all experiments unless mentioned otherwise. For CIFAR-10, we set it to 24 pixel-widths. We do not add noise at the last step of sampling since that cannot increase image quality.

**Generative sequences.** Fig. 5a showcases the generative sequences for the data sets. Generation starts from a blank image and progressively adds more fine-scale structure. Since the model is trained to reverse the heat equation, it effectively redistributes the original image mass to a random image with the same average colour. We visualize the stochasticity of the process in Fig. 5b, where we split the sequence into two at specified time steps. The large-scale structure gets determined in the beginning, and successive bifurcations lead to smaller and smaller changes in the output image. We present uncurated samples from all five data sets in App. D.1.

**Quantitative evaluation.** We evaluate the FID scores (Heusel et al., 2017) on the chosen data sets with IHDM, and list them on the right side of Fig. 5a. While the results are not yet as good as state-of-the-art diffusion models and GANs, we find the image quality promising. In particular, the CIFAR-10 FID of DDPM (Ho et al., 2020) was 3.17 in the original paper, and the current state-of-the art methods (Sauer et al., 2022) have FID scores of 1.85, compared to our 18.96. For qualitative comparison, we refer the reader to Fig.1. in (Ho et al., 2020) and our Fig. 19. In App. C.3, we also look at the marginal log-likelihood values and note that their optimal values do not correspond to optimal FID values when we vary the sampling noise parameter $\delta$.

**The importance of $\sigma$.** We already noted that a non-zero $\sigma$ is essential for the probabilistic model to be sensible. In App. C.5, we show empirically that with $\sigma = 0$, the model fails even on MNIST, producing random images. An intuition is that directly trying to reverse the heat equation without any regularization is unstable. In that sense, a non-zero $\sigma$ regularizes the reverse process.

**The effect of noise $\delta$.** As noted in Sec. 2.1, we can expect the optimal ratio of $\delta/\sigma$ to be larger than one. In Fig. 6, we see that $\delta = 0$ mainly sharpens the prior image into a shape. As we increase $\delta$ above $\sigma$, more and more detail appears until the images degenerate into noise. The optimal ratio $\delta/\sigma$ is approximately from 1.25 to 1.3, and 1.25 works as a good default value on all data sets. In App. C.1, we show that the optimal ratio of $\delta/\sigma$ does not depend on the absolute value of $\sigma$. We provide thorough $\delta$ sweeps in App. D.3. Deterministic sampling is possible in standard diffusion models through the probability flow ODE (Song et al., 2021d) or the DDIM (Song et al., 2021a) formalisms, but here it seems that the noise is a key factor in producing the information content.

**Disentangling colour and shape.** We show that the overall colour and other characteristics of the generated image can become disentangled with the model. Fixing the noise steps and only changing the prior $\mathbf{u}_K$, the process carves out a similar image with different average colours, visualized in Fig. 7a for a $128{\times}128$ FFHQ model with $\sigma_{B,\max} = 128$.

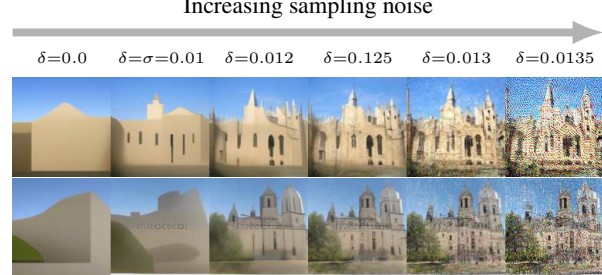

Increasing sampling noise

$\delta{=}0.0$   $\delta{=}\sigma{=}0.01$   $\delta{=}0.012$   $\delta{=}0.125$   $\delta{=}0.013$   $\delta{=}0.0135$

Figure 6: When the sampling noise parameter $\delta{=}0$, the model effectively sharpens the blurred image, with no new details added. As $\delta$ increases over the training noise $\sigma$, the results become more fine-grained and detailed, and finally noisy. Here, the noises are sampled once and scaled with $\delta$ for easier comparison.

**Smooth interpolation.** We can also use the latent $\mathbf{u}_{1:K}$ to interpolate between generated images in a perceptually smooth way, as shown in Fig. 7b. We use a linear interpolation on the input state $\mathbf{u}_K$ and a spherical interpolation on the noise (see App. D). As noted in Ho et al. (2020), the corresponding trick results in non-smooth interpolations with a standard diffusion model (DDPM, Fig. 9 in their Appendix). We showcase the result with a DDPM in Fig. 7b, where the interpolation passes through features that are not present in either endpoint. A connection to previous work is the StyleGAN architecture (Karras et al., 2019), where explicit modulation of resolution scales resulted in smoother interpolations. Our latent is similarly hierarchical, with different steps corresponding to different resolutions.

**Inductive bias on the learned function.** The heat forward process sets an inductive bias and encourages structure on the function the neural network tries to approximate. While we use the DCT-based approach to simulate the forward process, a more elementary approach would have been a grid-based finite difference approximation. It yields the following Euler step for the reverse heat equation:

$$u(x, y, t - \mathrm{d}t) \approx u(x, y, t) - \begin{pmatrix} 0 & 1 & 0 \\ 1 & -4 & 1 \\ 0 & 1 & 0 \end{pmatrix} * u(x, y, t)\, \mathrm{d}t. \tag{11}$$

Here, $(x, y) \in \mathbb{I}^2$ are locations on the discrete pixel grid of the image, and '$*$' is a discrete convolution with the given sharpening kernel that corresponds to the negative Laplace operator. Since small

Generation w.r.t. input state $\mathbf{u}_K$

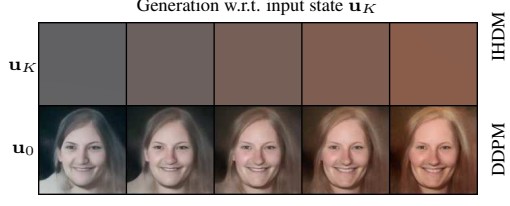

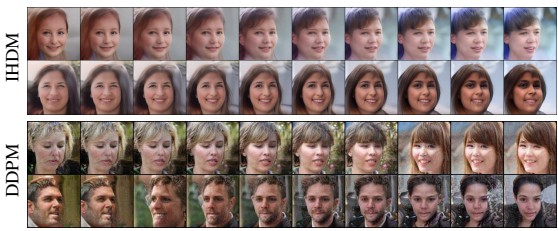

(a) Disentanglement of colour and other features

(b) Interpolations

Figure 7: (a) By setting the noise added during the generative process to a constant value and changing the starting image, which only contains information about the colour, the process creates highly similar faces with different overall colour schemes. (b) By also interpolating the noise steps in addition to the starting image, we obtain smooth interpolations between any two images in our model (IHDM). In a standard diffusion model (DDPM), the corresponding interpolation is non-smooth, passing through features that are not present in either endpoint, similarly as in Ho et al. (2020).

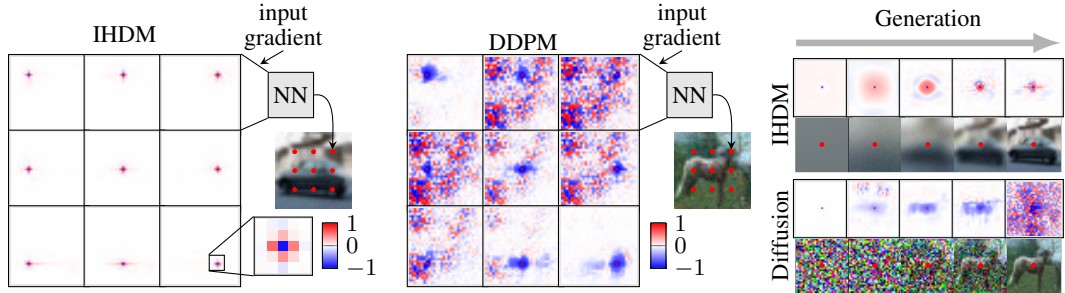

Figure 8: Left: The input gradients towards the end of the process for the first channels of the pixels highlighted in the image for our model (IHDM). Middle: The same for a standard diffusion model (DDPM). Right: The input gradients for a pixel at the center of the image with respect to number of generation steps. The images correspond to the generation step in question. The colours are scaled to a symlog scale with a linear cut-off at 0.1 on the left and middle and 0.01 on the right.

updates along the ideal reverse heat equation are approximately convolutions, this should be reflected in the learned function. We probe into the network by visualizing the input gradients in Fig. 8 for our model and a denoising diffusion probabilistic model (DDPM) for comparison. The learned functions are circularly symmetric and localized in the sense that perturbations at far-away pixels do not affect the output. In contrast, the dependence is more global and complex for DDPM. This reveals the well-localized nature of the IHDM compared to the DDPM (see also Fig. 13 in App. C.2 for the correlation structure in samples pre-convergence during training).

**Few-shot learning.** Since the inductive bias of the generative process sets a prior on natural images, the model can be highly data efficient. This is showcased in App. C.2 by training a standard DDPM and IHDM on the first 20 digits of MNIST. While DDPM either fails to produce convincing samples or overfits the data, IHDM can produce meaningful generalisation with only 20 data points.

## 4 RELATED WORK

Diffusion models (Sohl-Dickstein et al., 2015) have seen fast development since the first papers on score-based generative modelling (Song & Ermon, 2019; 2020). Score-based models were later shown to be connected with the original diffusion probabilistic models by (Ho et al., 2020). A reverse SDE formalism unified the framework in Song et al. (2021d) (later extended in Song et al., 2021b; Huang et al., 2021). Dhariwal & Nichol (2021) obtained state-of-the-art performance on ImageNet. Theoretical developments include the works by De Bortoli et al. (2021); Kingma et al. (2021). Excellent performance has been shown in other domains, such as audio (Chen et al., 2021a; Kong et al., 2021; Chen et al., 2021b). Recent ideas introduced for diffusion models also include training diffusion models on different levels of resolution and cascading them together to improve the model performance (Dhariwal & Nichol, 2021; Saharia et al., 2021; Ho et al., 2022; Ramesh et al., 2022; Saharia et al., 2022). The difference is that we consider the resolution-increasing as a basis of our model instead of a performance-boosting addition, and all computational steps increase the resolution.

Explicitly utilizing the hierarchy of resolutions in natural images has resulted in improved performance, *e.g.*, by training a stack of upsampling GAN layers that join to create a single image generator (Denton et al., 2015). Other famous examples are the progressive GAN (Karras et al., 2018) and later the StyleGAN architectures (Karras et al., 2019; 2020; 2021; Sauer et al., 2022). Resolution-based hierarchies in VAEs have brought them to rival GANs and other state-of-the-art models on different benchmarks (Razavi et al., 2019; Vahdat & Kautz, 2020; Child, 2021). Aside from these works, the architectures of other standard GANs and VAEs have been such that they start from low-resolution feature maps and increase the feature map resolution through upsampling layers.

Multi-scale ideas in the context of autoregressive models have also been proposed, starting with the multi-scale pixelRNN model (van den Oord et al., 2016). Possibly the closest one to our model is the work by Reed et al. (2017), where the authors factorize the joint distribution as a subsampling pyramid such that generation starts from a pixelated image and progresses towards a high-resolution version. Menick & Kalchbrenner (2019) suggest a similar method based on resolution and bit-depth upscaling.Previously, generative models, including diffusion models, have been utilized for image

deblurring (Kupyn et al., 2018; 2019; Asim et al., 2020; Whang et al., 2021), super-resolution (Ledig et al., 2017; Sajjadi et al., 2017; Dahl et al., 2017; Parmar et al., 2018; Chen et al., 2018; Saharia et al., 2021; Chung et al., 2022), and other types of inverse problems (Kawar et al., 2021; Chung et al., 2021; Jalal et al., 2021; Song et al., 2021c; Chung & Ye, 2022; Kawar et al., 2022). While our model effectively performs deblurring/super-resolution, the main difference to these works is that instead of using a pre-existing generative model to solve the inverse problem, we do the exact opposite and create a new generative model that directly reverses the heat equation with a simple MSE loss. Thus, our goal is not to do, *e.g.*, deblurring in itself, but to do unconditional generative modelling.

**Parallel work.** Concurrently, Lee et al. (2022) incorporate Gaussian blur into standard diffusion models, with a difference being that their work generalises the standard diffusion framework to include blur along with increasing noise, while we focus on creating a generative model that explicitly reverses the heat equation (blur process), and step out of the standard Markovian forward framework in the process. Another concurrent work is Daras et al. (2022), where the authors derive a generalised score-matching objective that allows incorporating Gaussian blur into the model training and sampling. We view these works as complementary: Lee et al. (2022) and Daras et al. (2022) show that it is possible to improve diffusion model performance by applying blur, whereas we investigate the inductive bias brought by a blurring process by proposing a model that generates images using deblurring in a highly explicit way. Work in combining the inductive biases provided by our model with the flexibility of standard diffusion could be a valuable direction for future research. Bansal et al. (2022) consider deterministic blurring and other operations as *cold diffusions*, and show an intriguing result that a method similar to the standard diffusion model training or sampling routines can be effectively used to approximately invert different deterministic operations deterministically, that is, to perform a type of conditional generation. They also propose to use a Gaussian mixture model prior for unconditional generation. The main difference is that their method maps a given blurry image deterministically to one possible solution. In contrast, our method produces a distribution of images. Hoogeboom & Salimans (2022) look into bridging inverse heat dissipation and denoising diffusion to have them meet in the middle.

## 5    DISCUSSION AND CONCLUSIONS

We have proposed a new approach for generative modelling by explicitly reversing the heat equation, with the goal of exploring inductive biases in diffusion-like models. An intriguing point of view on the model is that simply alternating a type of regularized deblurring and adding noise results in a generative model, without much need for hyperparameter tuning. We showed useful properties such as smooth interpolation, latent disentanglement, and data efficiency, highlighting the potential of the idea. We believe that our work is a first step in this direction, and that our results will allow future researchers to better reason about inductive biases in related generative models.

Potential future directions include more research into the probabilistic model formulation and its statistical properties, which could allow us to reason about how to improve the model. Based on the experiments, it appears that the inductive bias of the generative process effectively regularises the model compared to diffusion models. While IHDM seems to set a smoothness prior to images, diffusion models are free to even overfit slightly to parts of the data distribution if necessary. Ways to loosen this regularisation could be a fruitful direction of research. It is also possible that the used U-Net architecture has been optimized for standard diffusion models and is not ideally suited to our method. Research into the neural network could be valuable in improving the model.

The central idea here is also generalizable to other domains whenever a natural coarse-graining operator exists on the data. For instance, the heat equation can be defined straightforwardly on 1D audio data and on graphs, one could use the graph Laplacian to define a heat dissipation process based on Newton's law of cooling. This would also be in spirit with the geometric deep learning framework (Bronstein et al., 2017), where the Laplacian operator also plays a prominent role. Finally, a broader point is that our work opens up the potential for designing other types of generative sequences. While our heat dissipation process is arguably natural for images, others can be considered.

ACKNOWLEDGMENTS

We acknowledge funding from the Academy of Finland (334600, 339730, 324345) and the computational resources provided by the Aalto Science-IT project and CSC – IT Center for Science, Finland. We thank Paul Chang, Riccardo Mereu, Zheyang Shen, Valerii Iakovlev, Pashupati Hedge, and Ella Tamir for useful comments, and Shreyas Padhy for inspiring the data-efficiency experiment. We also thank Emiel Hoogeboom for discussions towards the end of the project.

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

# Appendix

## Table of Contents

# A    DERIVATION OF METHOD DETAILS

## A.1    NUMERICAL SOLUTION OF THE HEAT EQUATION WITH NEUMANN BOUNDARY CONDITIONS USING THE DISCRETE COSINE TRANSFORM

We expand the presentation of the approach taken in Sec. 2 and provide some further details on the derivation that highlights various appealing aspects of our approach. The material here is not novel, but is included for completeness.

We use the partial differential (PDE) model

$$\frac{\partial}{\partial t} u(x, y, t) = \Delta u(x, y, t), \tag{12}$$

where $u : \mathbb{R}^2 \times \mathbb{R}_+ \to \mathbb{R}$ is the idealized, continuous 2D plane of one channel of the image, and $\Delta = \nabla^2$ is the Laplace operator. Rather than discretizing the operator by a finite-difference scheme (see Eq. (11) for discussion in the main paper), we take an alternative approach where we solve the heat equation in the function space by projecting the problem onto the eigenbasis of the operator. The benefits are good numerical accuracy and scalability to large images. The workflow is as follows: *(i)* Rewrite the PDE as an evolution equation. *(ii)* Choose boundary conditions (Neumann, *i.e.*, the image 'averages out' as $t \to \infty$). *(iii)* Solve the associated eigenvalue problem (eigenbasis of the operator) which in this case results in a cosine basis. *(iv)* The image is on a regular grid, so we can use the Discrete Cosine Transform (DCT) for projection. This also means that there is a cut-off frequency for the problem that makes it finite-dimensional.

**Evolution equation**    Following the steps written out above, the PDE model in Eq. (12) can be formally written in evolution equation form as

$$u(x, y, t) = \mathcal{F}(t) \, u(x, y, t_0), \tag{13}$$

where $\mathcal{F}(t) = \exp[(t - t_0) \, \Delta]$ is an evolution operator given in terms of the operator exponential function (see, *e.g.*, Da Prato & Zabczyk, 1992). It is worth noting that instead of the Laplacian we could also consider more general (pseudo-)differential operators to describe more complicated dissipation processes.

**Choice of boundary conditions**    We choose to Neumann boundary conditions ($\partial u / \partial x = \partial u / \partial y = 0$) with zero-derivatives at boundaries of the image bounding box. This means that as $t \to \infty$, the image will be entirely averaged out to the mean of the original pixel values in the image. Formally:

$$\left. \begin{array}{ll} \dfrac{\partial}{\partial x} u(0, y, t) &= 0 \qquad \dfrac{\partial}{\partial x} u(W, y, t) = 0 \\[2mm] \dfrac{\partial}{\partial y} u(x, 0, t) &= 0 \qquad \dfrac{\partial}{\partial y} u(x, H, t) = 0 \end{array} \right\} \text{Boundary conditions} \tag{14}$$

$$u(x, y, t_0) = f(x, y), \qquad\qquad\qquad \text{Initial condition}, \tag{15}$$

where $W$ is the width of the image and $H$ is the height. We could choose some other boundary conditions as well, in many cases without loss of generality.

**Solve associated eigenvalue problem**    For the (negative) Laplace operator, which is positive definite and Hermitian, the solutions to the eigenvalue problem

$$-\Delta \phi_j(x, y) = \lambda_j \phi_j(x, y), \qquad\qquad \text{for } (x, y) \in \Omega, \tag{16}$$

$$\frac{\partial \phi_j(x, y)}{\partial x} = \frac{\partial \phi_j(x, y)}{\partial y} = 0, \qquad\qquad \text{for } (x, y) \in \partial\Omega, \tag{17}$$

yields orthonormal eigenfunctions $\phi_j(\cdot)$ with respect to the associated inner product, meaning that the corresponding operator is diagonalizable. If $\Omega$ is a rectangular domain and we consider the problem in Cartesian coordinates, the eigenbasis (solution to the eigenvalue problem above under the boundary conditions) turns out to be a (separable) cosine basis:

$$\phi_{n,m}(x, y) \sim \cos\left(\frac{\pi n x}{W}\right) \cos\left(\frac{\pi m y}{H}\right) \tag{18}$$

$$\lambda_{n,m} = \pi^2 \left(\frac{n^2}{W^2} + \frac{m^2}{H^2}\right). \tag{19}$$

Writing out the result of the evolution operator $\mathcal{F}(t) = \exp[(t - t_0)\,\Delta]$ explicitly,

$$u(x, y, t) = \sum_{n=0}^{\infty} \sum_{m=0}^{\infty} A_{nm} e^{-\pi^2 (\frac{n^2}{W^2} + \frac{m^2}{H^2})(t-t_0)} \cos\left(\frac{\pi n x}{W}\right) \cos\left(\frac{\pi n y}{H}\right), \tag{20}$$

which is a Fourier series where $A_{n,m}$ are the coefficients from projecting the initial state $u(x, y, t_0)$ to the eigenfunctions $\phi_{n,m}(x, y)$.

**Leverage the regularity of the image**  If we consider the pixels in the image to be samples from the underlying continuous surface $u(x, y)$ that lie on a regular grid, the projection onto this basis can be done with the discrete cosine transform ($\tilde{\mathbf{u}} = \mathbf{V}^\top \mathbf{u} = \mathrm{DCT}(\mathbf{u})$). As the observed finite-resolution image has a natural cut-off frequency (Nyquist limit due to 'distance' between pixel centers), the projection onto and from the cosine basis can be done with perfect accuracy due to the Nyquist-Shannon sampling theorem. Formally, a finite-dimensional Laplace operator can be written out as the eigendecomposition $\Delta \triangleq \mathbf{V}\mathbf{\Lambda}\mathbf{V}^\top$, where $\mathbf{\Lambda}$ is a diagonal matrix containing the negative squared frequencies $-\pi^2(\frac{n^2}{W^2} + \frac{m^2}{H^2})$ and $\mathbf{V}^\top$ is the discrete cosine transform projection matrix. The evolution equation, and our numerical solution to the heat equation, can thus be described by the finite-dimensional evolution model (in image $\Leftrightarrow$ Fourier space):

$$\mathbf{u}(t) = \mathbf{F}(t)\,\mathbf{u}(0) = \exp(\mathbf{V}\mathbf{\Lambda}\mathbf{V}^\top t)\,\mathbf{u}(0) = \mathbf{V}\exp(\mathbf{\Lambda}t)\mathbf{V}^\top\mathbf{u}(0) \;\Leftrightarrow\; \tilde{\mathbf{u}}(t) = \exp(\mathbf{\Lambda}t)\tilde{\mathbf{u}}(0), \tag{21}$$

where $\mathbf{F}(t) \in \mathbb{R}^{N \times N}$ is the transition model and $\mathbf{u}(0)$ the initial state. $\mathbf{F}(t)$ is not expanded out in practice, but instead we use the DCT and inverse DCT, which are $O(N \log N)$ operations. As $\mathbf{\Lambda}$ is diagonal, the Fourier-space model is fast to evaluate. The algorithm in practice is summarized as a Python snippet in Alg. 3.

**Algorithm 3** Python code for calculating the forward process

```python
import numpy as np
from scipy.fftpack import dct, idct
def heat_eq_forward(u, t):
  # Assuming the image u is an (KxK) numpy array
  K = u.shape[-1]
  freqs = np.pi*np.linspace(0,K-1,K)/K
  frequencies_squared = freqs[:,None]**2 + freqs[None,:]**2
  u_proj = dct(u, axis=0, norm='ortho')
  u_proj = dct(u_proj, axis=1, norm='ortho')
  u_proj = np.exp( - frequencies_squared * t) * u_proj
  u_reconstucted = idct(u_proj, axis=0, norm='ortho')
  u_reconstucted = idct(u_reconstucted, axis=1, norm='ortho')
  return u_reconstucted
```

## A.2  DERIVATION OF THE VARIATIONAL LOWER BOUND

This section contains a derivation for the variational bound in more detail than what was presented in the main text. It is mainly intended for readers not already familiar with the diffusion model mathematics. Recall the definitions for the generative Markov chain and the inference distribution:

Reverse process /
Generative model
$$p_\theta(\mathbf{u}_{0:K}) = p(\mathbf{u}_K) \prod_{k=1}^{K} p_\theta(\mathbf{u}_{k-1} \mid \mathbf{u}_k) = p(\mathbf{u}_K) \prod_{k=1}^{K} \mathcal{N}(\mathbf{u}_{k-1} \mid \boldsymbol{\mu}_\theta(\mathbf{u}_k, k), \delta^2 \mathbf{I})$$

Forward process /
Inference distribution
$$q(\mathbf{u}_{1:K} \mid \mathbf{u}_0) = \prod_{k=1}^{K} q(\mathbf{u}_k \mid \mathbf{u}_0) = \prod_{k=1}^{K} \mathcal{N}(\mathbf{u}_k \mid \mathbf{F}(t_k)\mathbf{u}_0, \sigma^2 \mathbf{I}).$$

Taking the negative of the evidence lower bound, we get

$$-\log p_\theta(\mathbf{u}_0) \leq \mathbb{E}_{q(\mathbf{u}_{1:K} \mid \mathbf{u}_0)} \left[ -\log \frac{p_\theta(\mathbf{u}_{0:K})}{q(\mathbf{u}_{1:K} \mid \mathbf{u}_0)} \right] \tag{22}$$

$$= \mathbb{E}_{q(\mathbf{u}_{1:K} \mid \mathbf{u}_0)} \left[ -\log \frac{p(\mathbf{u}_K) \prod_{k=1}^K p_\theta(\mathbf{u}_{k-1} \mid \mathbf{u}_k)}{\prod_{k=1}^K q(\mathbf{u}_k \mid \mathbf{u}_0)} \right] \tag{23}$$

$$= \mathbb{E}_{q(\mathbf{u}_{1:K} \mid \mathbf{u}_0)} \left[ -\log \frac{p_\theta(\mathbf{u}_K)}{q(\mathbf{u}_K \mid \mathbf{u}_0)} - \log \prod_{k=2}^K \frac{p_\theta(\mathbf{u}_{k-1} \mid \mathbf{u}_k)}{q(\mathbf{u}_{k-1} \mid \mathbf{u}_0)} - \log p_\theta(\mathbf{u}_0 \mid \mathbf{u}_1) \right] \tag{24}$$

$$= \mathbb{E}_{q(\mathbf{u}_{1:K} \mid \mathbf{u}_0)} \left[ -\log \frac{p_\theta(\mathbf{u}_K)}{q(\mathbf{u}_K \mid \mathbf{u}_0)} - \sum_{k=2}^K \log \frac{p_\theta(\mathbf{u}_{k-1} \mid \mathbf{u}_k)}{q(\mathbf{u}_{k-1} \mid \mathbf{u}_0)} - \log p_\theta(\mathbf{u}_0 \mid \mathbf{u}_1) \right] \tag{25}$$

$$= \mathbb{E}_{q(\mathbf{u}_{1:K-1} \mid \mathbf{u}_0)} \int q(\mathbf{u}_K \mid \mathbf{u}_0) \log \frac{q(\mathbf{u}_K \mid \mathbf{u}_0)}{p_\theta(\mathbf{u}_K)} d\mathbf{u}_K$$
$$+ \sum_{k=2}^K \mathbb{E}_{q(\mathbf{u}_{\{1:K\} \setminus \{k-1\}} \mid \mathbf{u}_0)} \int q(\mathbf{u}_{k-1} \mid \mathbf{u}_0) \log \frac{q(\mathbf{u}_{k-1} \mid \mathbf{u}_0)}{p_\theta(\mathbf{u}_{k-1} \mid \mathbf{u}_k)} d\mathbf{u}_{k-1}$$
$$- \mathbb{E}_{q(\mathbf{u}_{1:K} \mid \mathbf{u}_0)} \log p_\theta(\mathbf{u}_0 \mid \mathbf{u}_1) \tag{26}$$

$$= \mathbb{E}_{q(\mathbf{u}_{1:K-1} \mid \mathbf{u}_0)} \mathrm{D_{KL}}[q(\mathbf{u}_K \mid \mathbf{u}_0) \,\|\, p(\mathbf{u}_K)]$$
$$+ \sum_{k=2}^K \mathbb{E}_{q(\mathbf{u}_{\{1:K\} \setminus \{k-1\}} \mid \mathbf{u}_0)} \mathrm{D_{KL}}[q(\mathbf{u}_{k-1} \mid \mathbf{u}_0) \,\|\, p(\mathbf{u}_{k-1} \mid \mathbf{u_k})]$$
$$- \mathbb{E}_{q(\mathbf{u}_{1:K} \mid \mathbf{u}_0)} \log p_\theta(\mathbf{u}_0 \mid \mathbf{u}_1). \tag{27}$$

In the KL terms $\mathrm{D_{KL}}[q(\mathbf{u}_{k-1} \mid \mathbf{u}_0) \,\|\, p(\mathbf{u}_{k-1} \mid \mathbf{u_k})]$, the time steps $k-1$ have already been integrated over and the term is dependent only on $\mathbf{u}_k$ when $\mathbf{u}_0$ is constant. The term $\mathrm{D_{KL}}[q(\mathbf{u}_K \mid \mathbf{u}_0) \,\|\, p(\mathbf{u}_K)]$ is not dependent on any of the variables $k$ and $\log p_\theta(\mathbf{u}_0 \mid \mathbf{u}_1)$ only on $\mathbf{u}_1$. We can proceed in two ways: Either *(i)* add dummy integrals over the already marginalized over dimensions to get Eq. (7) in the main text, or *(ii)* marginalize out all redundant expectation values to explicitly get the final loss function. We first look at *(i)*:

$$\mathbb{E}_{q(\mathbf{u}_{1:K-1} \mid \mathbf{u}_0)} \int q(\mathbf{u}'_K \mid \mathbf{u}_0) d\mathbf{u}'_K \underbrace{\mathrm{D_{KL}}[q(\mathbf{u}_K \mid \mathbf{u}_0) \,\|\, p(\mathbf{u}_K)]}_{\text{Not dependent on } \mathbf{u}'_K}$$
$$+ \sum_{k=2}^K \mathbb{E}_{q(\mathbf{u}_{\{1:K\} \setminus \{k-1\}} \mid \mathbf{u}_0)} \int q(\mathbf{u}'_{k-1} \mid \mathbf{u}_0) d\mathbf{u}'_{k-1} \underbrace{\mathrm{D_{KL}}[q(\mathbf{u}_{k-1} \mid \mathbf{u}_0) \,|\, p(\mathbf{u}_{k-1} \mid \mathbf{u_k})]}_{\text{Not dependent on } \mathbf{u}_{k-1}}$$
$$- \mathbb{E}_{q(\mathbf{u}_{1:K} \mid \mathbf{u}_0)} \log p_\theta(\mathbf{u}_0 \mid \mathbf{u}_1) \tag{28}$$

$$= \mathbb{E}_{q(\mathbf{u}_{1:K} \mid \mathbf{u}_0)} \left[ \underbrace{\mathrm{D_{KL}}[q(\mathbf{u}_K \mid \mathbf{u}_0) \,\|\, p(\mathbf{u}_K)]}_{L_K} + \sum_{k=2}^K \underbrace{\mathrm{D_{KL}}[q(\mathbf{u}_{k-1} \mid \mathbf{u}_0) \,\|\, p_\theta(\mathbf{u}_{k-1} \mid \mathbf{u}_k)]}_{L_{k-1}} \right.$$
$$\left. \underbrace{- \log p_\theta(\mathbf{u}_0 \mid \mathbf{u}_1)}_{L_0} \right], \tag{29}$$

which is the formula presented in the main text. On path *(ii)*, marginalizing redundant integrals to get the loss function, we get

$$\underbrace{\mathrm{D_{KL}}[q(\mathbf{u}_K \mid \mathbf{u}_0) \,|\, p(\mathbf{u}_K)]}_{L_K} + \sum_{k=2}^K \mathbb{E}_{q(\mathbf{u}_k \mid \mathbf{u}_0)} \underbrace{\mathrm{D_{KL}}[q(\mathbf{u}_{k-1} \mid \mathbf{u}_0) \,\|\, p_\theta(\mathbf{u}_{k-1} \mid \mathbf{u}_k)]}_{L_{k-1}} - \mathbb{E}_{q(\mathbf{u}_1 \mid \mathbf{u}_0)} \underbrace{\log p_\theta(\mathbf{u}_0 \mid \mathbf{u}_1)}_{L_0}. \tag{30}$$

The first term is constant. As all distributions are defined as Gaussians with diagonal covariance matrices, we get

$$\mathbb{E}_{q(\mathbf{u}_k \mid \mathbf{u}_0)}[L_{k-1}] = \frac{1}{2}\left(\frac{\sigma^2}{\delta^2}N - N + \frac{1}{\delta^2}\mathbb{E}_{q(\mathbf{u}_k \mid \mathbf{u}_0)}\left[\|\boldsymbol{\mu}_\theta(\mathbf{u}_k, k) - F(t_{k-1})\mathbf{u}_0\|_2^2\right] + 2N\log\frac{\delta}{\sigma}\right),$$
(31)

$$\mathbb{E}_{q(\mathbf{u}_1 \mid \mathbf{u}_0)}[L_0] = \mathbb{E}_{q(\mathbf{u}_1 \mid \mathbf{u}_0)}[-\log p_\theta(\mathbf{u}_0 \mid \mathbf{u}_1)]$$
(32)

$$= \frac{1}{2\delta^2}\mathbb{E}_{q(\mathbf{u}_1 \mid \mathbf{u}_0)}\left[\|\boldsymbol{\mu}_\theta(\mathbf{u}_1, 1) - \mathbf{u}_0\|_2^2\right] + N\log(\delta\sqrt{2\pi}).$$
(33)

Taking a Monte Carlo estimate of the expectations by sampling once from $\mathbf{u}_k$ or $\mathbf{u}_1$ and passing it through the neural network $\boldsymbol{\mu}_\theta$, we arrive at our final loss function.

### A.3   Variational Upper Bound on $L_K$

In the main text, we noted that $L_K = \mathrm{D}_{\mathrm{KL}}[q(\mathbf{u}_K|\mathbf{u}_0)|p(\mathbf{u}_K)]$ is not trivial to evaluate if we define $p(\mathbf{u}_K)$ to be a kernel density estimator over the training set, since evaluation of $p(\mathbf{u}_K)$ is heavy due to the large amount of components. For each spatial location in the integral, we would need to re-evaluate its distance to all blurred training data points, and this is highly inefficient. We can, however, provide a further variational upper bound, similarly to (Hershey & Olsen, 2007):

$$\mathrm{D}_{\mathrm{KL}}[q(\mathbf{u}_K \mid \mathbf{u}_0) \,\|\, p(\mathbf{u}_K)] = \mathbb{E}_{q(\mathbf{u}_K \mid \mathbf{u}_0)}[\log q(\mathbf{u}_K \mid \mathbf{u}_0) - \log p(\mathbf{u}_K)]$$
(34)

$$= \mathbb{E}_{q(\mathbf{u}_K \mid \mathbf{u}_0)}[\log q(\mathbf{u}_K \mid \mathbf{u}_0) - \log\frac{1}{N_T}\sum_{i=1}^{N_T}\mathcal{N}(\mathbf{u}_K \mid \mathbf{u}_K^i, \sigma^2)],$$
(35)

where $N_T$ is the training set size and $\mathbf{u}_K^i$ is an example from the training set. The first term is simply the negative entropy of a Gaussian. For the second term, introduce variational parameters $\phi_i > 0$ such that $\sum_i^N \phi_i = 1$:

$$-\mathbb{E}_{q(\mathbf{u}_K \mid \mathbf{u}_0)}[\log\frac{1}{N}\sum_{i=1}^{N_T}\mathcal{N}(\mathbf{u}_K \mid \mathbf{u}_K^i, \delta^2)] = -\mathbb{E}_{q(\mathbf{u}_K \mid \mathbf{u}_0)}\left[\log\frac{1}{N}\sum_{i=1}^{N_T}\phi_i\frac{\mathcal{N}(\mathbf{u}_K \mid \mathbf{u}_K^i, \delta^2)}{\phi_i}\right]$$
(36)

$$\leq -\mathbb{E}_{q(\mathbf{u}_K \mid \mathbf{u}_0)}\sum_i \phi_i\log\left[\frac{\mathcal{N}(\mathbf{u}_K \mid \mathbf{u}_K^i, \delta^2)}{N_T\phi_i}\right]$$
(37)

$$= -\sum_i \phi_i\left[\mathbb{E}_{q(\mathbf{u}_K \mid \mathbf{u}_0)}[\log\mathcal{N}(\mathbf{u}_K \mid \mathbf{u}_K^i, \delta^2)] - \log N_T\phi_i\right]$$
(38)

$$= -\sum_i \phi_i\left[-H\left(q(\mathbf{u}_K \mid \mathbf{u}_0), \mathcal{N}(\mathbf{u}_K \mid \mathbf{u}_K^i, \delta^2)\right) - \log N_T - \log\phi_i\right].$$
(39)

Note that the cross-entropy between Gaussians has an analytical formula that can be precomputed for all training set - test set sample pairs. Minimizing the upper bound w.r.t. $\phi_i$, we get

$$\phi_i = \frac{e^{-\mathrm{D}_{\mathrm{KL}}(q(\mathbf{u}_K \mid \mathbf{u}_0) \,\|\, \mathcal{N}(\mathbf{u}_K \mid \mathbf{u}_K^i, \delta^2))}}{\sum_j e^{-\mathrm{D}_{\mathrm{KL}}(q(\mathbf{u}_K \mid \mathbf{u}_0) \,\|\, \mathcal{N}(\mathbf{u}_K \mid \mathbf{u}_K^j, \delta^2))}}.$$
(40)

Now, we just need to evaluate the KL divergences between data points in the training set and the test set, and then we can use those to calculate the upper bound efficiently, without need to integrate over and evaluate $p(\mathbf{u}_K)$ multiple times in a very high-dimensional space. Calculation of the KL divergences amounts to calculating the distances of blurry data points in the training and test sets.

While the calculations are straightforward to implement in practice, one needs to be a bit careful when doing the computations numerically. In particular, we can use the log-sum-exp trick to estimate $\log\phi_i$ in a numerically stable way, and $\phi_i$ is obtained from the $\log\phi_i$ values.

### A.4   Limit of $K \to \infty$ and Convergence to Denoising Score Matching and Langevin Dynamics

In this section, we show in detail the result that in the limit $K \to \infty$, the loss function $L_k$ converges to the denoising score matching loss function and the sampling process becomes equivalent to Langevin

dynamics sampling. Let's start with our loss for the k:th level:

$$D_{KL}[q(\mathbf{u}_k \mid \mathbf{u}_0) \| p_\theta(\mathbf{u}_{k-1} \mid \mathbf{u}_k)] \sim \mathbb{E}_{q(\mathbf{u}_k \mid \mathbf{u}_0)}\|\boldsymbol{\mu}_\theta(\mathbf{u}_k, k) - \mathbf{F}(t_{k-1})\,\mathbf{u}_0\|_2^2 \tag{41}$$

$$= \mathbb{E}_{q(\mathbf{u}_k \mid \mathbf{u}_0)}\|f_\theta(\mathbf{u}_k, k) - (\mathbf{F}(t_{k-1})\,\mathbf{u}_0 - \mathbf{u}_k)\|_2^2. \tag{42}$$

where $\sim$ denotes that overall multiplicative and additive constants have been removed. Now note that if we let $K \to \infty$ and redefine $f_\theta(\mathbf{u}_k, k) = f'_\theta(\mathbf{u}_k, k)\sigma^2$, then $\mathbf{F}(t_{k-1}) \to \mathbf{F}(t_k)$ and Eq. (42) will approach

$$\mathbb{E}_{q(\mathbf{u}_k \mid \mathbf{u}_0)}\|f_\theta(\mathbf{u}_k, k) - (\mathbf{F}(t_{k-1})\,\mathbf{u}_0 - \mathbf{u}_k)\|_2^2 \tag{43}$$

$$\to \mathbb{E}_{q(\mathbf{u}_k \mid \mathbf{u}_0)}\|f_\theta(\mathbf{u}_k, k) - (\mathbf{F}(t_k)\,\mathbf{u}_0 - \mathbf{u}_k)\|_2^2 \tag{44}$$

$$= \mathbb{E}_{q(\mathbf{u}_k \mid \mathbf{u}_0)}\|f'_\theta(\mathbf{u}_k, k)\sigma^2 - (\mathbf{F}(t_k)\,\mathbf{u}_0 - \mathbf{u}_k)\|_2^2 \tag{45}$$

$$= \sigma^4 \mathbb{E}_{q(\mathbf{u}_k \mid \mathbf{u}_0)}\|f'_\theta(\mathbf{u}_k, k) - \frac{\mathbf{F}(t_k)\,\mathbf{u}_0 - \mathbf{u}_k}{\sigma^2}\|_2^2, \tag{46}$$

which is equivalent to the denoising score matching loss with a Gaussian kernel of variance $\sigma^2$ on a data set blurred out with the matrix $\mathbf{F}(t_k)$, up to an arbitrary scaling. Now, optimizing this results in an estimate of the score $\nabla \log p_k(\mathbf{u})$, where $p_k(\mathbf{u})$ is the kernel density estimate of the data at level $k$, so that $f_\theta(\mathbf{u}, k) \approx \nabla \log p_k(\mathbf{u})\sigma^2$. We can then construct the Langevin SDE that has a stationary distribution $p_k(\mathbf{u})$:

$$d\mathbf{u} = \nabla \log p_k(\mathbf{u})dt + \sqrt{2}dW, \tag{47}$$

where $W(t)$ is a standard Brownian motion. Note that here we overload notation slightly: $t$ is the time dimension of the SDE, not the time dimension of the heat equation. An Euler–Maryama discretized step along the SDE yields

$$\Delta\mathbf{u} = \nabla \log p_k(\mathbf{u})\Delta t + \sqrt{2\Delta t}\mathbf{z}, \quad \text{where } \mathbf{z} \sim \mathcal{N}(\mathbf{0}, \mathbf{I}) \tag{48}$$

$$\approx f'_\theta(\mathbf{u})\Delta t + \sqrt{2\Delta t}\mathbf{z} \tag{49}$$

$$= \frac{f_\theta(\mathbf{u})}{\sigma^2}\Delta t + \sqrt{2\Delta t}\mathbf{z}. \tag{50}$$

Now if we choose the step size to be $\Delta t = \sigma^2$, we get the following update

$$\Delta\mathbf{u} = f_\theta(\mathbf{u}) + \sqrt{2}\sigma\mathbf{z}, \quad \text{where } \mathbf{z} \sim \mathcal{N}(\mathbf{0}, \mathbf{I}), \tag{51}$$

which is exactly equivalent to our sampling update step with $\delta = \sqrt{2}\sigma \approx 1.41\sigma$.

Note that this result is not meant to be a derivation for the model itself, but instead to provide a preliminary statistical analysis of our method in an asymptotic limit, helping reasoning with the model and showing theoretical connections to other ideas in the diffusion model literature. In particular, we do not in practice simply perform this type of Langevin dynamics with a distribution $p_k(\mathbf{u})$ and slowly shift it towards $p_0(\mathbf{u})$, but instead take directed steps backward in the heat equation. Thus, the $\delta \approx \sqrt{2}\sigma$ also simply provides an intuitive rough scale for $\delta$, and is not necessarily the value we want to use in practice.

## A.5 ANALYSIS FOR THE POWER SPECTRAL DENSITY IN THE DIFFUSION FORWARD PROCESS

Here we analyse explicitly the PSD behaviour in the diffusion forward process. In particular, the expected value of the PSD of a noised image equals to the PSD of the original image added with the noise variance. We define the PSD for individual frequency components as $\text{PSD}(\mathbf{u})_i = |\mathbf{v}_i^\top \mathbf{u}|^2$, where $\mathbf{v}_i$ is the projection vector to the $i^{\text{th}}$ frequency in the DCT/DFT basis. These individual components compose the PSD of the image, $\text{PSD}(\mathbf{u})$.

Without loss of generality, we consider a diffusion process where we do not scale the original image $\mathbf{u}$ and simply add noise $\varepsilon$ with covariance $\gamma^2\mathbf{I}$. The expected PSD of the original image plus noise is:

$$\mathbb{E}_\varepsilon[\text{PSD}(\mathbf{u}+\varepsilon)_i] = \mathbb{E}_\varepsilon[|\mathbf{v}_i^\top(\mathbf{u}+\varepsilon)|^2] \tag{52}$$

$$= \mathbb{E}_\varepsilon[(\mathbf{v}_i^\top\mathbf{u})^2 + 2(\mathbf{v}_i^\top\mathbf{u})(\mathbf{v}_i^\top\varepsilon) + (\mathbf{v}_i^\top\varepsilon)^2] \tag{53}$$

$$= |\mathbf{v}_i^\top\mathbf{u}|^2 + 2(\mathbf{v}_i^\top\mathbf{u})(\mathbf{v}_i^\top\underbrace{\mathbb{E}_\varepsilon[\varepsilon]}_{=0}) + \mathbb{E}_\varepsilon[|\mathbf{v}_i^\top\varepsilon|^2] \tag{54}$$

$$= \text{PSD}(\mathbf{u})_i + \mathbf{v}_i^\top\mathbb{E}_\varepsilon[\varepsilon\varepsilon^\top]\mathbf{v}_i \tag{55}$$

$$= \text{PSD}(\mathbf{u})_i + \mathbf{v}_i^\top\gamma^2 I\mathbf{v}_i \tag{56}$$

$$= \text{PSD}(\mathbf{u})_i + \gamma^2 \tag{57}$$

meaning that the PSD of the image and the variance of the noise add together. This gives a formal explanation for the idea that isotropic noise effectively drowns out the frequency components in the data with a lower PSD than the variance of the noise. Note that in the 1D plot Fig. 4, the equal frequency components have been averaged out, and this is visualised in more detail in App. B.6. The result does not change in that case aside from replacing $\text{PSD}(\mathbf{u})_i$ with the mean of multiple PSD values that correspond to equal frequencies.

As a further curiosity, we also point out that a single PSD component is distributed as the sum of a normally distributed random variable and a chi-squared variable:

$$|\mathbf{v}_i^\top(\varepsilon+\mathbf{u})|^2 = (\mathbf{v}_i^\top\mathbf{u})^2 + 2(\mathbf{v}_i^\top\mathbf{u})\underbrace{(\mathbf{v}_i^\top\varepsilon)}_{\sim\mathcal{N}(0,\gamma^2)} + \underbrace{(\mathbf{v}_i^\top\varepsilon)^2}_{\sim\chi_1^2\gamma^2}. \tag{58}$$

Here we use the fact that a 1D orthonormal projection of an isotropic zero-mean Gaussian random variable is a simple 1D Gaussian.

# B EXPERIMENT DETAILS

## B.1 NEURAL NETWORK ARCHITECTURE AND HYPERPARAMETERS

Similarly to many recent works on diffusion models (Ho et al., 2020; Song et al., 2021d; Nichol & Dhariwal, 2021; Dhariwal & Nichol, 2021), we use a U-Net architecture for estimating the transitions, and in particular parametrize $\boldsymbol{\mu}_\theta(\mathbf{u}_k, k) = \mathbf{u}_k + f_\theta(\mathbf{u}_k, k)$, where $f_\theta(\mathbf{u}_k, k)$ corresponds to the U-Net, as explained in the main text. Following (Dhariwal & Nichol, 2021), we use self-attention layers at multiple low-resolution feature maps. Otherwise, the architecture follows the one in (Ho et al., 2020), and we do not, *e.g.*, use multiple attention heads or adaptive group normalization (Dhariwal & Nichol, 2021). We use two residual blocks per resolution layer in other models than the MNIST and CIFAR-10 models, where we use 4. We use 128 base channels, GroupNorm layers in the residual blocks and a dropout rate of 0.1, applied in each residual block. The downsampling steps in the U-Net were done with average pooling and upsampling steps with nearest-neighbour interpolation. The time step information is included using a sinusoidal embedding that is added to the feature maps in each residual block.

Table 1 lists the values of other hyperparameters that we used on different data sets. They are mostly based on the recent work with diffusion models (Ho et al., 2020; Nichol & Dhariwal, 2021; Dhariwal & Nichol, 2021), and have not been optimized over. For the lower resolution data sets, we used a higher learning rate of $10^{-4}$ or $2\cdot10^{-4}$, and a lower learning rate of $2\cdot10^{-5}$ for the higher resolutions for added stability, although we did not sweep over these values. An EMA rate of 0.9999 was also used instead of 0.999 for $256\times256$ images, instead of the 0.999 that was used for other data sets. We used the Adam optimizer with default hyperparameters $\beta_1 = 0.9, \beta_2 = 0.999$ and $\varepsilon = 10^{-8}$. We also use gradient norm clipping with rate 1.0 and learning rate warm up by linearly interpolating it from 0 to the desired learning during the first 5000 steps. We do not add noise on the final step of the generative process, as that cannot increase the output quality. Other details are included in the code release.

We used random horizontal flips on AFHQ, and no data augmentation on the other data sets.

Table 1: Neural network hyperparameters.

| Data | Resolution | Layer multipliers | Base channels | Learning rate | Self-attention resolutions | EMA | Batch size | # Res-blocks |
|---|---|---|---|---|---|---|---|---|
| MNIST | 28×28 | (1,2,2) | 128 | 1e-4 | 7×7 | 0.999 | 128 | 4 |
| CIFAR-10 | 32×32 | (1, 2, 2, 2) | 128 | 2e-4 | 8×8, 4×4 | 0.999 | 128 | 4 |
| FFHQ | 256×256 | (1, 2, 3, 4, 5) | 128 | 2e-5 | 64×64, 32×32, 16×16 | 0.9999 | 32 | 2 |
| FFHQ | 128×128 | (1, 2, 3, 4, 5) | 128 | 2e-5 | 32×32, 16×16, 8×8 | 0.999 | 32 | 2 |
| AFHQ | 256×256 | (1, 2, 3, 4, 5) | 128 | 2e-5 | 64×64, 32×32, 16×16 | 0.9999 | 32 | 2 |
| AFHQ | 64×64 | (1, 2, 3, 4) | 128 | 1e-4 | 16×16, 8×8 | 0.999 | 128 | 2 |
| LSUN CHURCHES | 128×128 | (1, 2, 3, 4, 5) | 128 | 2e-5 | 32×32, 16×16, 8×8 | 0.999 | 32 | 2 |

## B.2 TRAINING TIME AND COMPUTATIONAL RESOURCES

We use NVIDIA A100 GPUs for the experiments, with two GPUs for 256×256 resolution models and one GPU for all others. We use the Pytorch automatic mixed precision functionality for a higher per-GPU batch size. On CIFAR-10, we trained for 400 000 iterations, taking 7 hours per 100,000 steps. On the 256×256 images, 100,000 iterations takes about 40 hours, and we used 800,000 iterations on FFHQ and 400,000 iterations on AFHQ. On LSUN-Churches, 100,000 training steps takes 11 hours, and training was continued for one million iterations. On the smaller 64×64 AFHQ data set, we trained for 100,000 iterations, taking a total of 14 hours.

## B.3 FID SCORE CALCULATION

To calculate FID-scores, we used clean-fid (Parmar et al., 2022), where the generated images and reference images are scaled to 299×299 resolution with bicubic interpolation before passing them to the InceptionV3 network. We used 50,000 samples to calculate the scores for all other data sets than the 256×256 sets, where we used 10,000 samples. Using a lower amount of samples results in the values being slightly overestimated, which is not too much of an issue since the FID scores are not very close to state-of-the-art values. We used the training set to calculate the reference statistics on LSUN-CHURCHES and CIFAR-10, and the entire data sets for FFHQ and AFHQ.

## B.4 HYPERPARAMETERS RELATED TO THE NEW GENERATIVE PROCESS

The different hyperparameters related to the new generative process are listed in Table 2. Early on during experimentation, we noticed that $\sigma = 0.01$ for the training noise seems to work well, and use that for all experiments unless mentioned otherwise. We have not done an extensive study on the optimal value. The number of iteration steps $K$ was set to 200 for most experiments, except for the LSUN CHURCHES data set, where we noted that $K = 400$ increased sample quality slightly. Otherwise, in contrast to findings on diffusion models (Ho et al., 2020), we found in early experimentation that increasing the number of steps well above 200 did not seem to result in trivial improvements in sample quality. On MNIST, we used 100 steps. Although a $\delta$ of $0.0125 = 1.25 \times \sigma$ seemed to work well as a default value on all data sets, we tuned it to 0.01325 on CIFAR-10 by sweeping over the FIDs obtained with different values (results visualized in App. C.3). On AFHQ 256×256, we set it to 0.01275 because visual inspection showed slightly improved results. The maximal effective blurring length-scale, $\sigma_{B,\max}$, was set to half the size of the images in most experiments, although on MNIST and CIFAR-10 we used slightly higher values. We noticed during early experimentation that moving $\sigma_{B,\max}$ from the entire image width to half the width resulted in better image quality, although the information content present in the half-blurred image is intuitively not much different from the information content in the fully averaged out image. We also study how $\sigma_{B,\max}$ affects the value of the prior overlap term $D_{\mathrm{KL}}[q(\mathbf{u}_K \,|\, \mathbf{u}_0) \,\|\, p(\mathbf{u}_K)]$ in App. C.4 on CIFAR-10, providing justification to not having $\sigma_{B,\max}$ = entire image width.

## B.5 SCHEDULE ON $t_k$

In all experiments, we used a logarithmic spacing for the time steps $t_k$, where $t_K = \sigma_{B,\max}^2/2$ and $t_1 = \sigma_{B,\min}^2/2 = {}^{0.5^2}/2$, corresponding to sub-pixel-size blurring. Effective averaging sizes $\sigma_{B,\min}$ on other levels were then interpolated with $\sigma_{B,k} = \exp(\log \sigma_{B,\min} \frac{K-k}{K-1} + \log \sigma_{B,\max} \frac{k-1}{K-1})$

Table 2: Hyperparameters related to the generative process.

| Data | Resolution | K | $\sigma_{B,\max}$ | $\sigma$ | $\delta$ |
|------|-----------|---|-----------|-----------|-----------|
| MNIST | $28\times28$ | 100 | 20 | 0.01 | 0.0125 |
| CIFAR-10 | $32\times32$ | 200 | 24 | 0.01 | 0.01325 |
| FFHQ | $256\times256$ | 200 | 128 | 0.01 | 0.0125 |
| FFHQ | $128\times128$ | 200 | 128 | 0.01 | 0.0125 |
| AFHQ | $256\times256$ | 200 | 128 | 0.01 | 0.01275 |
| AFHQ | $64\times64$ | 200 | 32 | 0.01 | 0.0125 |
| LSUN CHURCHES | $128\times128$ | 400 | 64 | 0.01 | 0.0125 |

for an even spacing on a logarithmic axis. We can view the $\sigma_B$ schedule in two ways: First, it corresponds to a constant rate of resolution decrease, in the sense that $\frac{\sigma_{B,k+1}}{\sigma_{B_k}}$ is constant. Second, we can explicitly visualize the rate of effective dimensionality decrease by looking at the frequency components in the discrete cosine transform as we increase $\sigma_B = \sqrt{2t}$. When they pass well below the $\sigma^2 = 0.01^2$ line, the frequency components become indistinguishable from noise in the forward process $q(\mathbf{u}_k \,|\, \mathbf{u}_0) = \mathcal{N}(\mathbf{u}_k \,|\, \mathbf{F}(t_k)\,\mathbf{u}_0, \sigma^2 \mathbf{I})$. This is done in Fig. 9, where we see that with a logarithmic spacing on $\sigma_B$ (and $t_k$), the amount of remaining frequencies decreases at an approximately constant rate, although slows down somewhat towards the end.

## B.6 CALCULATION OF POWER SPECTRAL DENSITIES

We define the power spectral density of a frequency as the squared absolute value of the DCT coefficient for that frequency. So we start by taking the 2D DCT of the image to get the 2D frequency coefficients. Then we square those values to get the corresponding PSDs. To get the 1D plots of PSD with respect to frequency, *e.g.*, as in Fig. 4, we take the average PSDs over equal-frequency contours. This is visualized in Fig. 10 for an example image. We use the orthogonal version of DCT. Aside from DCT, we could also use the discrete cosine transform (DFT).

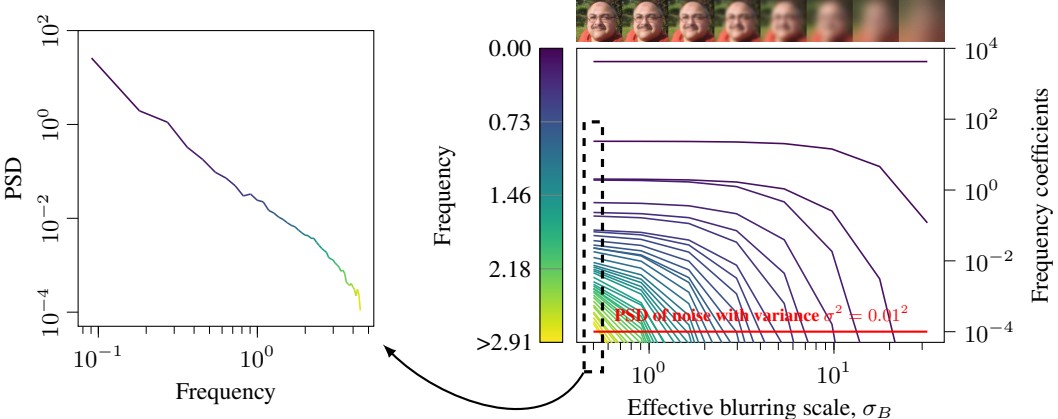

Figure 9: Squared frequency components in the discrete cosine transform of the example image as a function of the effective blurring width $\sigma_B$. As the heat equation progresses, the more frequency components become indistinguishable from the training noise $\sigma$. Each vertical slice on the plot on the right corresponds to a 1D PSD, such as the one on the left. The values are averages over equal-frequency contours in the full 2D DCT.

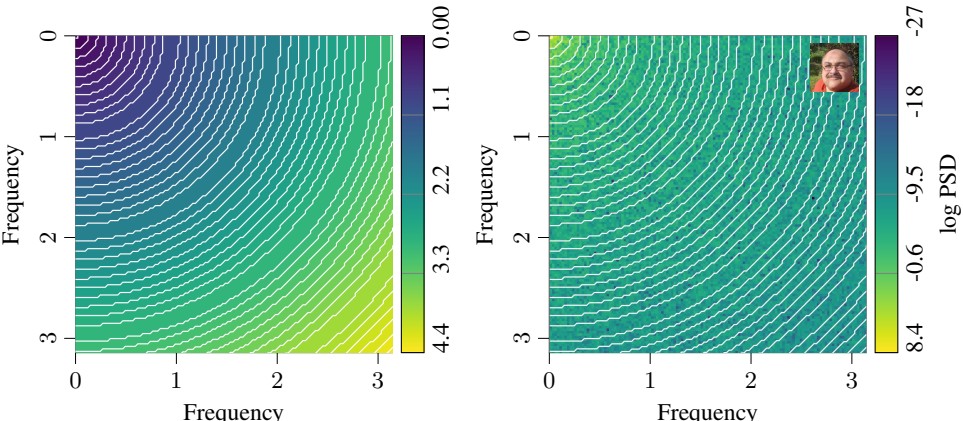

Figure 10: Left: The partitioning of frequencies in the 2D Discrete Cosine Transform. Right: The PSD from the discrete cosine transform of an example image, with the partitions. The one-dimensional power spectral densities, *e.g.*, in Fig. 4, are obtained by averaging the PSDs along the equal-frequency contours.

# C  ADDITIONAL EXPERIMENTS

## C.1  ROBUSTNESS TO $\sigma$ AND $\delta$

In this section, we empirically investigate the relationship between the parameters $\delta$ and $\sigma$ and their robustness to different choices. We ran models on CIFAR-10 with $\sigma \in \{0.005, 0.0075, 0.01, 0.0125, 0.015, 0.02\}$ and calculated FID scores for different $\delta$ values, which are shown in Fig. 11. Note that *(i)* The model is quite robust to the choice of $\sigma$, as long as it is not too close to zero. *(ii)* The parameter of interest here is $\delta/\sigma$ instead of the absolute value of $\delta$, and optimal FID values are obtained approximately at $\delta = 1.3 \times \sigma$ for all choices of $\sigma$, while other nearby values, such as $\delta = 1.25 \times \sigma$ work also. The overall pattern seems to also be that the model becomes somewhat less sensitive to the choice of $\delta$ with increasing $\sigma$. We also give example images from the different models in Fig. 12 with $\delta = 1.3 \times \sigma$, showing that visual differences between the samples are rather small.

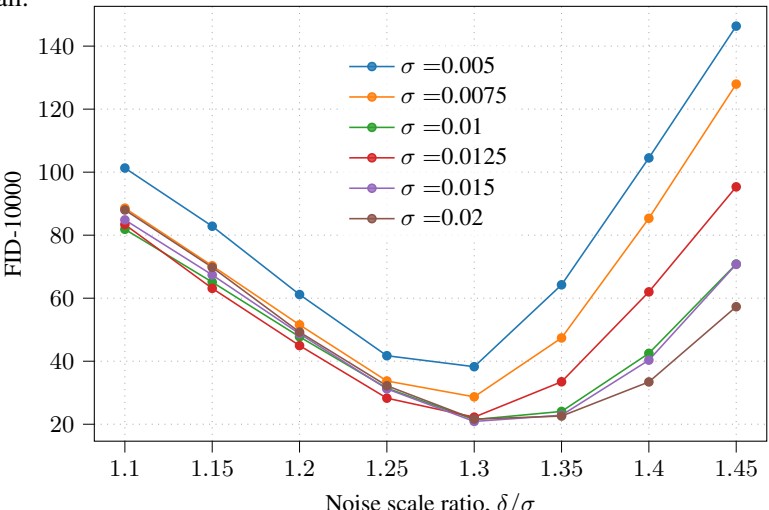

Figure 11: FID-scores for CIFAR-10 models trained with different $\sigma$ values, varying as a function of $\delta$. Note that the optimal values are reached with the same $\delta/\sigma$ ratio, which is about 1.3. The FID scores here are calculated with 10,000 samples, and are slightly overestimated compared to Fig. 14.

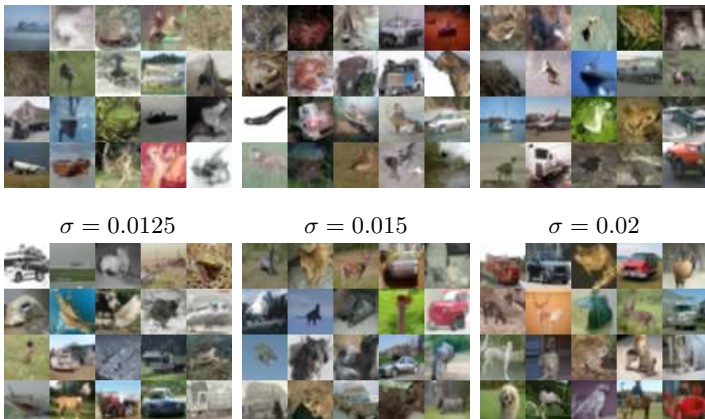

Figure 12: Example images from the different CIFAR-10 models trained with different $\sigma$ and $\delta = 1.3 \times \sigma$. Visual differences are qualitatively not large.

## C.2  DATA EFFICIENCY / FEW-SHOT GENERALIZATION

As said in the main text, the explicit inductive biases introduced in the model allow it to effectively generalize beyond the data set even in very low data regimes. As an extreme example of this, we train

the model and a diffusion model with the first 20 MNIST digits. As shown in Fig. 13, the diffusion model either fails to produce plausible digits or completely overfits the training data. IHDM, however, generalizes to new digit shapes and does not overfit noticeably. Intuitively, the multi-resolution nature of the model provides a very explicit way to generalise on image data: The model can combine learned features on different resolution scales to form new images and meaningful variation with even a few data points.

## C.3 THE OPTIMAL $\delta$ WITH RESPECT TO MARGINAL LOG-LIKELIHOOD VS. FID

We point out that the optimal NLL scores with respect to $\delta$ do not correspond to the optimal $\delta$ for FID values. We plot the negative per-sample ELBO and FID values in Fig. 14, where we see that the lowest NLL is achieved close to $\sigma = 0.01$, whereas the lowest FID score is got somewhere near 0.01325, which is our chosen value for $\delta$, listed in Table 2. To get an intuition to the result, consider the $L_k$ terms in the loss ELBO:

$$\mathbb{E}_q[L_{k-1}] = \mathbb{E}_q\big[D_{\mathrm{KL}}[q(\mathbf{u}_{k-1} \,|\, \mathbf{u}_0) \,\|\, p_\theta(\mathbf{u}_{k-1} | \mathbf{u}_k)]\big] \tag{59}$$

$$= \frac{1}{2}\left(\frac{\sigma^2}{\delta^2}N - N + \frac{1}{\delta^2}\mathbb{E}_{q(\mathbf{u}_k \,|\, \mathbf{u}_0)}\Big[\|\boldsymbol{\mu}_\theta(\mathbf{u}_k, k) - \mathbf{F}(t_{k-1})\,\mathbf{u}_0\|_2^2\Big] + 2N\log\frac{\delta}{\sigma}\right). \tag{60}$$

Without the MSE term in the KL divergences $L_k$, the optimal values for $\delta$ is always $\sigma$, as can be seen by straightforward differentiation of the $\frac{\sigma^2}{\delta^2}N$ and $2N\log\frac{\delta}{\sigma}$ terms. The inclusion of the MSE term nudges the optimal $\delta$ to a higher value, but if the MSE loss is not very high, then it does not get nudged by a lot. Changing the value of $\sigma$ would likely change the picture, but it appears that our model is in a regime where $\frac{\sigma^2}{\delta^2}N$ and $2N\log\frac{\delta}{\sigma}$ dominate the NLL scores. To improve our model as a marginal log-likelihood maximizer, the interplay of these terms could be studied further to get to a hyperparameter regime where the optimal FID values and NLL values are obtained simultaneously.

## C.4 EVALUATION OF THE PRIOR IN TERMS OF OVERLAP WITH THE TEST SET

Overlap between the blurred out train and test sets, measured by $L_K$, can be used as a prerequisite measure of how we can expect the model to generalize beyond the train set, since we use the training data to draw samples from the prior $p(\mathbf{u}_K)$. If $\sigma_{B,\max} \approx 0$, then there is almost no overlap at all and the generative process will amount to just a memorization of the train set. On the other hand, if the data set it fully averaged out at the end of the forward process, then we expect that the using samples from the blurry train set as the prior to have a high overlap with the blurry test set since both distributions are essentially low-dimensional. To showcase the situation with our CIFAR-10 model, we plot the average $L_K$ values on the test set with respect to $\sigma_{B,\max}$. We see that it decreases as the maximal effective length-scale is increases, and does not change much moving from 24 to 32. Thus, there does not seem to be reason to believe that increasing $\sigma_{B,\max}$ would result in much better generalization.

## C.5 THE IMPORTANCE OF NON-ZERO TRAINING NOISE $\sigma$

As noted in the main paper, the training noise $\sigma$ is necessary for the model to be defined in a sensible way. Mechanistically, it also acts as a regularization parameter: Training the neural network to directly solve the exact reverse heat equation, that is, estimate the extremely ill-conditioned inverse $\mathbf{F}(t_K)^{-1}$, does not work. To showcase this, Fig. 16a shows samples from a model trained on MNIST with $\sigma = 0$. The produced images are essentially random patterns. Figure 16b shows the generative process and visualizes neural network input gradients during the process, both for $\delta = 0$, and with a non-zero amount of sampling noise. Starting from the flat prior, the image quickly blows up into a random pattern. Note that the input gradients are similar to the ones seen in Fig. 8 for our model, but the signs are the opposite. Intuitively, the model has learned a generic sharpening filter where the response of the output increases more as the image gets less blurry. This is very unstable, and small errors in the reverse steps are amplified. With a non-zero $\sigma$, the model becomes more robust and is forced to take the training data distribution into account.

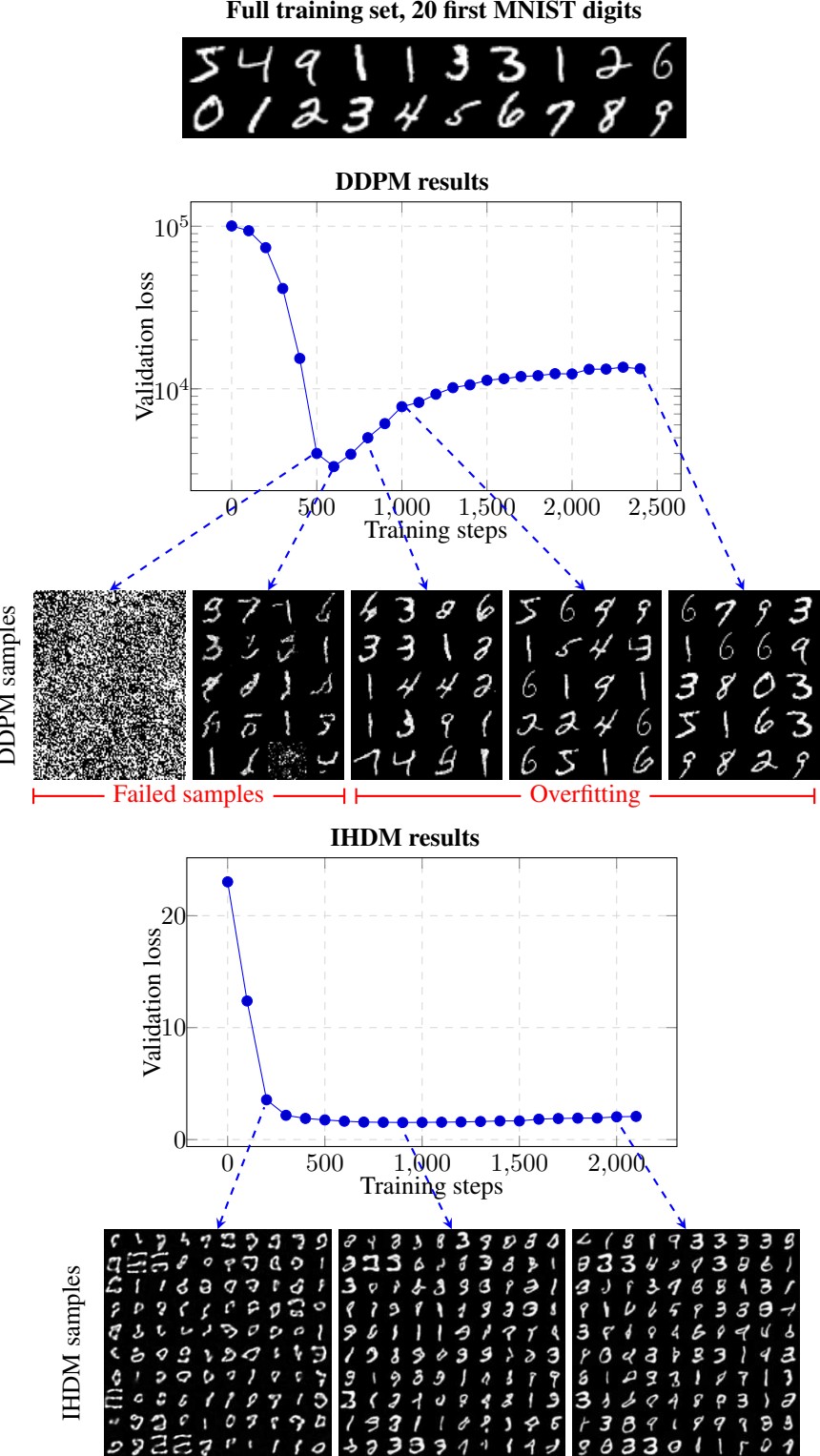

Figure 13: Experiment results on few-shot learning. Top: The entire training set that consists of the 20 first MNIST digits. Middle: The evaluation loss for a denoising diffusion probabilistic model and generated samples at different points during training. The model either fails to produce plausible digits or overfits the training data. Bottom: The evaluation loss and generated samples for IHDM. The model does not overfit to the training data and is able to produce meaningful variation from just the 20 training examples.

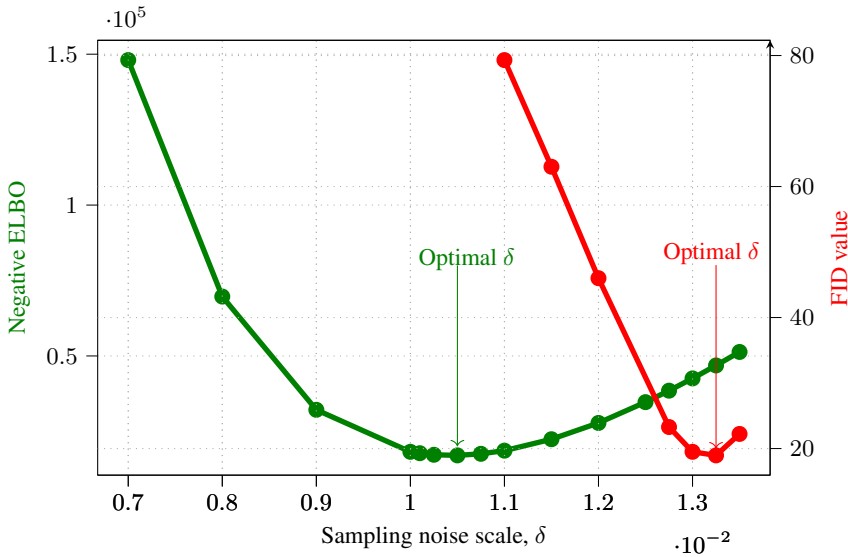

Figure 14: The per-sample negative ELBO and FID values as a function of $\delta$ on our CIFAR-10 model.

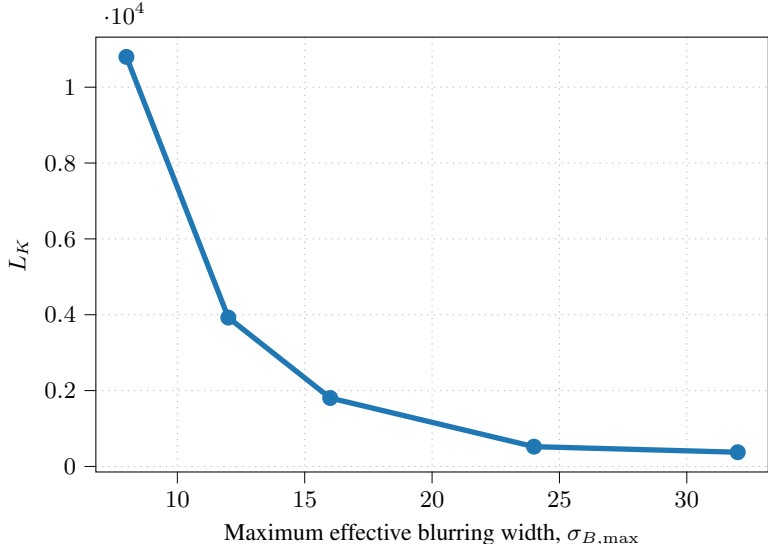

Figure 15: Average values of the $L_K$ term on the CIFAR-10 test set, with respect to the maximum effective blurring width $\sigma_{B,\max}$. $L_K$ effectively measures how large is the overlap between the averaged out test distribution and the averaged out train distribution, which is used as the prior distribution in the model.

## C.6 COMPARISON WITH GAUSSIAN BLUR IMPLEMENTED WITH A CONVOLUTIONAL FILTER

We also experimented with implementing the forward heat dissipation process, or blur, using a convolutional filter with a sampled Gaussian blur. This is a reasonable approach as well, although somewhat computationally slower and does not expose directly the intuitions about frequency decay or the heat equation boundary conditions, as the DCT-based approach does.

To test this, we trained a convolutional filter-based model on CIFAR-10 with otherwise the same parameters as our standard CIFAR-10 one. We use a convolutional kernel size $(2*N-1)\times(2*N-1)$, where $N$ is the width and height of the image in pixels, guaranteeing that all pixels can affect all pixels with large enough blur widths. We then fill the kernel with samples from the Gaussian pdf of different blur standard deviations $\sigma_B$. We use zero-padding at the image edges. The method achieves

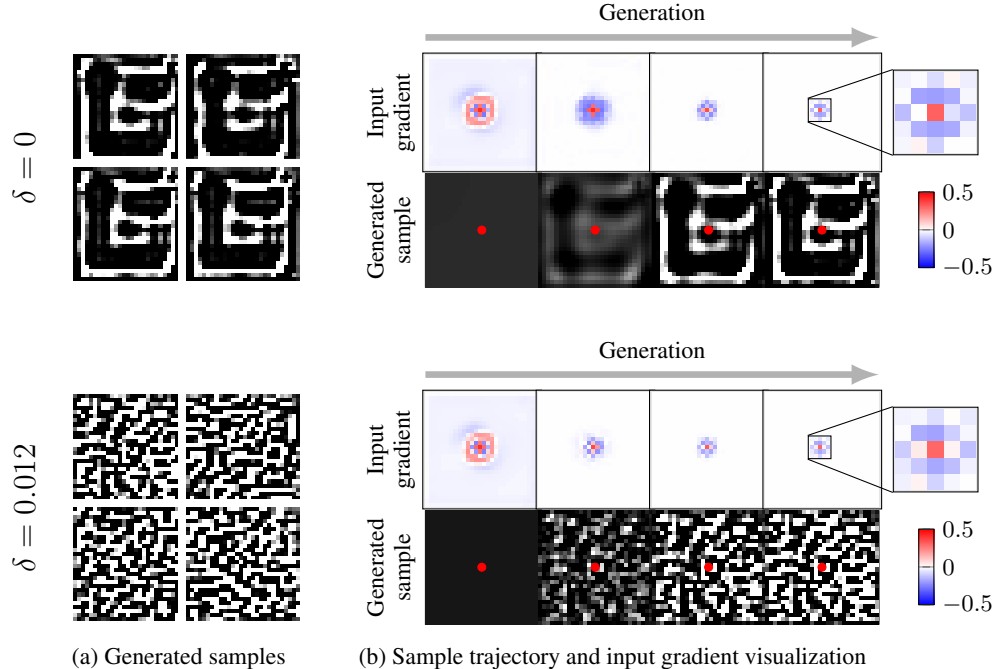

(a) Generated samples   (b) Sample trajectory and input gradient visualization

Figure 16: Failure of the model when $\sigma = 0$. (a) Generated samples from a model trained on MNIST with the training noise parameter $\sigma = 0$. (b) Generative trajectory and neural network input gradients for the pixel highlighted in red. Note that the input gradients resemble the ones seen in Fig. 8 for our model, but the signs are exactly the opposites. The colours are scaled to a symlog scale with a linear cutoff at 0.002.

a FID score of 22.44 as opposed to 18.96 with the DCT-based method, indicating that the DCT-based method may have an edge, although the difference is minor.

A third approach could be to use a finite difference based approximation to the heat equation and use a standard numerical solver to simulate the differential equation forward in time. This should work as well in principle, but the problem is that large blur widths $\sigma_B$ and thus long simulation times $t = \frac{\sigma_B^2}{2}$ translate into lots of sequential computations. This means that the method would be very slow especially for blurring out larger images.

## D  SAMPLE VISUALIZATIONS

In this section, we start by showcasing uncurated samples on the different data sets we trained our models on in App. D.1. We then showcase the finding that interpolating the noise and starting image results in smooth interpolations in the output image in our model in App. D.2. In App. D.3 we illustrate the behaviour of the $\delta$ parameter in more detail, and in particular point out that $\delta = 1.25 \times \sigma$ results in good image quality across data sets and resolutions. In App. D.4 we provide further examples of the result where the overall colour and other features of images can become disentangled in our model. Finally, in App. D.5, we plot example Euclidean nearest neighbours of samples from our model, showcasing that the generated samples are not just approximations of training set images.

In the interpolation results, the prior states $\mathbf{u}_K$ are interpolated linearly. The noises are interpolated with a spherical interpolation $\sin(\phi)\nu_1 + \cos(\pi/2 - \phi)\nu_2$, where $\nu_1$ and $\nu_2$ are the noise vectors and $\phi \in [0, \frac{\pi}{2}]$. The reason is that when we sample two random high-dimensional standard Gaussian noise vectors, they are, with high probability, approximately orthogonal to each other, with approximately equal magnitudes. In a linear interpolation between two orthogonal, equal magnitude vectors, the magnitude of the interpolated vector will decrease half-way. In Fig. 6 we saw that decreasing the magnitude of the sampling noise has a systematic qualitative effect on the results, and really we want the magnitude to remain constant during interpolation. This is achieved with spherical interpolation,

where the vector is moved along the surface of a hypersphere between the two orthogonal vectors $\nu_1$ and $\nu_2$.

## D.1 ADDITIONAL SAMPLES

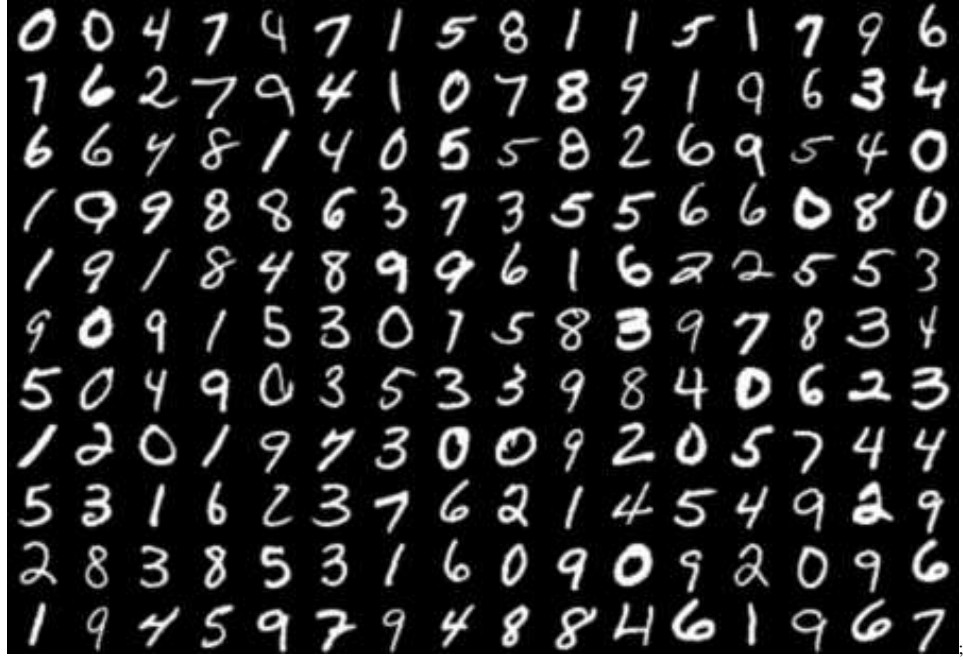

Figure 17: Uncurated samples on MNIST.

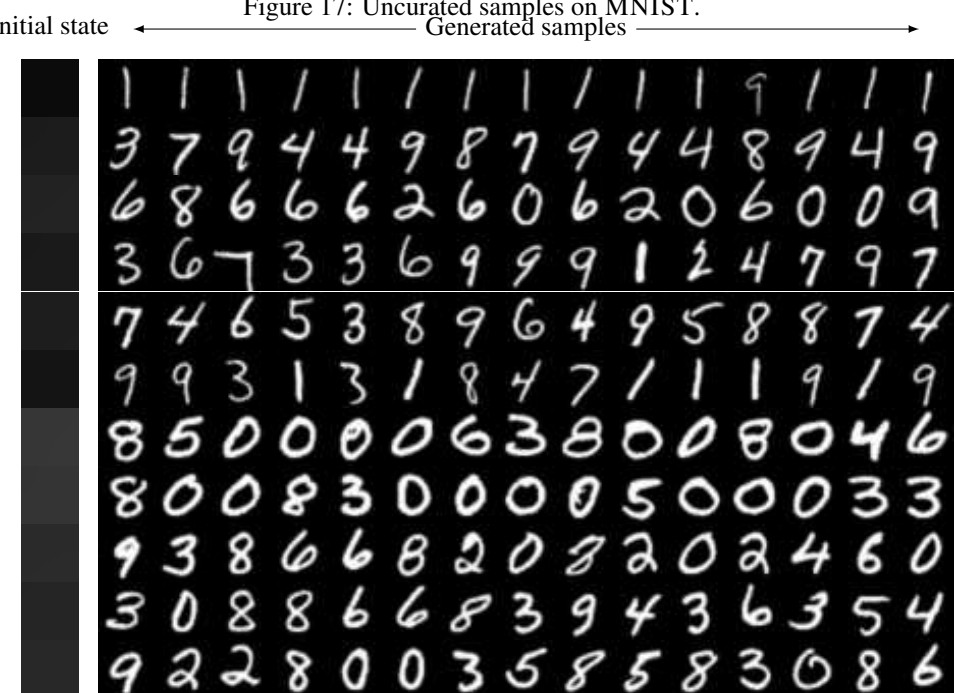

Figure 18: Uncurated samples on MNIST, with shared initial states $\mathbf{u}_K$.

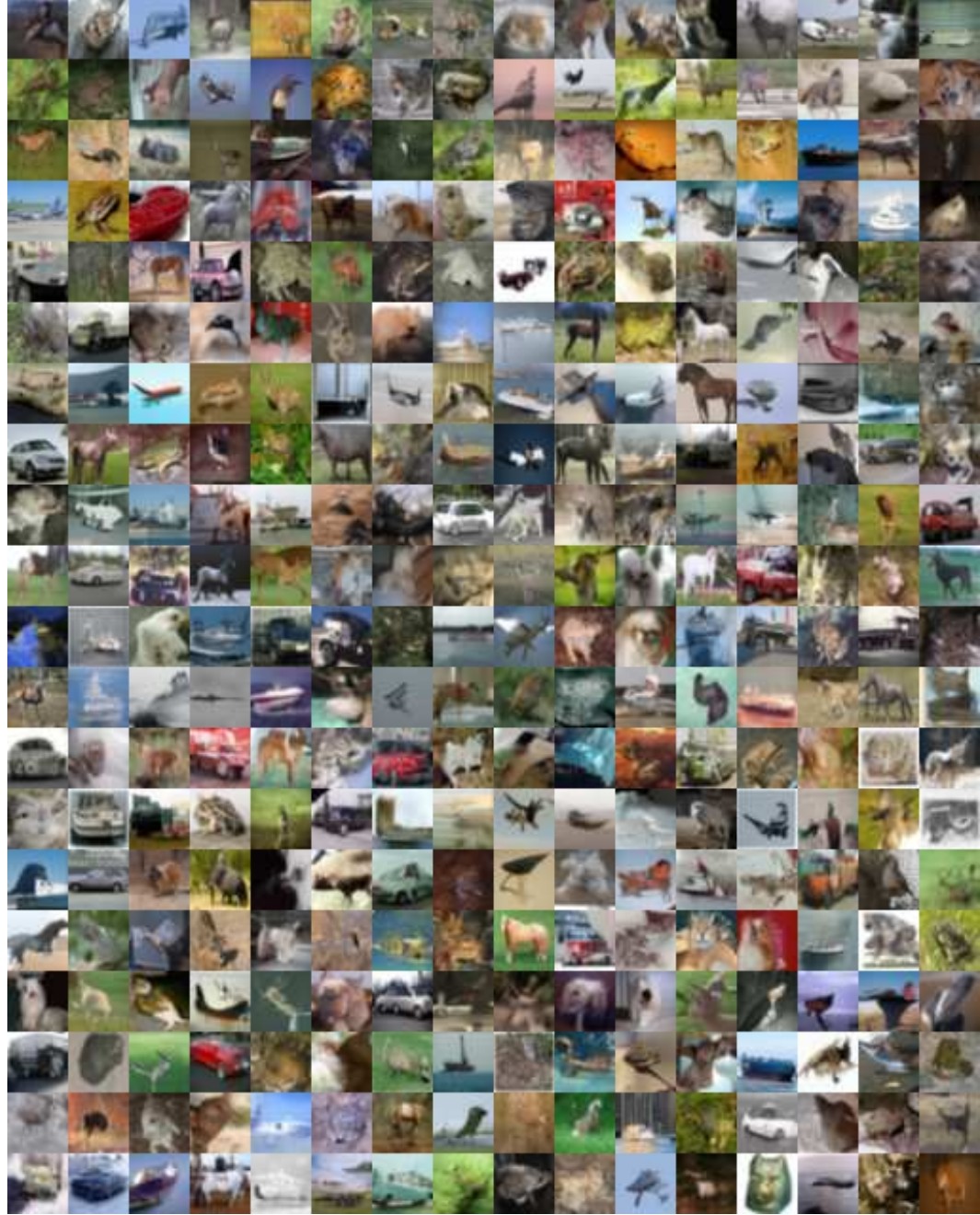

Figure 19: Uncurated samples on CIFAR-10. FID 18.96.

Initial state ← ——————— Generated samples ——————— →

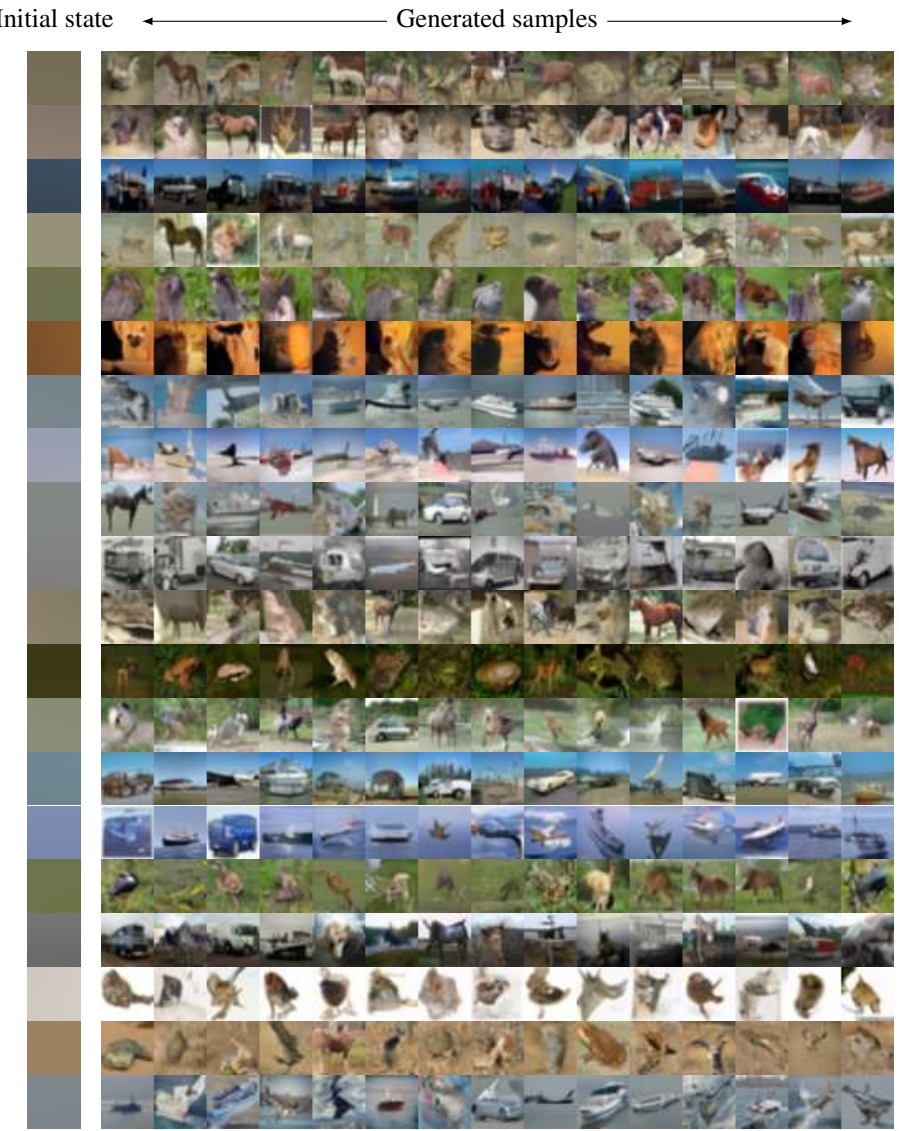

Figure 20: Uncurated samples on CIFAR-10, with shared initial states $\mathbf{u}_K$.

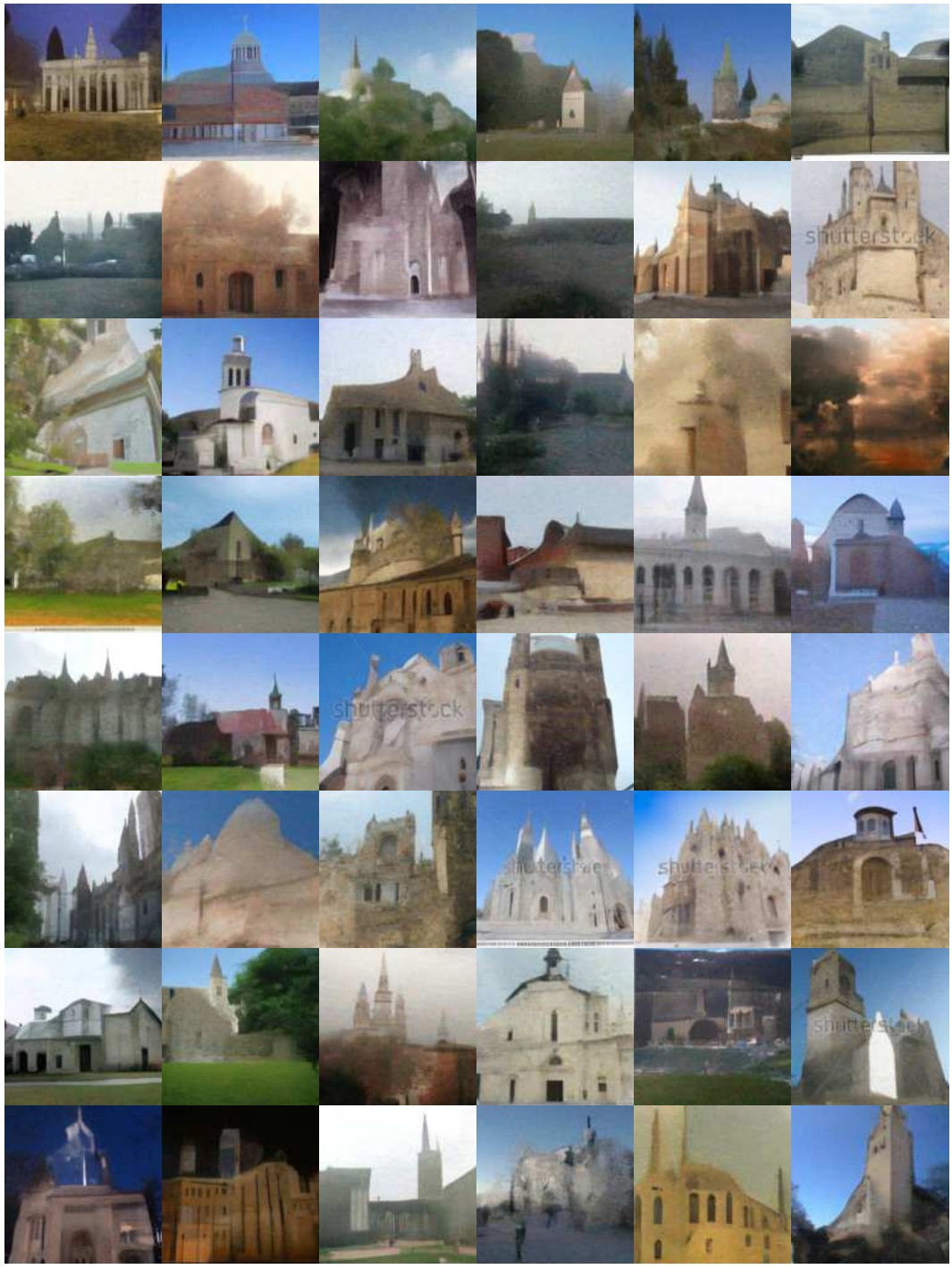

Figure 21: Uncurated samples on LSUN-CHURCHES 128×128. FID 45.06.

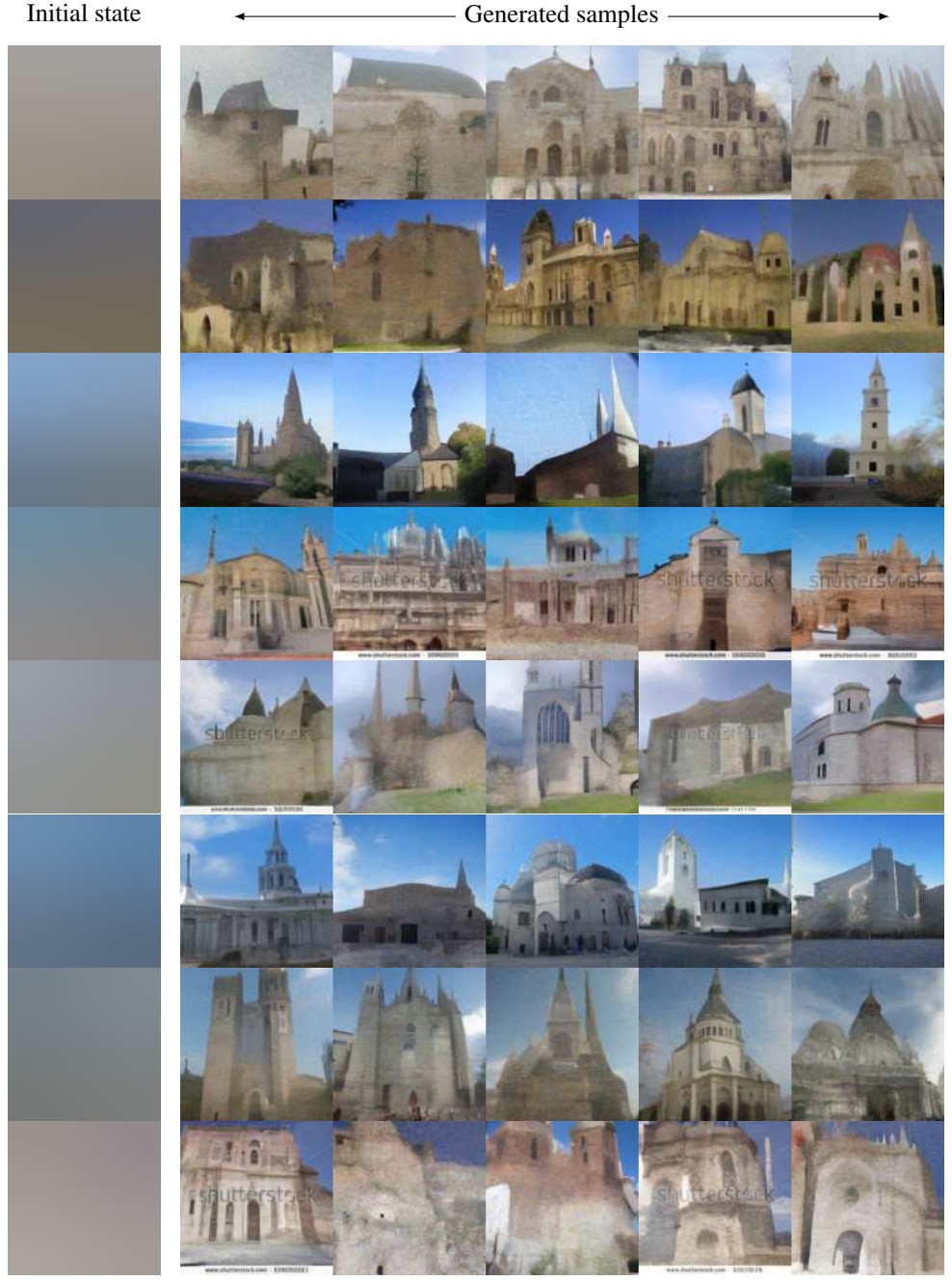

Figure 22: Uncurated samples on LSUN-CHURCHES $128\times128$, with shared initial states $\mathbf{u}_K$.

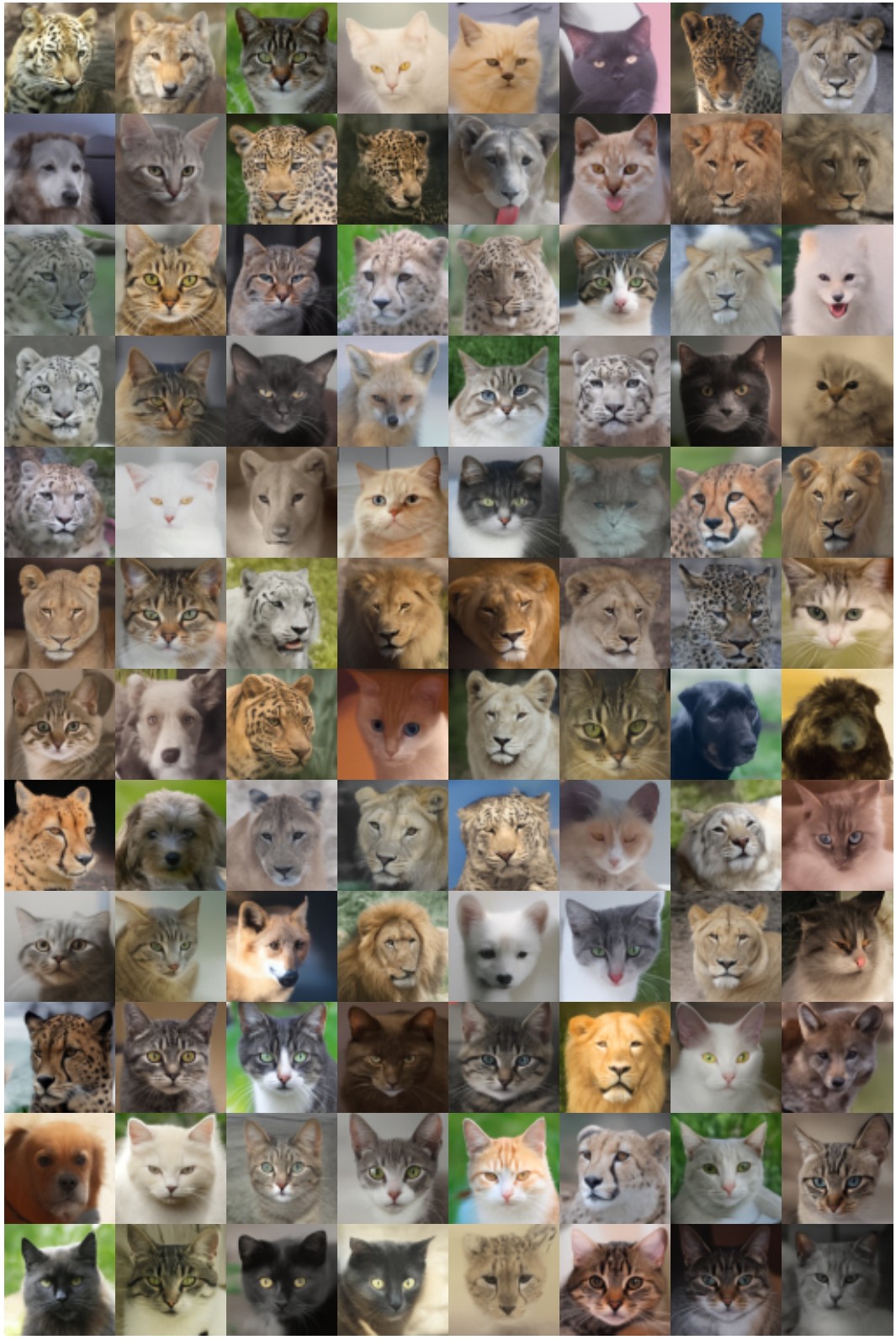

Figure 23: Uncurated samples on AFHQ $64{\times}64$. FID 14.78.

Initial state ←——————— Generated samples ————————→

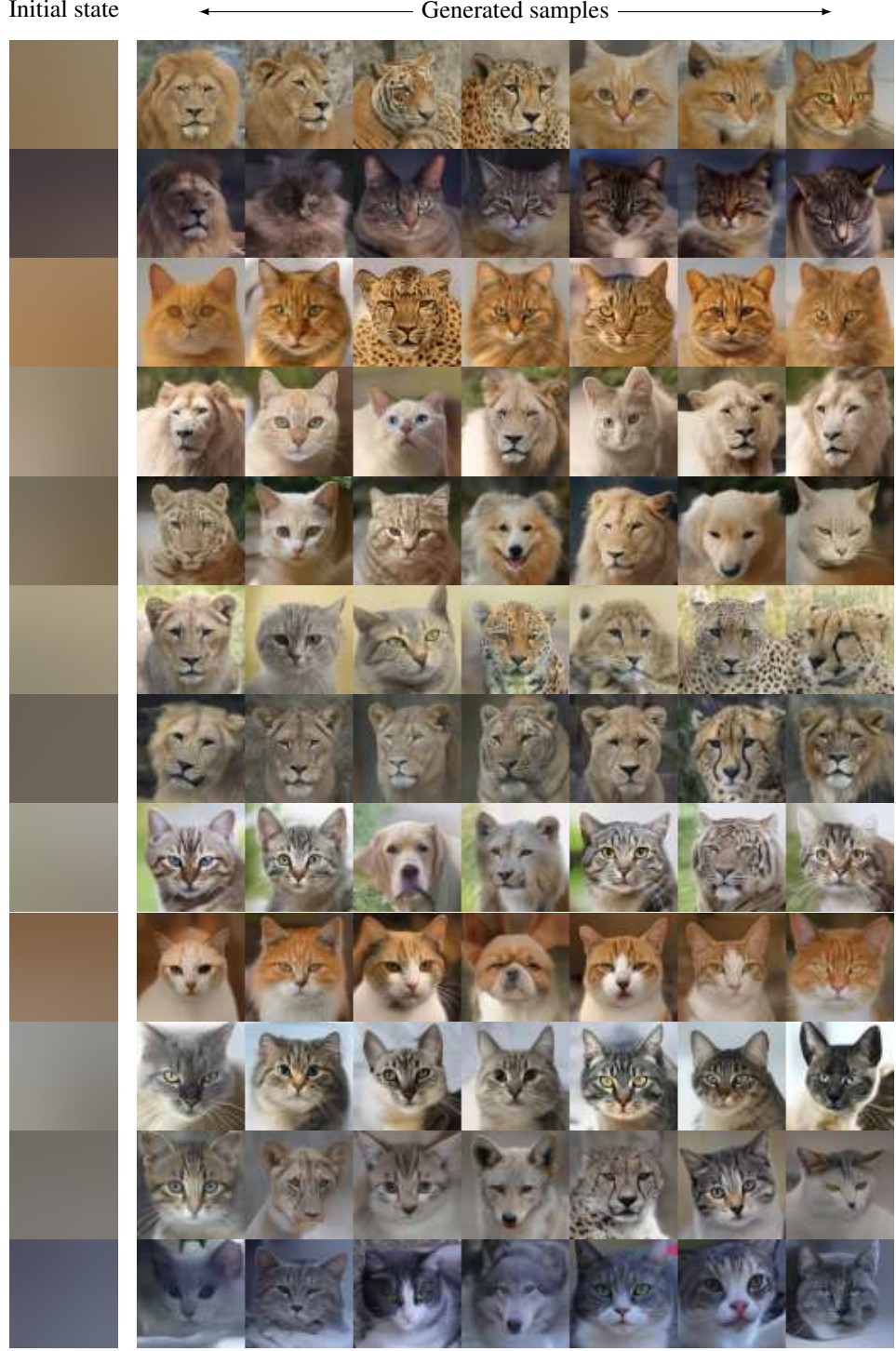

Figure 24: Uncurated samples on AFHQ $64 \times 64$, with shared initial states $\mathbf{u}_K$.

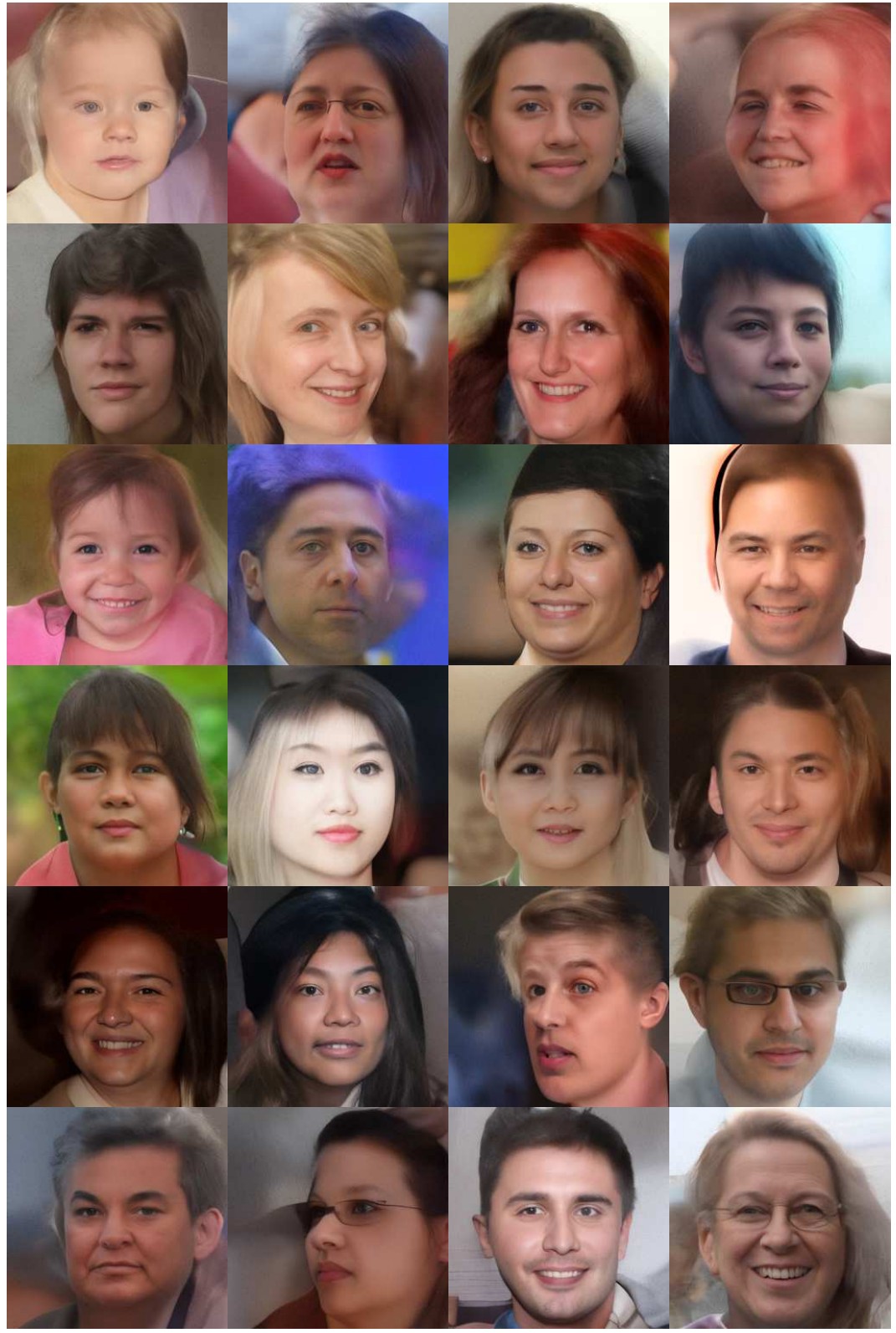

Figure 25: Uncurated samples on FFHQ $256 \times 256$. FID 64.91.

Initial state ⟵ Generated samples ⟶

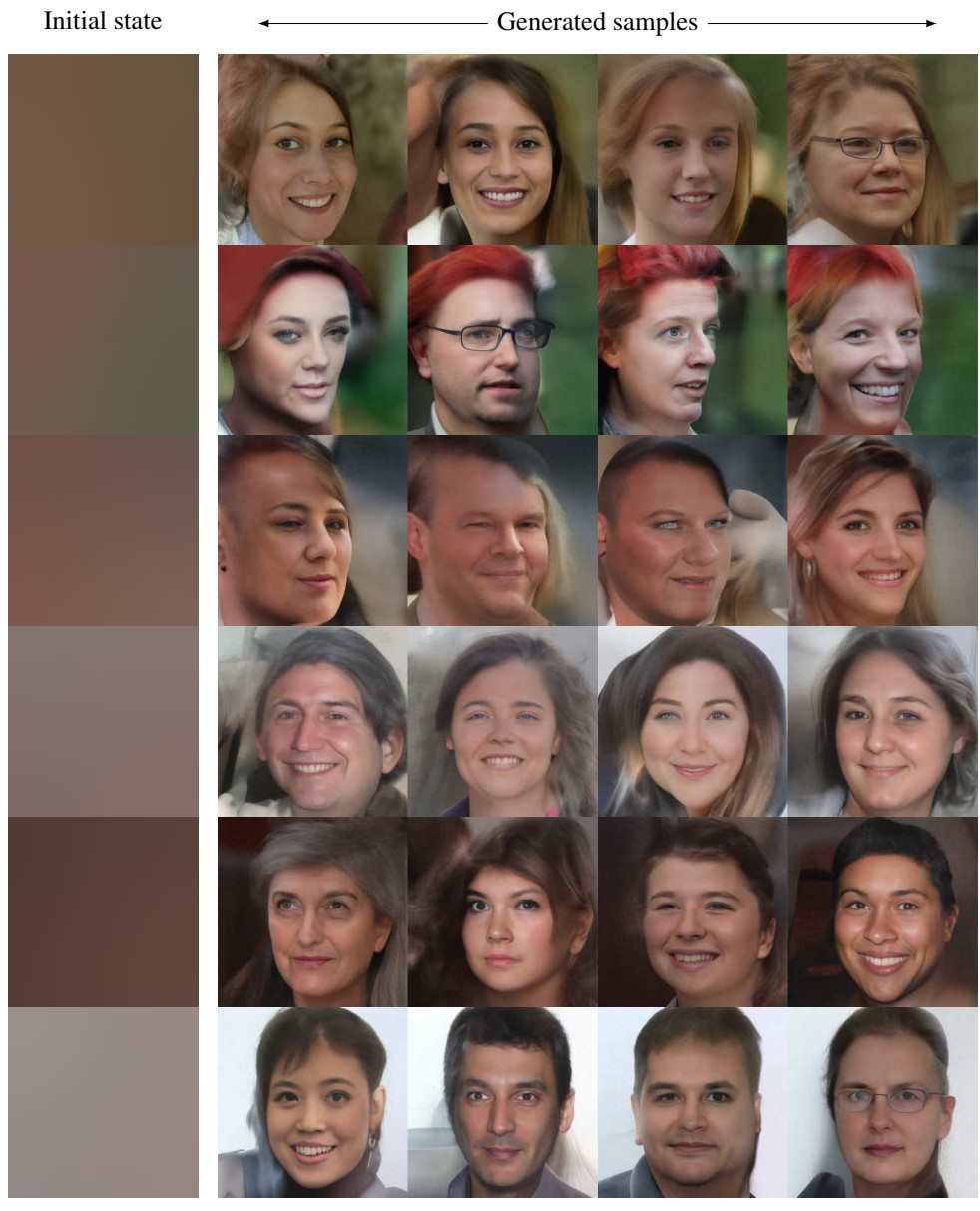

Figure 26: Uncurated samples on FFHQ $256\times256$, with shared initial states $\mathbf{u}_K$.

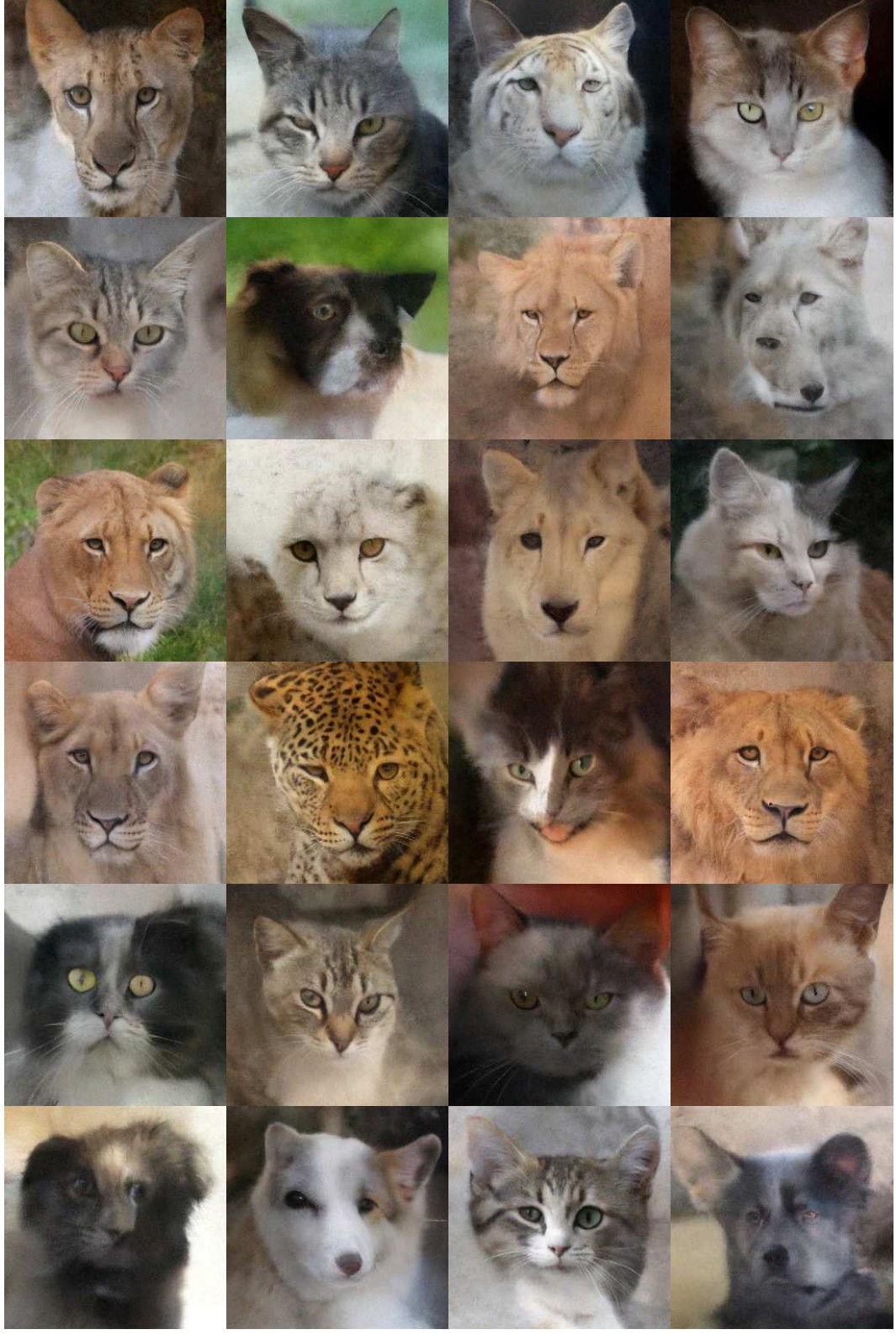

Figure 27: Uncurated samples on AFHQ $256 \times 256$. FID 43.49.

Initial state ←————— Generated samples —————→

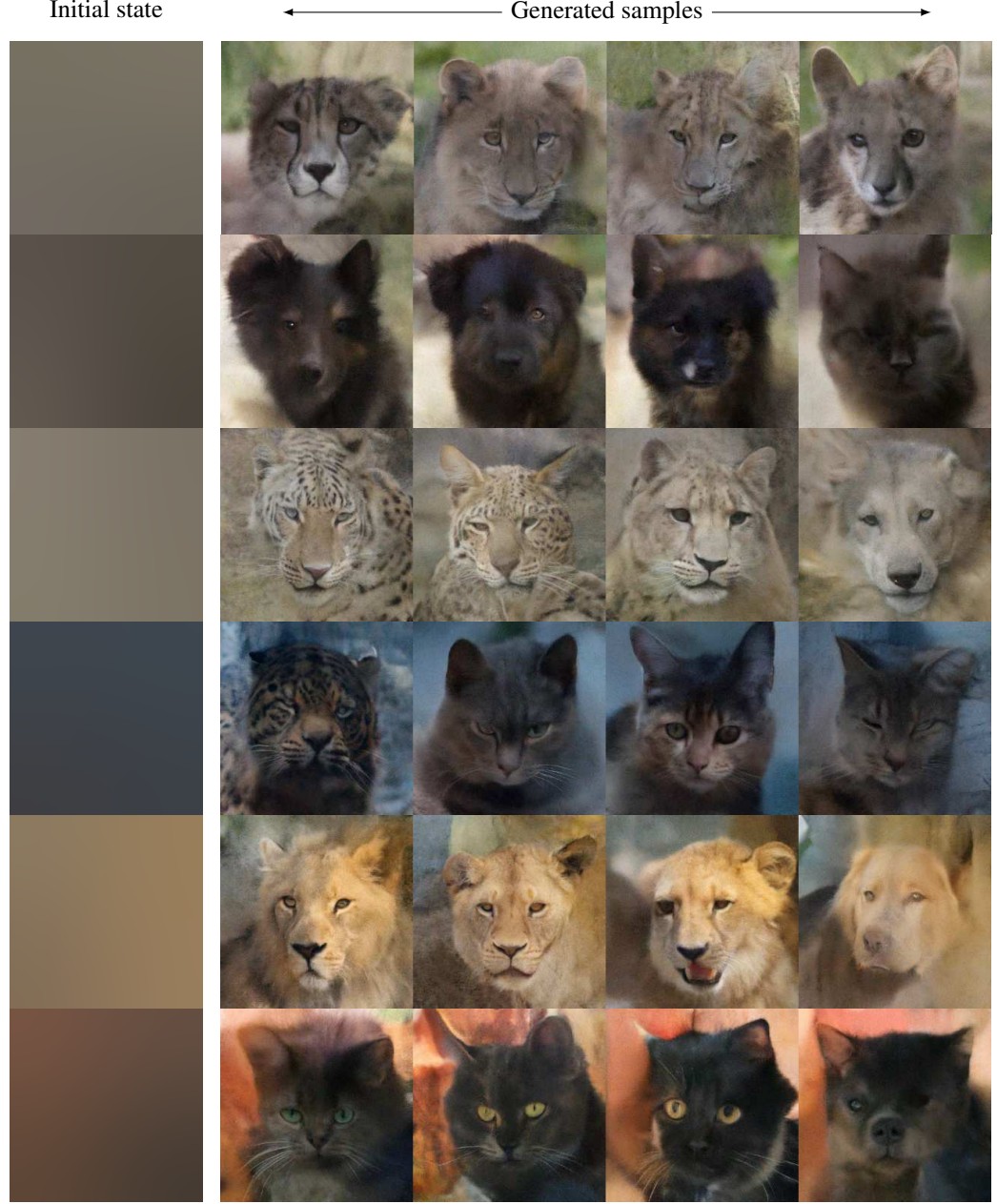

Figure 28: Uncurated samples on AFHQ $256\times256$, with shared initial states $\mathbf{u}_K$.

## D.2 INTERPOLATIONS

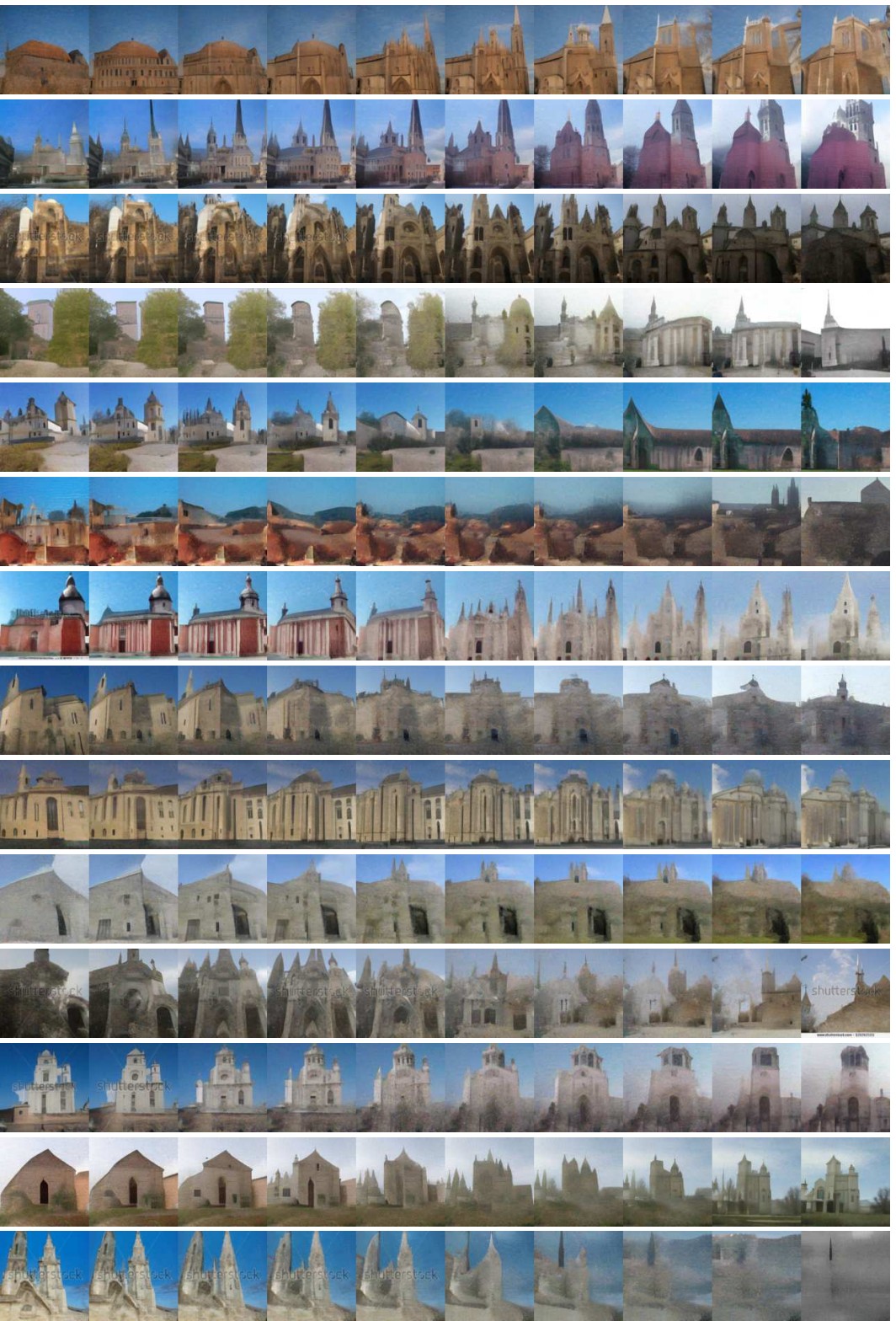

Figure 29: Interpolations between two random images on LSUN-CHURCHES $128\times128$.

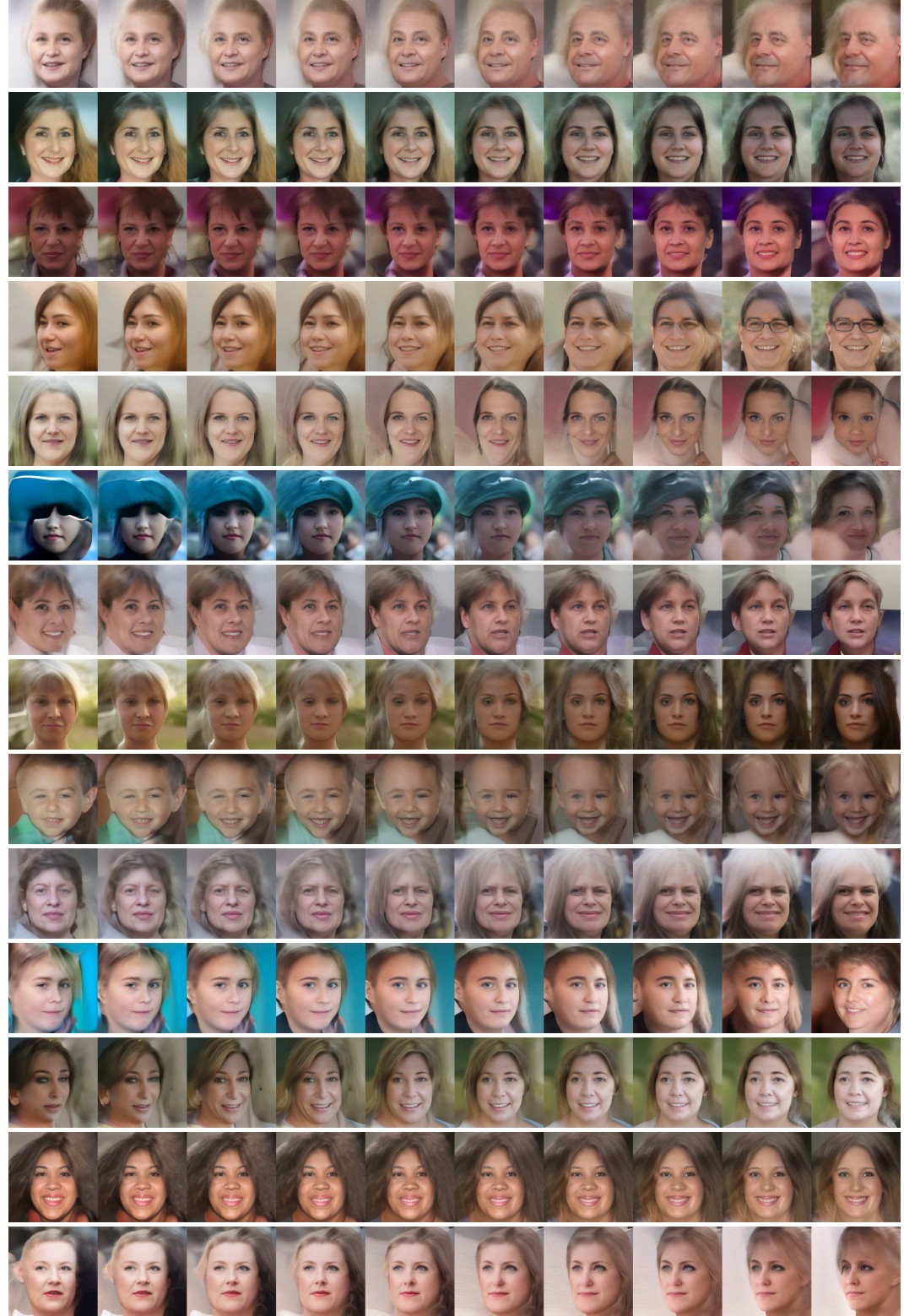

Figure 30: Interpolations between two random images on FFHQ $256 \times 256$.

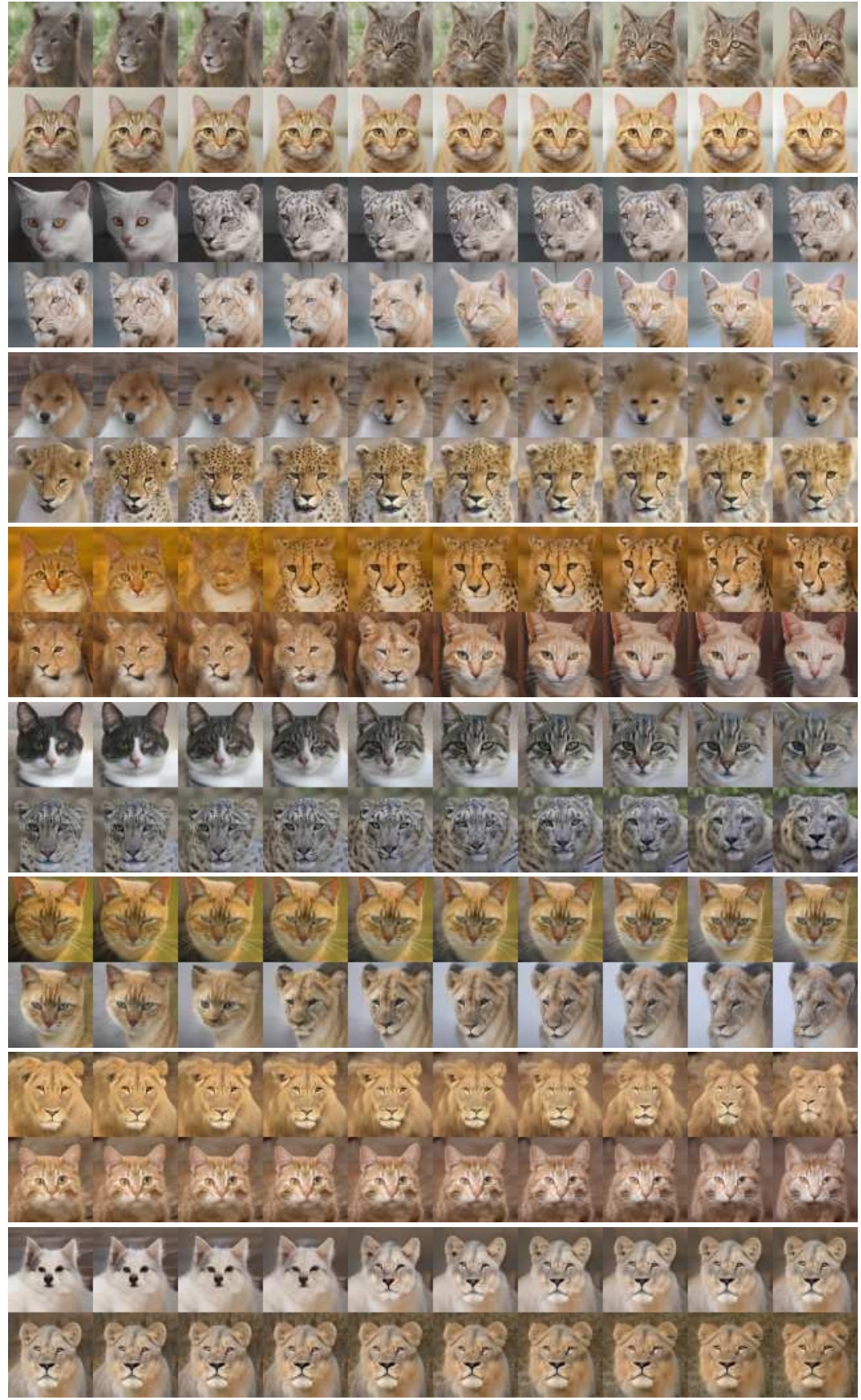

Figure 31: Interpolations between two random images on AFHQ $64\times64$.

### D.3 EFFECT OF SAMPLING NOISE $\delta$

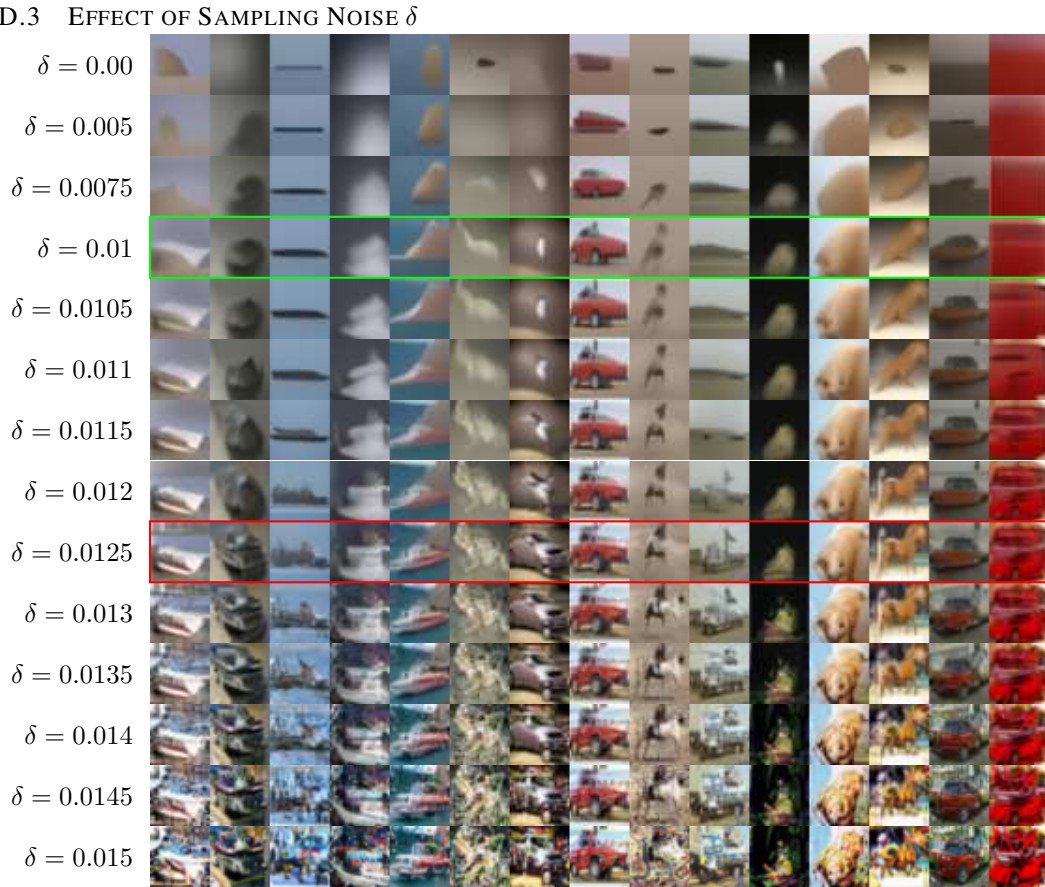

Figure 32: Illustration of the effect of the sampling noise parameter $\delta$ on our CIFAR-10 model. For each column, the sampling noise added during the generative process is sampled only once, and scaled with the different $\delta$ values on the different rows to allow for easier comparison. We highlight $\delta = \sigma = 0.01$, before which changes are slow, and a good default value $\delta = 1.25 \times \sigma$ that works well across data sets.

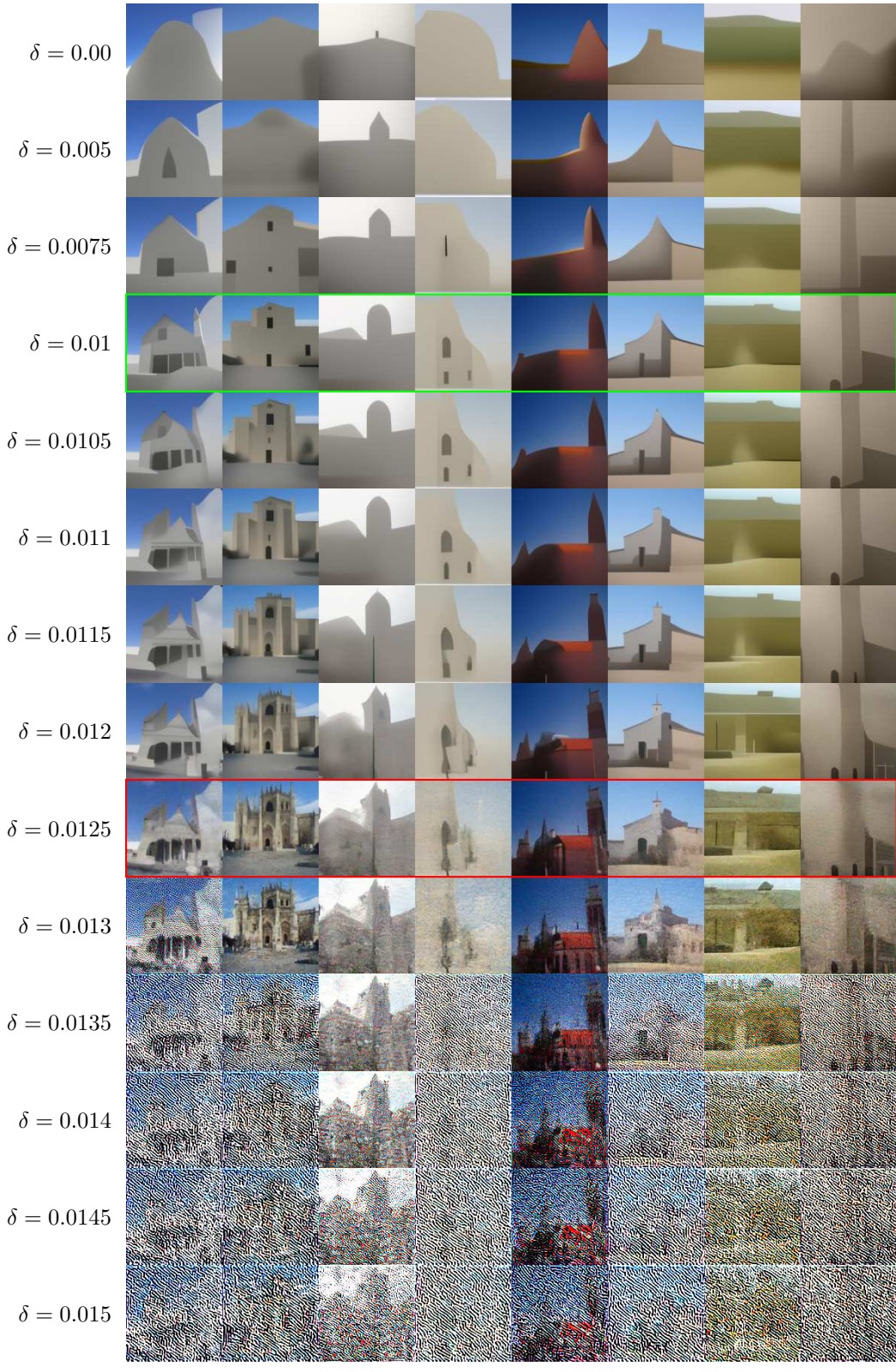

Figure 33: Illustration of the effect of the sampling noise parameter $\delta$ on our LSUN-Churches model. For each column, the sampling noise added during the generative process is sampled only once, and scaled with the different $\delta$ values on the different rows to allow for easier comparison. We highlight $\delta = \sigma = 0.01$, before which changes are slow, and a good default value $\delta = 1.25 \times \sigma$ that works well across data sets.

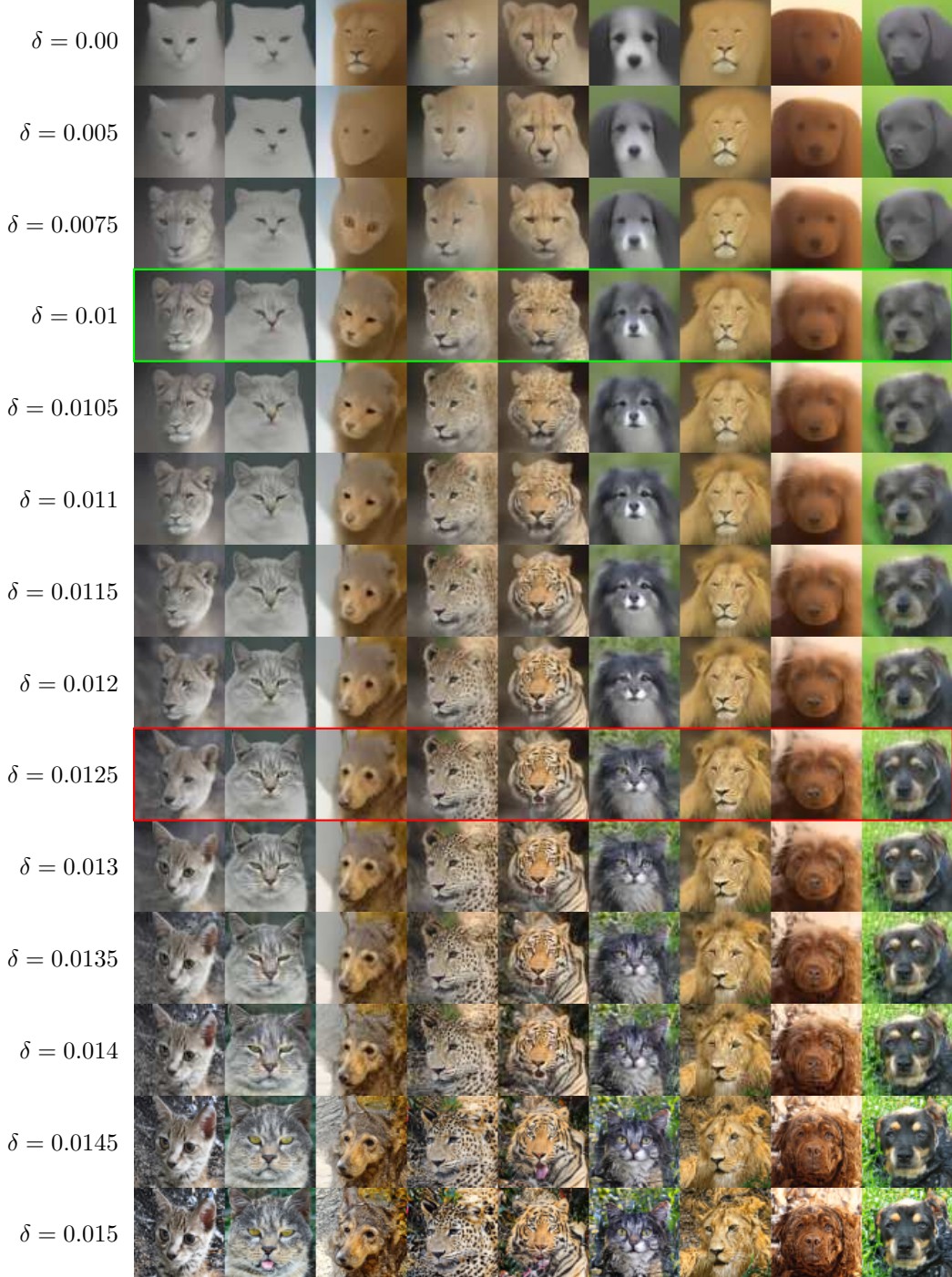

Figure 34: Illustration of the effect of the sampling noise parameter $\delta$ on our AFHQ $64\times64$ model. For each column, the sampling noise added during the generative process is sampled only once, and scaled with the different $\delta$ values on the different rows to allow for easier comparison. We highlight $\delta = \sigma = 0.01$, before which changes are slow, and a good default value $\delta = 1.25 \times \sigma$ that works well across data sets.

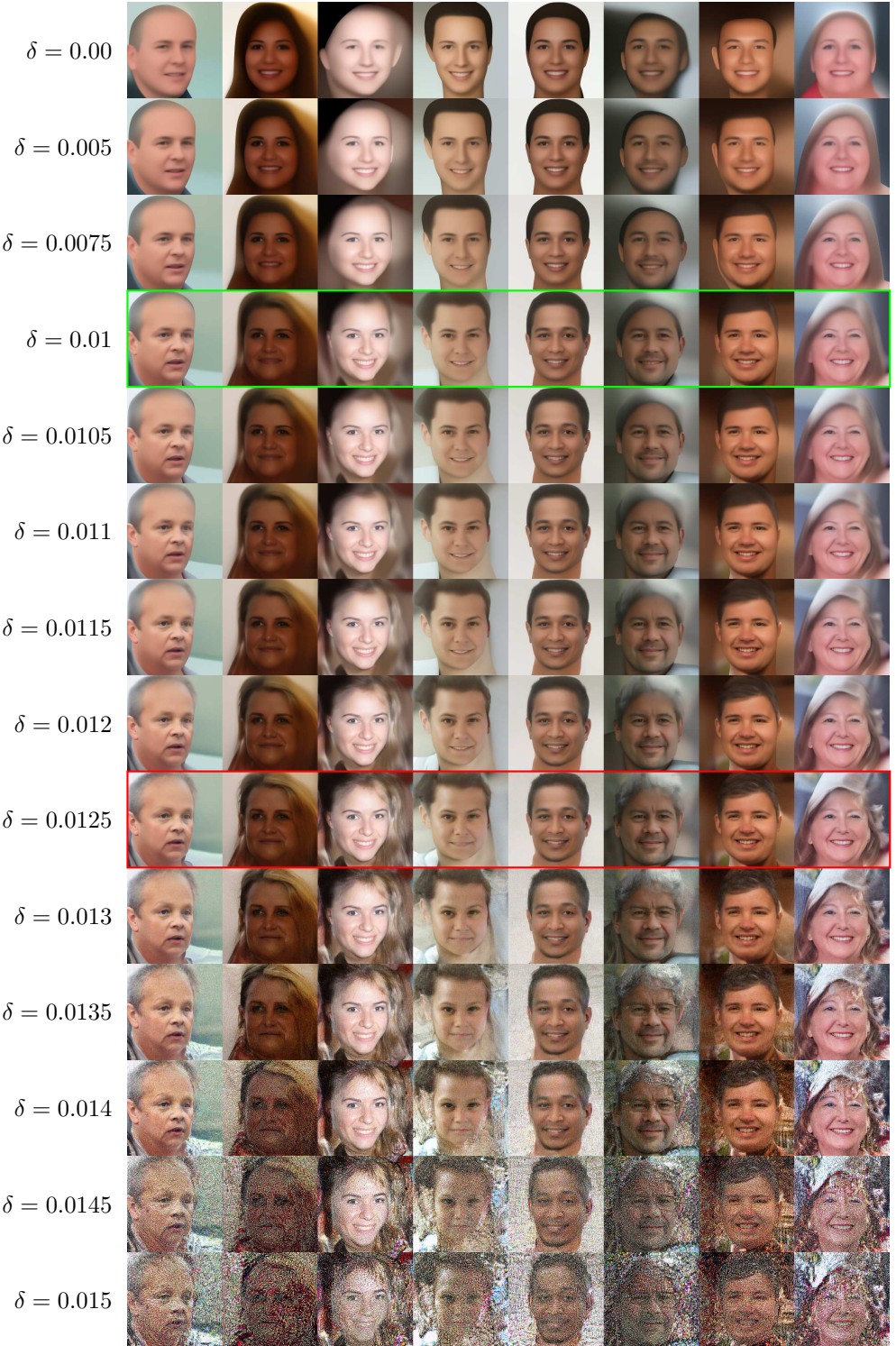

Figure 35: Illustration of the effect of the sampling noise parameter $\delta$ on our FFHQ model. For each column, the sampling noise added during the generative process is sampled only once, and scaled with the different $\delta$ values on the different rows to allow for easier comparison. We highlight $\delta = \sigma = 0.01$, before which changes are slow, and a good default value $\delta = 1.25 \times \sigma$ that works well across data sets. Best seen zoomed in.

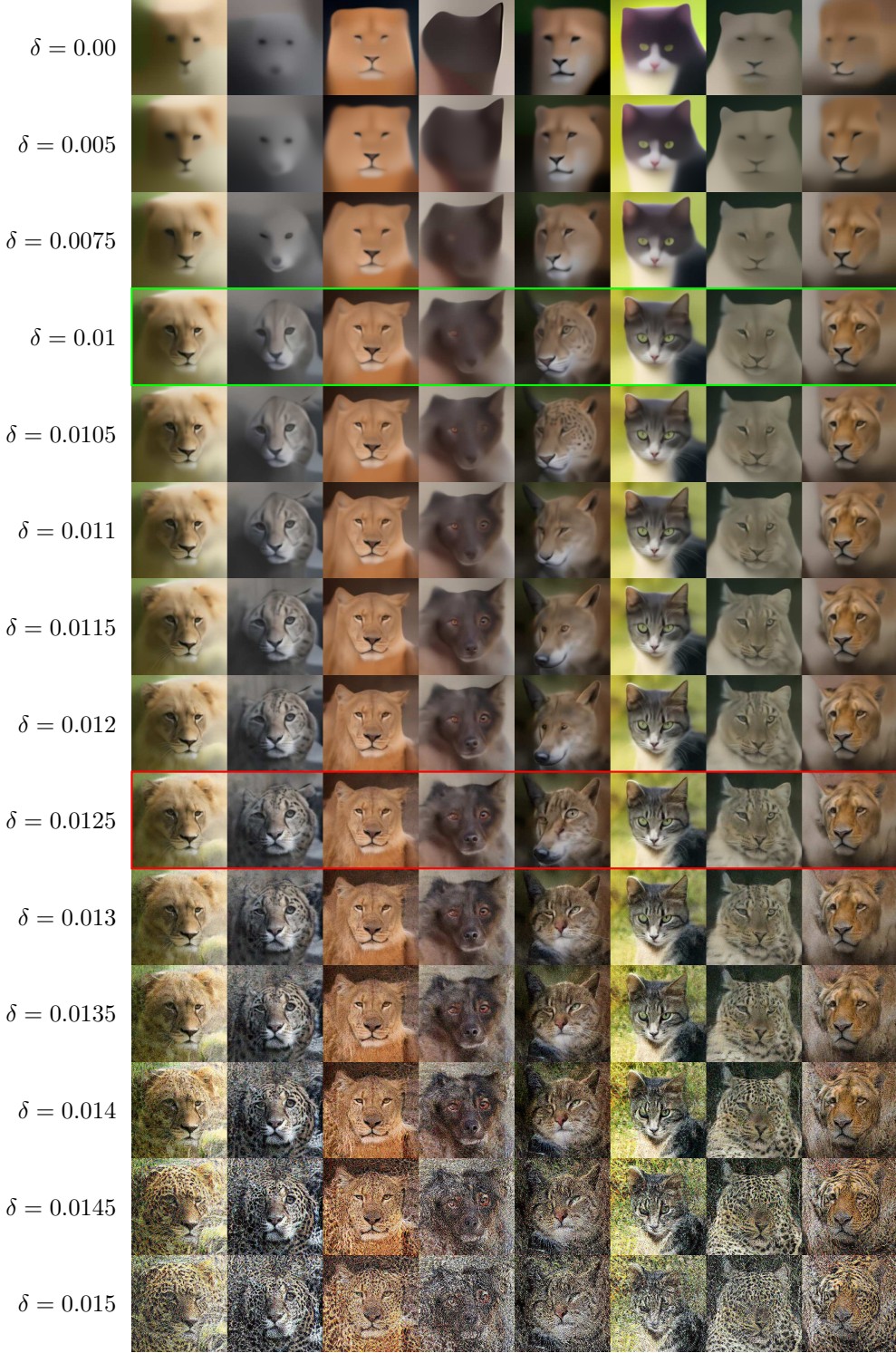

Figure 36: Illustration of the effect of the sampling noise parameter $\delta$ on our AFHQ $256 \times 256$ model. For each column, the sampling noise added during the generative process is sampled only once, and scaled with the different $\delta$ values on the different rows to allow for easier comparison. We highlight $\delta = \sigma = 0.01$, before which changes are slow, and a good default value $\delta = 1.25 \times \sigma$ that works well across data sets. Best seen zoomed in.

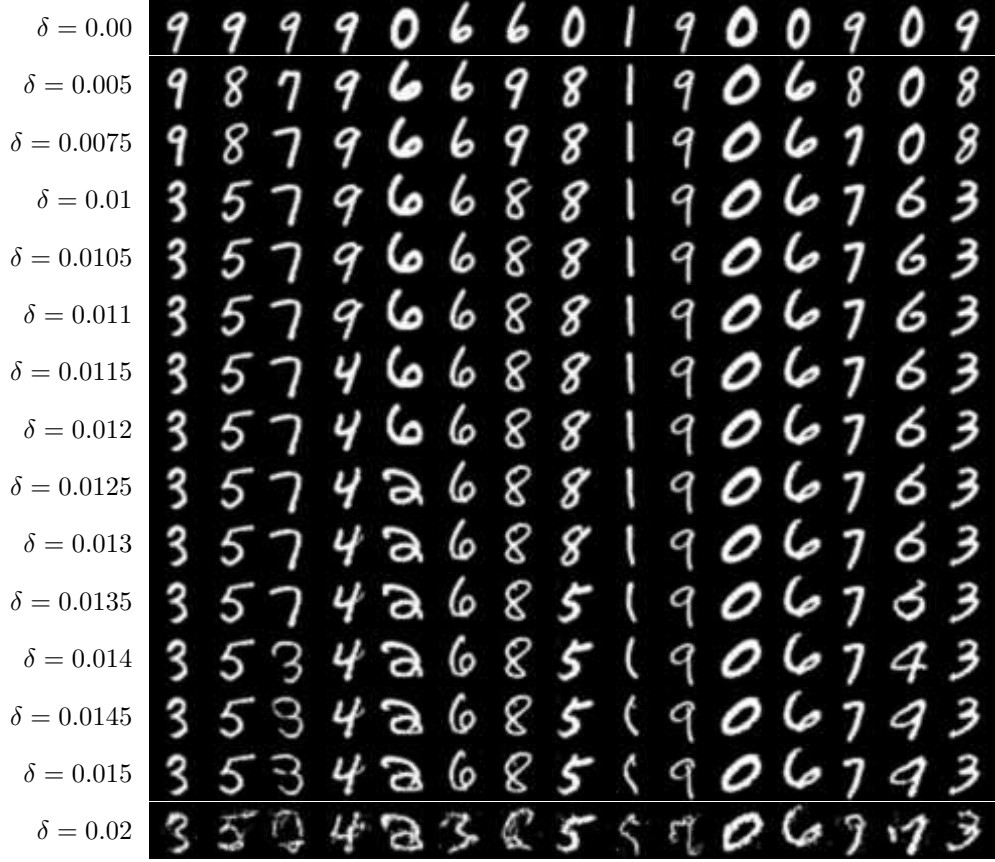

Figure 37: Illustration of the effect of the sampling noise parameter $\delta$ on our MNIST model. For each column, the sampling noise added during the generative process is sampled only once, and scaled with the different $\delta$ values on the different rows to allow for easier comparison. Interestingly, the model performs well with a wide range of $\delta$ on MNIST, including with deterministic sampling. Intuitively, this may be because the MNIST data set does not contain any small-scale details apart from the edges of the digits, and as seen on other data sets, the model is able to reshape the prior image mass into simple shapes with $\delta = 0$. We add results from $\delta = 0.02$ (not shown on other data sets) to show that the results do start degenerating with high enough sampling noise on MNIST as well.

## D.4 DISENTANGLEMENT OF COLOUR AND SHAPE BY FIXING THE SAMPLING NOISE

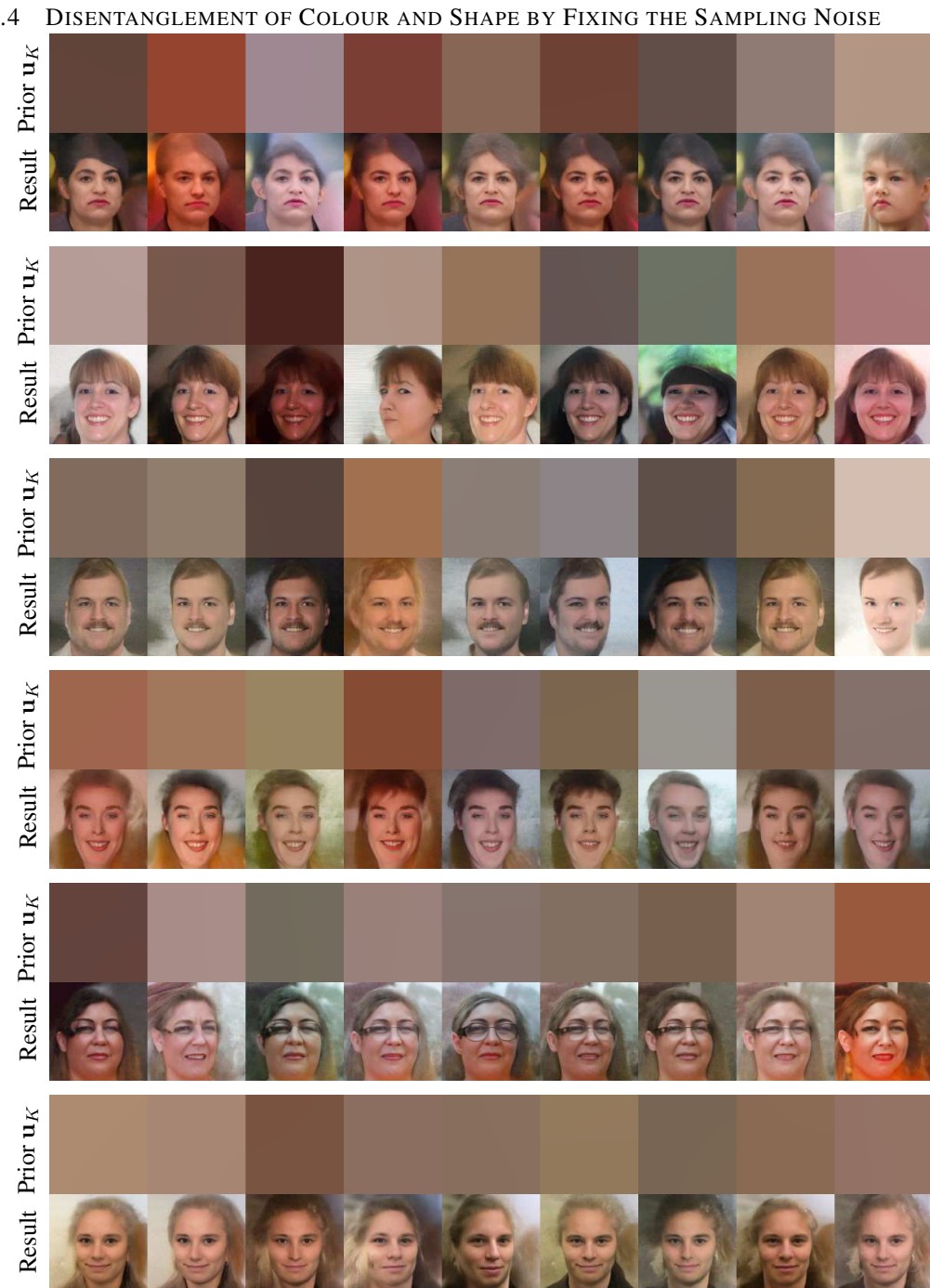

Figure 38: Disentanglement of colour and shape on FFHQ $128\times128$, where $\sigma_{B,\mathrm{max}} = 128$. The noise steps added during the process are fixed, and only the initial state $p(\mathbf{u}_K)$ is changed, resulting in the model carving out images with very similar characteristics, but with different average colours. The sample quality is somewhat lower on this model than the regular FFHQ model, but it illustrates the effect.

## D.5 NEAREST NEIGHBOURS ON SAMPLED IMAGES

Generated image ⟵————————— Nearest neighbours —————————⟶

Figure 39: Samples on FFHQ vs. their Euclidean nearest neighbours on the training data.

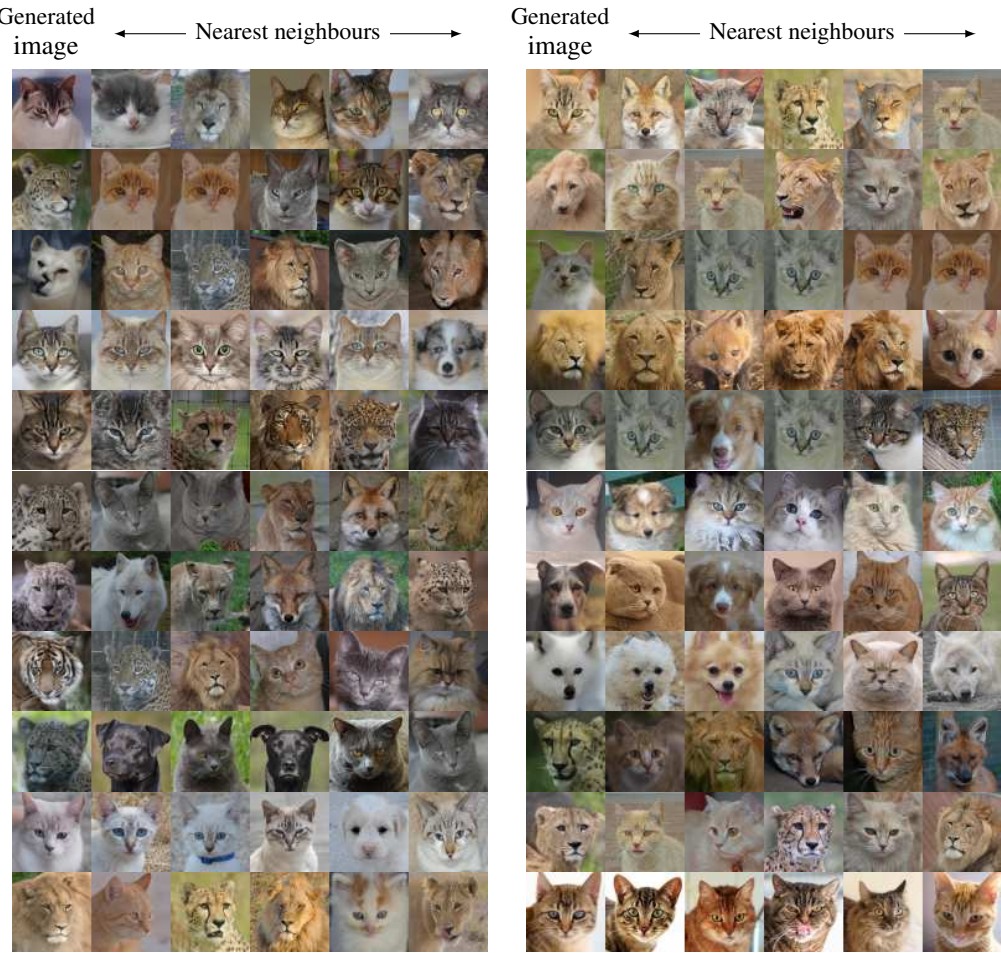

Figure 40: Samples on AFHQ $64 \times 64$ vs. their Euclidean nearest neighbours on the training data.

Generated image ◀——— Nearest neighbours ———▶

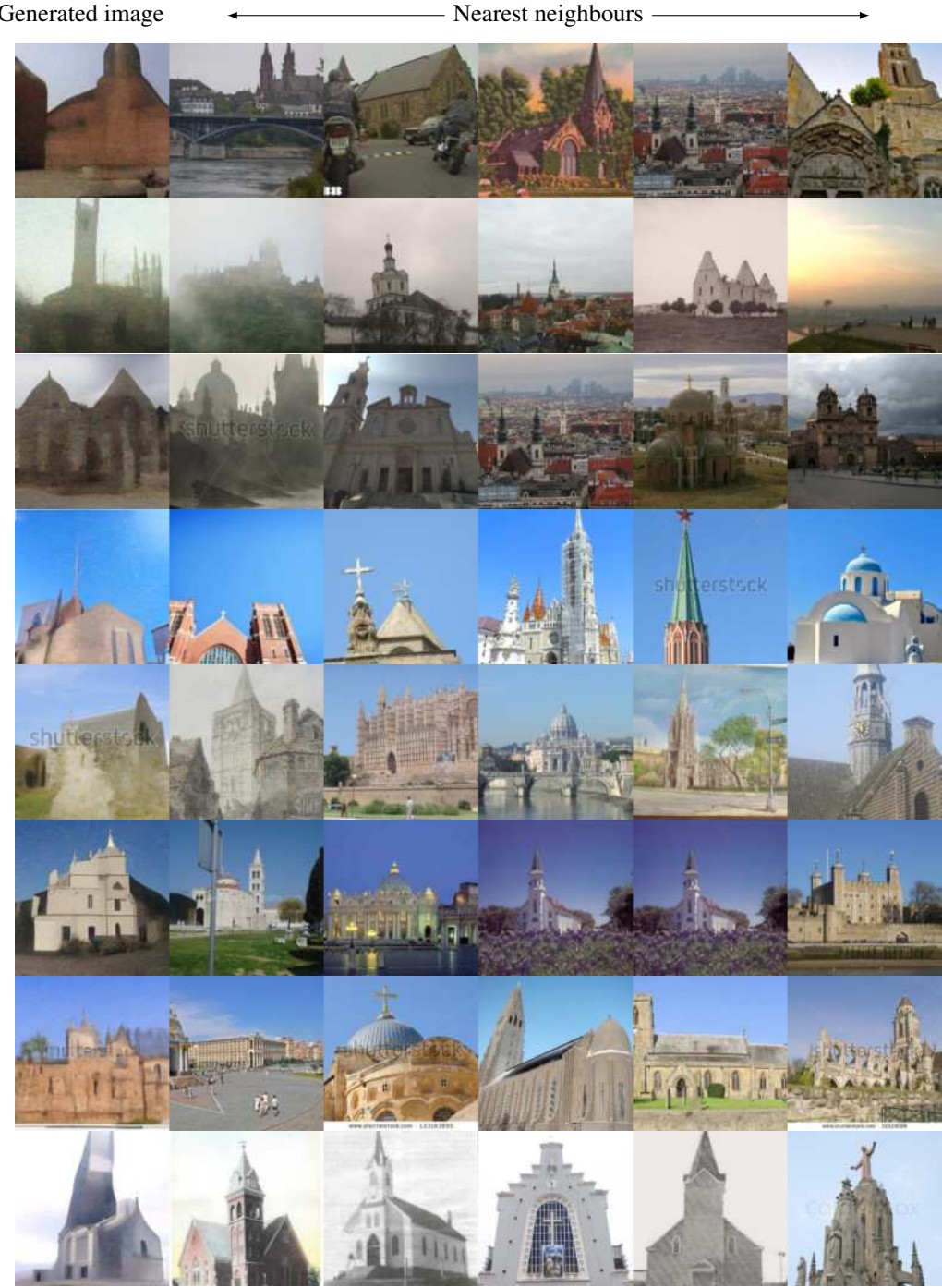

Figure 41: Samples on LSUN-CHURCHES $128 \times 128$ vs. their Euclidean nearest neighbours on the training data.

