# OpenReview forum: "Generative Modelling with Inverse Heat Dissipation"
_ICLR.cc/2023/Conference — ICLR 2023 poster_

### Official Review · Reviewer_ZA3r · 2022-10-13

**Confidence:** 3
**Correctness:** 3
**Technical Novelty And Significance:** 2
**Empirical Novelty And Significance:** 3
**Recommendation:** 5

**Clarity, Quality, Novelty And Reproducibility:**

Questions:

1. Can you give a definition of the power spectral density (PSD) of natural images?
2. Seems the prior is estimated using KDE ---- that means we need extra space complexity for that model?

Overall, the clarity is good. The novelty is less due to my comments in weakness.

**Strength And Weaknesses:**

Strength:
1. Overall the idea is valid and interesting;
2. This paper provides several interesting analysis such as PSD analysis.
3. Lots of experiments are conducted to show interesting properties of the model.

Weakness:
1. There is not much supportive evidence to demonstrate the unique advantage of using heat dissipation. This greatly reduces the novelty of the paper. I believe there should be some important undiscovered advantages to using this approach. As the effective resolution is changing --- can it save the inference time?

2. There are several numerical approach to deal with the Laplace ops: 1. the eigendecomposition used in this paper; 2. directly using Gaussian blurring; 3. grid-based finite difference; To me there seems no significant difference between these three approaches. Although author mentioned numerical stability and some gradient analysis, I would say it is worth really running those models to exam the outcomes.



**Summary Of The Paper:**

This paper proposes to use heat dissipation to define a forward process (blurring) in the diffusion model. It uses eigendecomposition to decompose the Laplace operation and uses a variational model, similar to DDPM to learn the generative process. Gaussian noise is added to ensure the diversity of the generated images.

Overall using different blurring ops is an interesting exploration direction.

**Summary Of The Review:**

Please see above. I will adjust my score based on the rebuttal.

---

> ### Author Response · Authors · 2022-11-07
> **Reply to reviewer ZA3r**
>
> We thank the reviewer for the very useful comments and questions, and also the interest towards our work. Answers to concerns in order received:
> - **Unique advantage of heat dissipation.** This is an excellent question. Based on the results, it appears that restricting the generative process to move along the reverse heat equation effectively regularises the model and incorporates prior information about natural images. The main potentials are then new properties and aspects of generalisation, such as the ability to combine different shapes with different average colours, smooth interpolation and potential to generalise from limited amounts of data (experiment on the 20 first MNIST digits). This has the tradeoff that too much regularisation can also hurt the visual quality in some settings, which is likely what happens with the current model. It is possible that the current idea can, e.g., be mixed with standard diffusion models to relax this regularisation to an optimal level.
> - **Other numerical approaches with the Laplace operator.** It is a good point that other methods for simulating the heat equation exist, and indeed the difference would likely not be large between them. Our motivation for the Fourier-space model is that it is efficient, easy to implement, and exposes directly the intuitions about frequency decay. In contrast, a finite-difference scheme would be very slow due to sequential computations and the large simulation times that we need for large levels of blur. A Gaussian blur implemented as a sampled convolutional filter could very well work, however. This could indeed be interesting as well, so we are now training a model on CIFAR-10 that is based on it. Half-way during training, the FID score is 27.2, as opposed to 18.96 for the fully-trained DCT-based model. We will update this. EDIT: The model training has now completed. The achieved FID score is 22.44, which is still slightly worse than the DCT-based model. We have added this result in Appendix C.6 along with implementation details.
> - **PSD definition.** We define the PSD as follows: We first take the 2D DCT/DFT of the image. We then take the absolute values of each frequency component and square them, giving us a PSD value for each frequency component in the 2D frequency plane. These values are averaged as shown in Fig.10 in App B.6. to get the 1D power spectral densities. We have now added this explanation in Appendix B.6., and refer to it in Section 2.2 of the main paper.
> - **Extra space complexity for the prior.** Indeed, using the KDE prior requires saving some information from the training data, but given that the blurry data points are effectively very low-dimensional, the added space complexity is minor. We could use, e.g., PCA to compress the blurry data to a few dimensions, or have a learnable prior. In case you are interested in more details, see our answer to reviewer D1k3.

---

### Official Review · Reviewer_D1k3 · 2022-10-24

**Confidence:** 4
**Correctness:** 4
**Technical Novelty And Significance:** 3
**Empirical Novelty And Significance:** 3
**Recommendation:** 6

**Clarity, Quality, Novelty And Reproducibility:**

* Quality is high, experiments are well-conducted and analyzed.
* Clarity is moderate, with too many details cluttering the main text.
* Originality is high.
* Reproducibility is high - the authors included the source code in a supplementary material.

**Strength And Weaknesses:**

\+ This is an interesting topic and it improves our understanding of what is possible in generative modeling.

\+ The experiments are well-analyzed.

\- The main text of the paper contains excessive details which make it harder to read.

\- Experimental results do not show that the proposed method is a promising direction, at least in terms of the sample quality.

Questions/minor comments:
* "Their generative process does not explicitly consider the inductive biases of natural images, such as their multi-scale nature" - Imagen (https://arxiv.org/abs/2205.11487) does use upsamplers, some of its predecessors also do.
* This work is not just heat dissipation, but rather a combination of heat dissipation with gaussian noising - this needs to be highlighted early in the paper. "noise-relaxed solution of the forward heat equation" in the abstract sounds rather obscure, just saying that it's a combination with gaussian diffusion would be more clear.
* " We obtain samples by taking a training example, blurring it with F(tK), and adding noise with variance δ2" - does it mean that the prior is non-parametric, so we would need the training set during the generation time? That sounds like a limitation which should be mentioned early on?

**Summary Of The Paper:**

The paper proposes to combine heat dissipation forward process with gaussian diffusion.

**Summary Of The Review:**

This is an interesting work proposing a novel approach to generative modeling of images based on heat dissipation. I am learning towards acceptance.

---

> ### Author Response · Authors · 2022-11-07
> **Answer to reviewer D1k3**
>
> We thank the reviewer for the very useful comments and questions and noting the new understanding that our paper brings. We answer the concerns in the order received:
> - **Promise of the proposed method.** While it is a fair point of critique that results are not yet close to SOTA, we would like to point out that the method is quite novel and all the established best practices developed for diffusion models do not apply here. Furthermore, we demonstrated promising properties induced by the blurring process, such as the disentanglement of colour and shape and potential for more effective generalisation from limited data (the 20 MNIST digits experiment). A flip-side is that this also appears to have a regularising effect, potentially smoothing the distribution too much. A possible direction for future work would be to find ways to combine heat dissipation with standard diffusion, hopefully relaxing the regularisation of IHDM while also improving diffusion model generalisation.
> - **Imagen and upsampling methods.** Indeed, cascading multiple super-resolution diffusion models has been noted to improve diffusion model performance in multiple works. The difference to our work is that we consider the resolution-increasing aspect a core part of the model itself, instead of a performance-boosting tweak to diffusion models. The related works section already contains some discussion on this, but we expanded it to cite DALLE-2 and Imagen as well, and clarified the wording in the introduction:“ The forward process and reverse processes of standard diffusion models do not explicitly consider the inductive biases of natural images”.
> - **Clarification of heat dissipation and noise.** We agree that the phrasing could be clarified, and we have updated it to the following: “We interpret the solution of the forward heat equation with small additive noise”. We believe that diffusion is not quite the right word here, as the formal forward process does not include a diffusion (Markov process).
> - **Needing training data during sampling.** Indeed, using the KDE prior requires some information from the training data, but given that the blurry data points lie on a very low-dimensional manifold, this is not an issue. We used the KDE as a good-enough option for our research purposes. Solutions that don’t require memorization of the entire data set include:
>   -  Compressing the blurry data, e.g, with PCA. In the limit of infinite blur, the data points are compressed down to the mean RGB values of the images, $\mathbb{R}^3$, and three floating point numbers per image are enough.
>   -  Using a learnable prior. A simple solution is fitting a multivariate Gaussian. One could also, e.g., train a low-dimensional VAE.
>   -  Using other boundary conditions for the heat equation, e.g. Dirichlet. In that case, all images are mapped to black colour, and the generation can always start with a black image if the forward process is run far enough.

---

> > ### Comment · Reviewer_D1k3 · 2022-11-15
> > **Reply**
> >
> > Thank you for addressing my questions. I am generally satisfied with the answers, although please mention KDE prior early on in the text - it should be clear to the reader that the method requires a more complicated prior than usual for diffusion methods.
> >
> > I am keeping my score and still leaning towards acceptance.

---

> > > ### Author Response · Authors · 2022-11-24
> > > **Reply to reviewer D1k3**
> > >
> > > Thank you for the update! We will add a comment about the KDE in the introduction in the potential camera-ready stage, apologies for the late reply.

---

### Official Review · Reviewer_kPGb · 2022-10-25

**Confidence:** 4
**Clarity, Quality, Novelty And Reproducibility:** Good.
**Correctness:** 3
**Technical Novelty And Significance:** 4
**Empirical Novelty And Significance:** 3
**Recommendation:** 8

**Strength And Weaknesses:**

Strengths:
This paper proposes a new type of generative model based on inverse heat dissipation, which possesses a different mechanism compared to existing diffusion models. Moreover, a solid and extensive mathematical derivation of method details has been provided in the Appendix. The proposed method shows emergent qualitative properties not seen in existing diffusion models, which has also been proved by experimental results.

Weaknesses:
From the qualitative results shown in this paper, it seems to me that existing SOTA generative models (e.g., StyleGAN, IMAGEN, DALLE) clearly outperform the proposed method. Yet, since the paper opens a new way for designing generative models, it is acceptable for me that the quality of synthesis results obtained by the proposed one is not as good as SOTA ones. However, there is no qualitative and quantitative performance comparison between the existing methods and the proposed one. Please explain the reasons. In fact, no matter whether the performance is good or not, it is still necessary to compare the proposed method against existing approaches.


**Summary Of The Paper:**

This paper proposes a new generative model by using inverse heat dissipation. The proposed model is motivated by diffusion models and the empirical success of coarse-to-fine modelling. Specifically, the authors propose to synthesize images by iteratively inverting the heat equation, which is basically a PDE that locally erases fine-scale information. Furthermore, they theoretically interpret the noise-relaxed solution of the forward heat equation as a variational approximation in a diffusion-like latent variable model. Extensive theoretical and experimental analyses have been conducted to demonstrate the effectiveness of the proposed model.

**Summary Of The Review:**

This paper proposes a new generative model which is both theoretically and technically sound. Extensive mathematical deviations of method details have also been provided making it a solid paper. Despite the above-mentioned weaknesses, I believe that this paper is still deserved to be published in ICLR after some necessary revisions. The proposed method is interesting and shows up some valuable potential for future research.

However, the quality of synthesized images of the proposed method is not as good as the state of the art, and there is no comparison between existing methods and the proposed one. Otherwise, I could give it a higher score.

---

> ### Author Response · Authors · 2022-11-07
> **Reply to reviewer kPGb**
>
> We thank the reviewer for acknowledging the novelty of our work and the very thoughtful comments. Initially we did not include explicit FID score comparisons to all the state-of-the-art methods, as our scores are not very close to them and at first judged it enough to say that the method is not competing against them. However, we now see and agree that it would be useful for the reader to have explicit comparisons, and have now included the FID scores of a baseline diffusion model (DDPM) and the current state-of-the-art on CIFAR-10 (StyleGAN-XL) in the Experiments section (“Quantitative comparison”), as well as explicit pointers for qualitative performance comparisons. We have also increased the font size of the FID scores in Fig.5. and highlight them in the caption. CIFAR-10 is the most widely reported data set, making it good for comparisons.

---

> > ### Comment · Reviewer_kPGb · 2022-11-29
> > **Reply to the authors**
> >
> > Thanks for the authors' response. I think it is necessary to add those comparison results no matter whether the proposed method could outperform them or not. The key advantage of this paper is its novelty and thus I want to keep my initial score (8: accept, good paper).

---

### Official Review · Reviewer_dZbG · 2022-10-30

**Confidence:** 4
**Correctness:** 3
**Technical Novelty And Significance:** 3
**Empirical Novelty And Significance:** 3
**Recommendation:** 6

**Clarity, Quality, Novelty And Reproducibility:**

Clarity & Quality:  the paper is clearly written and easy to follow. The mathematical formulation is technically meaningful.
Novelty: the paper proposes a novel method by inverting a stochastic heat equation, which only extends standard diffusion models with Gaussian noise.
Reproducibility: the paper includes a detailed description of implementation and training details.



**Strength And Weaknesses:**

Strength:
-- An interesting study is one of the first several works applying blurring transformations in the diffusion process. Unlike existing works, the paper directly derives the formulation from the heat equation SDE and analyzes the inductive bias compared to the standard diffusion models.

Weakness:
-- The empirical results are relatively much worse than standard diffusion models. The paper seems to hide the quantitative comparisons (only small captions in the figure). However, the visual quality is also over-smoothed compared to the baselines. Although it is brought by the change of diffusion process, it is hard to justify "why we want to apply heat equation in generative models in the first place," considering there is less merit in doing so.
-- Moreover, there are also existing works that try to incorporate Gaussian blurring in the diffusion process. For example,

Lee, Sangyun, Hyungjin Chung, Jaehyeon Kim, and Jong Chul Ye. "Progressive deblurring of diffusion models for coarse-to-fine image synthesis." arXiv preprint arXiv:2207.11192 (2022).

Similar to this submission, their method applies a stochastic blurring process in diffusion models. While having differences, their quantitative and qualitative results are much better and match the same performance of standard diffusion models. I think comparing more carefully than citing them solely and indicating solutions to improve the current method is necessary.

-- Noise scheduling is less clear than the standard diffusion models, especially since this paper involves two hyperparameters, "sigma" (forward) and "delta" (backward), which, however, are unclear how to choose both of them and how to adapt with the blurring schedule (Gaussian kernel)

**Summary Of The Paper:**

This paper proposes a variant of the diffusion-based model with "inverse heat dissipation" (equivalent to continuous de-blurring Gaussian blur). Compared to the standard diffusion model, inverting the heat equation -- a PDE locally erases fine-scale information -- makes the generation consider multi-scale structures naturally. The paper also shows empirical evidence on emergent properties when applying IHDM, such as disentanglement of color and shape.

**Summary Of The Review:**

Overall, based on the review above, I think the methodology and the proposed method are technically reasonable and novel, which can be a good contribution toward a better understanding of diffusion-based models beyond Gaussian noises. However, as the concerns raised in the weakness, the poor visual performance and lack of proper comparison/justification with baselines decrease the actual impact.

---

> ### Author Response · Authors · 2022-11-07
> **Answer to reviewer dZbG**
>
> We thank the reviewer for the fair and very useful comments, and also noting the novelty of our method and analysis. We answer the concerns in the order received:
>
> - **Hard-to-see FID scores & baselines.** In order to save space, we did not add a separate table for the FID scores, and rather added them next to the first figure in the experiments section. We have now increased the font size, highlighted the FID scores in the caption and added FID comparisons to DDPM as a baseline and the current state-of-the art method on CIFAR-10, StyleGAN-XL. Aside from FID scores, we compared against a baseline DDPM model for the interpolation and few-shot learning experiments. However, we still stress that the aim of the paper was not to improve state-of-the-art FID scores, but instead to explore new methods for generative modelling, analysis, and providing new understanding. We believe that the paper is strong in these regards.
>
> - **Why we want to apply the heat equation in the first place.** Briefly: Our hypothesis was that the heat equation is a natural way to add inductive biases to diffusion-like generative models. To investigate this as lucidly as possible, we proposed a model that very explicitly reverses the path made by the heat equation. The result is that restricting the generative process to the deblurring path effectively regularises the model, raising properties of disentanglement, smooth interpolation and the potential to generalise better than standard diffusion with limited data (from 20 MNIST digits). One way to benefit from these insights while enjoying the sample quality of standard diffusion models might be to merge the heat dissipation process with them: E.g., blurring and increasing the noise level at the same time in the forward.
>
> - **Comparisons to parallel work.** We have now added much more extensive discussion concerning Lee et al. in the related works section. The main difference is that their forward process includes only a small amount of blur (total blur standard deviation of the order ~2.1 pixels for 64x64 images for the quartic schedule). The model is then quite close to a standard diffusion model (special case of no blur), whereas our proposed process is a much larger step away. We consider these two works complementary: We analyse the inductive biases of a generative process based strongly and explicitly on deblurring, whereas Lee. et. al. show that mixing an increasing noise level with blur can improve empirical results of standard diffusion models. Thus, again, a promising future direction for improvement could be finding different ways to mix standard diffusion and heat dissipation.
>
> - **Noise/blur scheduling.** We found it easy to pick good values for all the hyperparameters. The exact value of the training noise sigma is not critical, and we investigate this in App C.1. Furthermore, we did not find a reason to tune the blurring schedule to different sigma: Since the frequencies are decayed exponentially, the point at which each frequency component decays into noise is not changed dramatically with different sigma values. For the delta parameter we provide empirical results in Appendices C.1. and D.3. that the optimal proportion delta/sigma is quite stable and should be slightly larger than 1, and mathematical analysis in App A.4. That said, more statistical analysis on the optimal reverse noise would indeed be a valuable research direction. Similarly, there exists also interesting work in estimating the optimal reverse noise in diffusion models [1].
>
> - **EDIT:** We would also like to point out that the choice of forward process hyperparameters is, in one sense, even *less* heuristic than the standard diffusion scheduling: E.g., the commonly used cosine schedule [2] commonly used in diffusion models have been tuned using visual inspection, whereas we simply choose a blurring rate that corresponds to constant rate of resolution decrease (blur width is spaced logarithmically, corresponding to how a subsampling pyramid would work). This seems like a natural option.
>
> **References**
>
> [1] Bao, Fan, et al. "Analytic-DPM: an Analytic Estimate of the Optimal Reverse Variance in Diffusion Probabilistic Models." International Conference on Learning Representations. 2021.
> [2] Nichol & Dharival. "Improved Denoising Diffusion Probabilistic Models." International Conference on Machine Learning, 2021.

---

> > ### Comment · Reviewer_dZbG · 2022-12-08
> > **Thank you very much for your response!**
> >
> > I have checked the author's response, and I am satisfied with the response. Overall it is a good alternative to the standard diffusion model with Gaussian transformations. I would like to keep my score (6 acceptance).

---

### Author Response · Authors · 2022-11-15
**List of updates made and new interpolation video**

**Gathering together the updates to the paper**

For your convenience, we have gathered together the main changes made to the submission. If you have any more concerns or questions, we are happy to reply!
- **Evaluation (reviewers kPGb and dZbG):** We increased FID score font, added a pointer to them in the caption of Fig.5. and added explicit comparisons to DDPM and StyleGAN-XL (state-of-the-art) in the “Quantitative evaluation” section.
- **Comparisons to related work (reviewer dZbG):** We have added more extensive discussion on Lee.et.al [1] in the related works section, and included a suggestion on future improvement.
- **On Imagen and upsampling methods (reviewer D1k3):** We now cite more recent work with cascaded diffusion models, and we have clarified the phrasing on which sense diffusion models don’t take the inductive biases into account: “The forward and reverse processes of standard diffusion models do not explicitly consider the inductive biases of natural images”
- **Clarification on using noise in the model in the beginning (reviewer D1k3):** We have updated the abstract and introduction to state that “We interpret the solution of the forward heat equation with small additive noise”.
- **On alternative implementations of heat equation / blur (reviewer ZA3r):** We have now added a new section in the Appendix that discusses alternatives to the DCT-based forward process, and included the results of an experiment with convolutional filter-based blur. They were somewhat worse than the DCT-based blur.
- **PSD definition (reviewer ZA3r):** We have added more explanation on the PSD calculations in Appendix B.6.

**Interpolation video**

To give a better intuitive understanding on the difference between interpolation in a diffusion model vs. our model, we have added a new video in the supplementary (videos/interpolations.mp4). Check it out! We have also included interpolation using DDIM, a deterministic sampling method for diffusion models. In standard diffusion, the interpolation passes through features not present in either endpoint and the features jitter abruptly. With IHDM, the interpolations are more smooth, highlighting the difference in how the data distribution is modelled. Note that the diffusion model has not yet been trained to converge for these videos so the individual image quality itself is not comparable here, and in particular the colours are often a bit saturated for generated images.

**References**
[1] Lee et. al. “Progressive Deblurring of Diffusion Models for Coarse-to-Fine Image Synthesis”

---

### Decision · Program_Chairs · 2023-01-20

**Decision:**

Accept: poster

**Justification For Why Not Higher Score:**

Unfortunately, the empirical results are not strong enough and the community may not immediately switch from diffusion models to models proposed in this work.

**Justification For Why Not Lower Score:**

The spectral analysis and connection between heat equation and blurring can benefit the community in long run.

**Metareview: Summary, Strengths And Weaknesses:**

This submission builds on top of a classical connection between the heat equation and the blurring process and introduces a new generative model by inverting the blurring process. Spectral analysis reveals nice connections to the implicit inductive biases in diffusion models, showing that the proposed process has potentially better corruption mechanism for natural images. Overall, the reviewers have recognized the novelty of this work and the importance of the spectral analysis. However, their major criticism is around the empirical results that show significantly inferior performance compared to existing diffusion models. After careful consideration, the AC believes that in the long run, the research community can benefit from this work and therefore recommends accept.

Minor comment: The spectral inductive biases of diffusion models are also concurrently discussed in the following tutorial [here](https://youtu.be/cS6JQpEY9cs?t=2571). Please include a reference to:
Kreis et al. CVPR 2022 Tutorial on Denoising Diffusion-based Generative Modeling: Foundations and Applications.

**Note From Pc:**

if the above contains the word "oral" or "spotlight" please see: "oral" presentation means -> notable-top-5% and "spotlight" means -> notable-top-25%. As stated in our emails, we are disassociating presentation type from AC recommendations

**Summary Of Ac-Reviewer Meeting:**

N/A